# Benefits from high density rain gauge observations for hydrological response analysis in a small alpine catchment

Anthony Michelon[1], Lionel Benoit[1], Harsh Beria[1], Natalie Ceperley[1,2], Bettina Schaefli[1,2]

[1] Institute of Earth Surface Dynamics (IDYST), Faculty of Geosciences and Environment, University of Lausanne, Lausanne, 1015, Switzerland

[2] Now at: Institute of Geography (GIUB), Faculty of Science, University of Berne, Switzerland

*Correspondence to*: Anthony Michelon (anthony.michelon@unil.ch)

**Abstract.**

Spatial rainfall patterns exert a key control on the catchment scale hydrologic response. Despite recent advances in radar-based rainfall sensing, rainfall observation remains a challenge particularly in mountain environments. This paper analyzes the importance of high-density rainfall observations for a 13.4 $km^2$ catchment located in the Swiss Alps where rainfall events were monitored during 3 summer months using a network of 12 low-cost, drop-counting rain gauges. We developed a data-based analysis framework to assess the importance of high-density rainfall observations to help predict the hydrological response. The framework involves the definition of spatial rainfall distribution metrics based on hydrological and geomorphological considerations, and a regression analysis of how these metrics explain the hydrologic response in terms of runoff coefficient and lag time. The gained insights on dominant predictors are then used to investigate the optimal rain gauge network density for predicting the streamflow response metrics, including an extensive test of the effect of down-sampled rain gauge networks and an event-based rainfall-runoff model to evaluate the resulting optimal rain gauge network configuration. The analysis unravels that besides rainfall amount and intensity, the rainfall distance from the outlet along the stream network is a key spatial rainfall metric. This result calls for more detailed observations of stream network expansions, as well as the parameterization of along stream processes in rainfall-runoff models. In addition, despite the small spatial scale of this case study, the results show that an accurate representation of the rainfall field (with at least three rain gauges) is of prime importance to capture the key characteristics of the hydrologic response in terms of generated runoff volumes and delay for the studied catchment (0.22 raingauges/km²). The potential of the developed rainfall monitoring and analysis framework for rainfall-runoff analysis in small catchments remains to be fully unraveled in future studies, potentially including also urban catchments.

## 1 Introduction

Rainfall is known to be highly variable in space even at small scales, in particular in mountain areas (Henn et al., 2018; Tetzlaff and Uhlenbrook, 2005). Despite recent progress in the observation of spatial rainfall in mountainous areas with the help of radar (Berne and Krajewski, 2013; Germann et al., 2006; Germann et al., 2015), it remains crucially difficult to observe and spatially interpolate (Foehn et al., 2018a; Sideris et al., 2014).

Understanding the interrelation between spatial rainfall patterns and the hydrologic response has been of concern for many decades, ranging from a theoretical viewpoint (Shah et al., 1996; Singh, 1997; Woods and Sivapalan, 1999), to a rainfall-runoff model perspective (Obled et al., 1994; Nikolopoulos et al., 2011), and extending to a hydrological process understanding perspective (Guastini et al., 2019; Zillgens et al., 2007). Even earlier work in this field focused on the model-based

investigation of optimal rain gauge density for reliable areal rainfall estimation (Bras and Rodriguez-Iturbe, 1976a) and runoff prediction (Bras and Rodriguez-Iturbe, 1976b; Tarboton et al., 1987). Chacon-Hurtado (2017) provides a recent review on rain gauge network optimisation.

A wide range of methods has been proposed to analyze the hydrologic response as a function of spatial rainfall patterns. We can broadly distinguish between empirical methods that identify systematic response patterns by scrutinizing individual

observed events (Blume et al., 2007) and model-based methods that try to identify systematic or theoretical relationships between rainfall and the hydrologic response. In this latter category, we first of all find stochastic methods that describe the stochastic aspects of the hydrologic response as a function of the rainfall field properties. These approaches range from simplified stochastic models (Tarboton et al., 1987) to full space-time representations of rainfall forcing and streamflow generation (Mei et al., 2014; Pechlivanidis et al., 2017; Viglione et al., 2010; Woods and Sivapalan, 1999; Zoccatelli et al.,

2015). These stochastic tools are developed to understand the relative importance of the two key components of the hydrologic response, i) the runoff generation processes at the hillslope scale and ii) the routing mechanisms in the channel network. Such an assessment of the relative role of unchanneled-state and channeled-state processes (Rinaldo et al., 1991; Rinaldo et al., 2006a) gives key insights into the relative role of runoff generation processes and of the geomorphology of a catchment. This can also be achieved with virtual modelling experiments with hydrological models that explicitly account for

geomorphological dispersion along the channel network. An example is the work of Nicótina et al. (2008) who assessed the importance of well representing spatial rainfall variability for medium size catchments (a few hundreds to thousands km²) where saturation-excess overland flow dominates (rather than Hortonian flow). They conclude that for rainfall events with a spatial correlation length larger than the hillslope size, an exact representation of the spatial rainfall variability is not required to well represent the hydrologic response - provided that the mean areal rainfall is preserved at each time step. They explain

this result by the fact that if the total catchment-scale residence time is controlled by the travel time within the hillslopes, large enough rainfall events sample all possible residence times, independent of the actual spatial rainfall configuration. Their findings were subsequently confirmed by the work of Volpi et al., (2012) amongst others, where a simplified modelling approach based on a geomorphological unit hydrograph was used. While the conclusions were similar, this study also added that spatial variability does not matter "when the integral scale of the excess-rainfall field is much smaller or much larger than

the basin drainage area".

Similar results were obtained in studies that assess the impact of undersampling or of coarse graining an observed rainfall field on the performance of streamflow simulations obtained with more or less complex process-based hydrologic models (Bardossy and Das, 2008; Moulin et al., 2009; Lobligeois et al., 2014; Shah et al., 1996; St-Hilaire et al., 2003; Stisen and Sandholt, 2010; Xu et al., 2013). A key result of these model-based studies is that the hydrologic response depends more on the accurate

estimate of the mean areal rainfall than on the actual exact form of the rainfall field, (Obled et al., 1994). However, such model-based studies face the challenge that conceptual hydrological models require recalibration when used with different input fields, which makes disentangling effects from rainfall versus parameters a cumbersome exercise (Bardossy and Das, 2008; Bell and Moore, 2000; Stisen and Sandholt, 2010).

The above hypothesis that the mean areal rainfall might play a more important role for the streamflow response than the actual
spatial rainfall pattern is largely based on modelling experiments and remains to be tested in the field. In this paper, we therefore propose to investigate this hypothesis with a data-based framework to analyze the importance of rain gauge density for the event-specific hydrologic response (Ross et al., 2019) of a small, high elevation Alpine headwater catchment (13.4 km$^2$) where the hydrologic processes have been intensely monitored since 2015. Studying such a small catchment has, in addition, the potential to shed new light on the often used assumption that for catchments smaller than a few tens of km$^2$ a single rain gauge
is sufficient for reliable runoff prediction. While our analysis focuses here on a small natural headwater catchment, it is noteworthy that the developed rainfall monitoring and data analysis framework might also be of interest for  urban hydrology, which deals with similar questions regarding how spatial rainfall patterns, runoff generation processes and flow network geometry lead to peak flows in urban drainage systems (for a review, see the work of Cristiano et al., 2017).

To assess the number of point observations required to properly capture the hydrologic response of our target catchment, we
set up a dense rain gauge network made of commercially available and low cost devices, which increases the interest of this case study for future hydrologic studies in similar settings. These high-density rain gauge observations (approximately one rain gauge per km$^2$) are then used to answer two key questions:

i.    Which spatial characteristics of the rainfall field explain the timing and the amplitude of the hydrologic response?

ii.   What is the required spatial design of the rain gauge network to capture these characteristics?

To answer these questions, we developed a methodological framework to analyze the rainfall events, the hydrological response, and ultimately the optimal rain gauge density. This framework can be summarized as follows: i) define appropriate metrics to describe the rainfall fields and the hydrological response, ii) understand the relationships between these metrics through correlation analysis, iii) identify the main drivers (i.e. the corresponding metrics) through regression analysis, and iv) use the gained insights to optimize the rain gauge network based on selected metrics. We conclude the analysis with an event-scale
modelling of all recorded runoff response events with a semi-distributed model to evaluate the identified rain gauge network configuration.

The remainder of the paper is structured as follows. First, Section 2 describes the target area of the study, namely the Vallon de Nant catchment located in the Western Swiss Alps. Next, Section 3 presents the observational methods and the analysis framework. The results are presented in Section 4 and discussed in Section 5, with a focus on the impact of rainfall
heterogeneity on the streamflow response. Section 6 summarizes the main conclusions.

## 2 Study area

The area of interest is the Vallon de Nant, a 13.4 km² catchment located in the Western Swiss Alps (Figure 1). The elevation ranges from 1,200 m asl at the outlet of the Avançon de Nant river to 3,051 m asl (Grand Muveran) and has a mean elevation of 1,975 m asl. The catchment benefits from a protected status (Natural Reserve of the Muveran) since 1969 and is of national importance for Switzerland in terms of biodiversity (Cherix and Vittoz, 2009). The Vallon de Nant has been intensively studied over the recent years, in disciplines ranging from hydrology (Beria et al., 2020a) and hydrogeology (Thornton et al., 2018), to geomorphology and pedology (Lane et al., 2016; Rowley et al., 2018), to biogeochemical cycling (Grand et al., 2016), and to stream ecology (Horgby et al., 2019).

The Vallon de Nant belongs to the reverse side of the Morcles nappe, a structural geological unit that determines the catchment's shape. The old Cretaceous and Tertiary layers are recognizable as a succession of thick, blocky lithologies overlooking and surrounding the valley. They lie on a substratum of flysch, i.e. softer rocks (schistose marls and sandstone benches), which explains the deepening and widening of the valley at its southern part (Badoux, 1991).

Figure 2 summarizes the dominant hydrological units of the Vallon de Nant. The western side is mainly characterized by grassy slopes, with deep soils and a relatively high water storage capacity as revealed by gauging along the stream during the late summer and autumn yearly streamflow recession period (Horgby, 2019). The northern part of these western slopes shows a less dense drainage network than the rest of the catchment (Figure 1), explained by steeper slopes, a large hydraulic conductivity or locally deeper soils.

The eastern side of the catchment is characterized by steep and rocky slopes that react quickly to rain events due to shallow soils that drain quickly. At the foot of the rock walls, large alluvial cones and screes extend down to the river. The bottom of the valley is mainly composed of fine alluvial deposits with a large water storage capacity. In the southern part of the valley, the Glacier des Martinets (area less than 1 km²) is now confined to a small area shaded by the Dents de Morcles. The water flow paths of rainfall inputs over this southern (and higher elevation) part of the catchment, composed of moraines and permafrost, remain unclear and have not been investigated so far.

The Avançon de Nant river shows a typical snow dominated streamflow regime marked by a high flow period during spring and early summer when the snowpack accumulated during the winter melts (Supplementary Material Figure S1). The river length within the study area reaches 6 km in early summer, while during autumn and winter low flow, the river may start to flow as low as 1480 m asl (close to the gauge No. 5 on the Figure 1), reducing the instream flow distance to the outlet to 2.95 km. The actual extent of the stream network is based on observations during Summer 2017 (dry and wet periods) and its exact path was calculated using the Swiss digital elevation model at a resolution of 2 m (swissALTI3D, 2012).

The streamflow at the outlet is monitored via river height measurements using an optical height sensor and is converted into streamflow using a rating curve (Supplementary Material Figure S3) based on 55 salt streamflow measurements (Ceperley et al., 2018).

The annual mean streamflow in 2018 is between 0.60 and 0.72 $m^3.s^{-1}$ (between 3.89 and 4.61 $mm.day^{-1}$); mean annual water temperature is 5.0°C, ranging from a frozen river during some days in winter to a mean temperature of 8.5°C during summer (from July $1^{st}$ to August $31^{st}$, 2017). The maximum streamflow measured at the gauging station was between 10.4 and 12.4 $m^3.s^{-1}$ (between 67.2 and 80.0 $mm.day^{-1}$) during an intense rainfall event (August $6^{th}$, 2018).

Meteorological variables are monitored at three locations (Michelon et al., 2017) along a north/south transect (at 1253 m asl, 1530 m asl and 2136 m asl) since September 2016. From these stations, the mean air temperature at the mean elevation of 1,975 m asl is estimated to 3.1 °C in 2017.

We do not use any further data from the Swiss meteorological network since there are no ground measurement stations nearby, and the Vallon de Nant catchment is largely in the shadow of the Swiss weather radar network (Foehn et al., 2018b), which might see here at best rainfalls above 2800 m asl (Marco Gabella personal communication, February $27^{th}$ 2019).

## 3    Instruments and methods

### 3.1    Instruments

A network of 12 Pluvimate drop-counting rain gauges (www.driptych.com) was distributed across the Vallon de Nant catchment from July $1^{st}$ to September $23^{rd}$ 2018 to monitor rainfall (Figure 1). A similar deployment during the cold season would not be possible due to snowfall at all elevations throughout the winter. The sites were selected to represent the distribution of slope orientations and elevation, but also to meet constraints of accessibility and disturbance risk (livestock, hikers). The distance between measurement locations within the network ranges from 350 m to 1,550 m (630 m on average), and the greatest distance from any point in the basin to a rain gauge is 1,670 m.

The gauges are low-cost (around 600 USD each), consisting of a tube (11 cm of diameter, 40 cm of length) mounted to an aluminum funnel (Figure 2). The collected rainwater is concentrated to a nozzle that creates a drop of water of calibrated size (0.125 mL), which then falls on the impact-sensitive surface of the sensor, 30 cm below. The datalogger counts and records the number of drops over a time set up to 2 minutes. In the field, the devices are set up vertically, attached to a wooden stick. The funnel aperture is between 0.8 and 1.2 m above the ground.

The Pluvimates were set-up to count drops over an interval of 2 minutes, with an accuracy of 0.3 mm/h. Benoit et al. (2018a) experimentally evaluated the device uncertainty to 5 % for rainfall intensities under 20 mm/h. Given that some of the rainfall intensities measured in the present study exceed this value (intensities up to 140 mm/h were recorded), we extended the calibration to intensities up to 150 mm/h, and few saturation effects were noticed (Appendix A).

To prevent clogging, steel sponges were disposed in the funnel of each Pluvimate. This appeared to have caused i) a dampening effect on low rainfall intensities as it delayed slightly the beginning of very small events (lower than 1 mm/h) and ii) created drops remaining after the end of an event. The data are not corrected for these effects.

Additional artefacts were recorded, probably generated by strong winds creating resonance. Some stations in fact recorded very strong and highly variable rainfall over several hours during periods with high wind velocity but during days without any

observed rainfall in the combined MeteoSwiss radar-rain gauge data (Sideris et al., 2014). Four periods (over 4 different days) have been manually removed from the data.

## 3.2      Rainfall event characterisation

### 3.2.1      Event identification

Before further analysis, the rainfall amounts measured by each station were interpolated to a 10 by 10 m grid at a 2 min time
step using a high-resolution stochastic approach developed by Benoit et al. (2018a). In a nutshell, it generates an ensemble of stochastic space-time rain fields constrained by the actual observations at the rain gauge locations. The resulting ensemble (here composed of 20 realizations) can be used to analyze spatial rainfall uncertainty or to construct a single rainfall estimator. Following Benoit et al. (2018a), a non-separable and asymmetric covariance function was used to perform the simulations, which allows modelling rainfall advection and diffusion observed in the raw data. Areal rainfall time series are calculated for
each of the 20 realization, and from these a single time series (mean and standard deviation) of the areal rainfall.

Using the areal rainfall time series, the rainfall events are identified as periods with rainfall higher than 1 mm separated by at least 90 minutes with rainfall smaller than 1 mm. This duration of 90 minutes corresponds to the delay between the rainfall onset and the streamflow response for the large event recorded on August 23rd (for details see Supplementary Material), which occurred during an otherwise dry period. The streamflow response to the first half-hour of this rainfall event was caused only
by rainfall in the southern half of the catchment (stations 8 to 12), corresponding thereby to the most distant event (from the outlet). Accordingly, we assume that this event gives a rough estimate of the catchment's response time (Beven, 2020) i.e. of the time required until the entire catchment contributes to the streamflow response, including the delay caused by runoff transfer to the stream network and from there to the outlet from the hydrologically most distant parts of the catchment. The 90 minutes were therefore selected to maximize the chances of observing a distinct streamflow response for two distinct
consecutive rainfall events.

### 3.2.2      Spatial rainfall pattern metrics

Spatial rainfall patterns are classically characterized with geostatistical tools, including variograms (Berne et al., 2004) or with spatial moments of rainfall (Smith et al., 2002; Zoccatelli et al., 2011; Mei et al., 2014), in particular in presence of observed rainfall fields, e.g. from radar images. Here we propose to use more hydrological-process oriented metrics that explicitly
account for known features of the catchment and the stream network.

To build a first such metric, the catchment is split into two parts of equal area by a west-east line (Figure 1), delimiting an area close to the outlet in the northern part, and an area farther away in the southern part. This heuristic splitting into two parts is interesting here due to i) the elongated catchment shape and furthermore ii) the clearly distinct stream network organisation in the upper (southern) part of the catchment with more branching than in the northern part (reflected in the Strahler stream order
that does not further increase in the norther part, see Figure 1). Accordingly, we assume the rainfall events falling exclusively

on one or the other part of the catchment lead to a distinct streamflow response, with a faster and stronger response for events falling on the northern part (closer to outlet, steeper hillslopes, less storage potential than for the southern part).

The interpolated amounts of rainfall received by the southern and northern parts of the catchment, $P_{NORTH}$ and $P_{SOUTH}$, are compared and normalized by the total amount of rainfall to create an index of spatial rainfall asymmetry $I_{ASYM}$:

$$I_{ASYM} = \frac{P_{SOUTH} - P_{NORTH}}{(P_{SOUTH} + P_{NORTH})},$$     (1)

If rainfall is equally distributed between the northern and the southern parts, then $I_{ASYM} = 0$. The extreme values -1 and 1 express rainfall concentration exclusively in the northern or the southern part of the catchment, respectively. A value of -0.33 or 0.33 indicates that the catchment received at least 2 times more rain over one part of the catchment than the other..

To further analyze the relationships between the spatial distribution of rainfall and the streamflow response, we characterize

the geomorphological distance of incoming rainfall from the outlet, assuming that this distance should reflect to some degree the timing and the shape of the streamflow response of the catchment: following the terminology of Rinaldo et al. (2006b), transport at the basin scale can be analyzed in terms of travel in the unchannelled state (i.e. in the hillslopes) and travel in the channelled state (i.e. in the stream network).

Accordingly, we estimate for each rainfall event the weighted mean unchannelled distance to the stream network as:

$$D_{HILLS} = \frac{1}{t} \sum_t \frac{\sum_i \sum_j (P(i,j,t) d_{HILLS}(i,j))}{\sum_i \sum_j P(i,j,t)} ,$$     (2)

where $i$ and $j$ are the coordinates of rainfall location within the grid, $P(i,j,t)$ is the rainfall amount previously calculated using the stochastic method (section 3.2.1) for each of the 10 x 10 meters grid cell at each 2-minute time step $t$, and $d_{HILLS}(i,j)$ is the distance of this grid cell to the nearest stream network grid cell (following the line of steepest descent in the 2 x 2 m DEM (swissALTI3D, 2012)).

Similarly, we compute the weighted mean channelled distance between a point of introduction into the stream network and the outlet as:

$$D_{STREAM} = \frac{1}{t} \sum_t \frac{\sum_i \sum_j (P(i,j,t) d_{STREAM}(i,j))}{\sum_i \sum_j P(i,j,t)} ,$$     (3)

where $d_{STREAM}(i,j)$ is the distance along the stream network from the point of introduction to the outlet. For each cell of the stream network, this distance is calculated once based on the 2 x 2 m DEM.

It is noteworthy that these two metrics, $D_{HILLS}$ and $D_{STREAM}$ correspond to the aforementioned first order spatial rainfall moments, albeit decomposed according to hillslope and stream network distances, similar to what was proposed by Zoccatelli et al., 2015 in their analytical framework to quantify the smoothing of spatial rainfall organisation effects by channel residence time. It would be tempting to use also higher order rainfall moments; however, no significant correlation could be found to retained the streamflow metrics.

In addition to the above two metrics related to the theory of geomorphological dispersion (Rinaldo et al., 2006b), we use the height above the nearest drainage ($H_{HAND}$) terrain metric (Renno et al., 2008; Gharari et al., 2011; Nobre et al., 2011) to account for the topography. Based on the 2 x 2 m DEM, the normalized terrain heights $h_{HAND}$ are calculated by comparing the elevation

of each grid cell to the elevation of the nearest stream network cell in which the water is routed. The mean $H_{\text{HAND}}$ value for a rainfall event is given by:

$$H_{HAND} = \frac{1}{t}\sum_t \frac{\sum_i \sum_j (P(i,j,t) h_{HAND}(i,j))}{\sum_i \sum_j P(i,j,t)}. \tag{4}$$

The 3 distance metrics are computed with respect to both the dry and wet river network extent; the network extent to be used per rainfall event is then determined during the rainfall-streamflow response analysis (Section 3.4.1).

## 3.3    Streamflow response

### 3.3.1    Identification of streamflow events and fast runoff

The beginning and the end of each streamflow event are identified manually using a data visualization tool (developed in MathWorks MatLab 2017a, see Figure 3 and Figure 4). This choice of a visual expertise was made based on the observation that automatic identification of streamflow events would require almost a case-by-case filtering and parametrization, and thus would not be generalizable. This is partly related to a potentially high signal-to-noise ratio for river stage recordings during sediment transport events, a phenomenon potentially very important after a strong streamflow variation. The result of this 235 visual identification for each streamflow event is displayed in part 2 of Supplementary Material.

The beginning and the end of the streamflow response determine the initial and final baseflow; the streamflow volume above the straight line connecting these two points is considered here as fast runoff. It is noteworthy that we do not use peak streamflow to characterize streamflow events, for two reasons: i) given the small size of the catchment and the complex temporal distribution of rain intensities, the streamflow response has rarely a single, well identifiable peak (all events are 240 plotted in Figure S5 in Supplementary Material Part 1); ii) peak streamflow identification is further complicated by the noise in the stage recordings.

### 3.3.2    Streamflow metrics

The key metrics to characterize the streamflow response are the peak flow, the fast streamflow volume, the lag time elapsed between rainfall and streamflow response, and the flatting behaviour. For technical reasons we discarded the peak flow (see 245 section 3.3.1) and consequently the flatting behaviour. We use the fast streamflow volume through the runoff coefficient (RC), which is obtained by dividing the fast runoff volume by the total rainfall for the given event.

The lag time is usually defined as the elapsed time between the start of excess rainfall (the part of rainfall that causes the streamflow response) and the peak flow (McCuen, 2009). Since the start of excess rainfall is not known, the concept of peak flow is difficult to apply to our observed events (Section 3.3.1) and given the varying shape of our hydrographs, we empirically 250 tested different lag formulations; the lag between 1/3 of the rainfall event volume and 1/3 of the streamflow event volume gives the best results in the regression analysis, and is therefore retained. It is noted $\Delta_{\text{P/Q}}$ in the following.

### 3.4 Rainfall-streamflow response characterization

#### 3.4.1 Pseudo-dynamic stream network extent

The extent of the stream network evolves as a function of the catchment wetness conditions. Its minimal and maximal extent (Figure 1) are determined manually by identifying the uppermost points of the catchment where streamflow was observed in the field during summer baseflow (minimum extent, called *dry* state) and during summer high flow (maximum extent, called *wet* state).

In absence of exact observations of the stream network extent before the start of each streamflow event, we propose here to use a pseudo-dynamic stream network extent which assigns the dry or the wet to each streamflow. The network state is chosen based on a measure of the initial catchment wetness conditions, which is known to be the major variable explaining the dynamics of the hydrological response to different rainfall events (Penna et al., 2011; Rodriguez-Blanco et al., 2012), in particular through the creation of runoff thresholds (Zehe et al., 2005; Tromp-van Meerveld and McDonnell, 2006). Many studies use the baseflow before the start of a streamflow event as an indicator for the antecedent wetness conditions of the catchment. For snow-influenced catchments with a highly seasonal streamflow regime, this indicator might not reflect the actual wetness conditions. Hence, we rather quantify initial wetness conditions in terms of antecedent rainfall, i.e. using the cumulative rainfall (in mm) that occurred during a period from 1 to 5 days before a given rainfall event. The actual time span is selected based on a correlation analysis between antecedent rainfall over 1 to 5 days and the retained streamflow metrics (Section 4.2.1 and following).

This correlation analysis yields an optimum antecedent wetness indicator corresponding to the rainfall over the 3 days preceding the start of a rainfall event, noted $W_{3days}$. Using this indicator, the pseudo-dynamic network extent is obtained by assigning the dry network state to rainfall events that have $W_{3days} < 20$ mm and the wet network state to rainfall events that show $W_{3days} \geq 20$ mm. This threshold of 20 mm is selected by maximizing the correlation coefficient between $D_{HILLS}$ and RC (see Section 4.2.3).

#### 3.4.2 Regression analysis

We analyze the relationships between the spatial distribution of rainfall and the hydrological response based on a correlation analysis between the spatial rainfall pattern metrics (Section 3.2.2) and the streamflow metrics (Section 3.3.2) at the event scale, followed by a regression analysis to identify the key variables that best explain the runoff coefficient, RC, and the streamflow lag time, $\Delta_{P/Q}$. All used metrics are summarized in Table 1.

After the initial screening via correlation analysis, we use a pure quadratic regression to further investigate which combination of rainfall pattern metrics and initial wetness condition yields the best prediction of RC and $\Delta_{P/Q}$. Pure quadratic regression (i.e. without multiplication of explanatory variables) is chosen because the small number of observed streamflow events prevents using more complex models. Model selection is performed using the Akaike Information Criterion (AIC)(Akaike, 1974), noted here as $I_{AIC}$:

$$I_{AIC} = n \ln\left(\frac{S_{RSS}}{n}\right) + 2k + C,$$ (5)

where $n$ is the number of events, $k$ the number of coefficients, $S_{RSS}$ the residual sum of squares and $C$ a constant that can be ignored when comparing different models based on the same data set. As we manage small sample sizes (Burnham et al., 2011), we compute and use a corrected version of the AIC (AICc, noted here $I_{AICc}$):

$$I_{AICc} = I_{AIC} + \frac{2k(k+1)}{n-k-1}$$ (6)

For both AIC and AICc, the best model is the one having the lowest score.

## 3.5    Rain gauge network configuration analysis

Assuming that the actual rainfall measurement network is sufficient to capture the full spatial distribution of rainfall in the studied catchment, we assess the ability of partial networks to reproduce the identified best explanatory variables. The aim is twofold: i) identifying the best configuration for a future permanent observation network and ii) evaluate the added value of additional rain gauges in a partial network with respect to the identified key metrics (Section 4.4 and 5.2).

The quality of a partial network configuration is evaluated comparing the value (e.g. total rainfall) by event obtained with the partial network to the reference value obtained with the full network setup. We evaluate all the possible combinations of partial networks composed of less than 12 stations, i.e. 4094 possibilities. Each configuration is evaluated based on the root mean square error (RMSE):

$$\text{RSME} := \sqrt{\sum_{t=1}^{n} \frac{\left(X_k(t) - X_{ref}(t)\right)^2}{N}},$$ (7)

where $X_k$ is the selected rainfall metric (e.g. rainfall amount) at time step $t$ corresponding to the $k$-th network configuration, $X_{ref}$ the respective value obtained reference network set-up, and $N$ the number of time steps. The rainfall amounts measured by each station were interpolated to a 10 by 10 m grid at a 2 min time step using the Thiessen polygons method. The interpolation method developed by Benoit et al. (2018a) (see Section 3.2) cannot be used in this context because i) it requires at least 5 measuring points to perform adequately and ii) the computation time would be excessive to explore the 4094 combinations of stations for each event.

The best network for each number of stations is the one with the lowest RMSE. A sensitivity analysis is completed by removing from 1 to 3 rainfall events to the 23 events dataset, yielding 2047 datasets evaluated for each partial network configuration. The most frequent network configuration validates the robustness of the result.

## 3.6    Rainfall-runoff model

To further validate the obtained optimal rain gauge network configuration, we set up a a semi-distributed, event-based rainfall-runoff model. This model first simulates the mobilization of water at the sub-catchment scale (25 sub-catchments) using a Soil Conservation Service Curve Number (SCS-CN) approach (SCS, 1972). Next, the streamflow response is obtained by convolving the resulting hillslope responses with a travel path distribution derived from the stream network geometry (Schaefli

et al., 2014). The subcatchments and the stream network geometry are identified using *TopoToolbox* (https://topotoolbox.wordpress.com), in which travel paths correspond to the distance between the bottom part of each sub-catchment and the catchment outlet. In this model we focus on the fast response (i.e. runoff) of the catchment, and baseflow (defined here as the average discharge during the 30 min preceding event start) is subtracted from the actual discharge prior to runoff modeling. The model is calibrated against observed runoff (i.e. discharge - baseflow) through likelihood maximization assuming that the model residuals are normally distributed (e.g. Schaefli et al., 2007). The reference input field for model calibration is the mean of the 20 stochastic rainfall realizations at each time step (note since all realizations are conditioned on the observed precipitation events, this mean preserves the individual observed peaks of precipitation). After calibration the event-based runoff model is applied to the different network configurations to test how rain gauge network geometry influences the simulated runoff response. As the stochastic rainfall interpolation cannot be performed with a number of observation points as low as 3 stations (or less), we use the Thiessen polygons method to interpolate the rainfall fields from the 1 to 3-station rain gauge network obtained during optimal network analysis. .

## 4  Results

### 4.1  Rainfall events

#### 4.1.1  Areal rainfall and asymmetry

The available 3-month measurements window between July $1^{st}$ and September $23^{th}$ 2018 captured 48 rain events (detailed in the part 2 of the Supplementary Material) for a total areal rainfall amount of 317.8 mm. The areal rainfall amount per event ranges from 1 mm to 43.5 mm (mean of 6.6 mm), and event duration ranges from 32 minutes to 10.5 hours (mean of 2.8 hours); these records do not show any evidence of altitude effect on the rainfall amount ($R^2 = 0.06$). Despite the sequential deployment of the 12 rain gauges and other technical issues (see section 3.1), the rainfall events were all measured by at least 7 stations; 36 out of 48 events were recorded by at least 10 stations and 23 events were recorded by 12 stations. The different subsets used in this study are detailed in Table 2. Details for all recorded rainfall events and the corresponding streamflow are shown in summary plots, as illustrated in Figure 3 and Figure 4 (all events are presented in the Supplementary Material). Most events show a relatively homogeneous spatial distribution of rainfall events (see an example in Figure 4), with only few events showing a strong asymmetry (Figure 5): the correlation between $P_{NORTH}$ and $P_{SOUTH}$ equals 0.91, with a median $I_{ASYM}$ of 0.025. Interestingly, strong spatial asymmetry mainly affects events with low rainfall amounts, with 7 out of 8 events with $|I_{ASYM}| >$ 0.33 receiving below 5 mm (Figure 5). For the events that actually triggered a streamflow response, the correlation between $P_{NORTH}$ and $P_{SOUTH}$ is thus significantly higher (r=0.69, Table 4).

One strong asymmetric and very intense event occurred on July $24^{th}$ at 6:32 PM (Figure 3). The rainfall map shows a heterogeneous distribution of rainfall, centered close to the outlet in the northern part of the catchment, over 6 out of the 12 stations. One of the rain gauges recorded up to 35.3 mm of rainfall, whereas 1.8 km upstream, half of the stations (on the

southern and western parts of the catchment) did not record any rainfall. The interpolated amount of rainfall over the basin was $8.0 \pm 1.3$ mm, and a fast runoff volume between 28.3 and 32.5 mm was measured, resulting in a runoff coefficient between 3.0 and 4.8 that remains difficult to explain. One possible explanation is that important rainfall amounts fell on the north-eastern part of the catchment, over steep slopes that are difficult to access and were therefore not gauged. This event and its streamflow response are excluded from further analysis involving the hydrological response (see also Section 4.2 and the summary of analysed events in Table 2).

### 4.1.2    Geomorphological and topographical distance metrics

For the 48 recorded rainfall events, the three distance metrics $D_{\mathrm{HILLS}}$, $D_{\mathrm{STREAM}}$ and $H_{\mathrm{HAND}}$ show significantly different median values if they are computed with respect to the wet network than with respect to the dry network; we can reject for each metric the hypothesis that they have the same median value for the wet state and the dry state with a Wilcoxon rank sum test at level 0.05 (see distributions in Figures S6 and S7 of the Supplementary Material part 1). However, each of the distance metrics shows a strong correlation between its values for the wet and for the dry network state (from 0.94 for $H_{HAND}$ to 1.00 for $D_{\mathrm{STREAM}}$, Figure 7). The between-metric correlation for all 48 rainfall events (Table S2 in Supplementary Material part 1) ranges for the wet state range from 0.78 ($D_{\mathrm{HILLS}}$ - $D_{\mathrm{STREAM}}$) to 0.95 ($D_{\mathrm{HILLS}}$ - $H_{HAND}$) and for the dry state from 0.70 ($D_{\mathrm{STREAM}}$ - $H_{HAND}$) to 0.95 ($D_{\mathrm{HILLS}}$ - $H_{HAND}$). Considering only the rainfall events with streamflow response, these correlations are slightly lower (Table 3), but with a clear correlation between $D_{\mathrm{HILLS}}$ and $H_{HAND}$ for both the wet and the dry state; accordingly, we do not further use the $H_{HAND}$ metric in this analysis. None of the distance metrics shows a strong correlation (>0.6) with the rainfall spatial distribution metrics, i.e. $P_{\mathrm{SOUTH}}$, $P_{\mathrm{NORTH}}$ or $I_{\mathrm{ASYM}}$. They also do not show any correlation higher than 0.6 with the hydrologic response metrics (Table 4). This confirms our hypothesis that the network state needs to be included in a dynamic way (see Section 4.2.3).

### 4.1.3    Temporal evolution of rainfall metrics

We computed the temporal evolution of the rainfall metrics to unravel potential temporal evolution patterns in $I_{\mathrm{ASYM}}$, $D_{\mathrm{HILLS}}$ and $D_{\mathrm{STREAM}}$ and their relation to the streamflow response (full results are available in the Supplementary Material part 1). The temporal evolution of the two distance metrics is overall rather flat with no clear fluctuation patterns. There is only one event with a pronounced temporal trend for $D_{\mathrm{HILLS}}$ (Q event #1).

For $I_{\mathrm{ASYM}}$, some events show interesting temporal patterns. For example, during the double peak runoff of Figure 3, $I_{\mathrm{ASYM}}$ shows an almost constant negative value suggesting that the corresponding double peak rainfall event remained stationary on the northern part of the catchment over its entire duration and therefore caused the double peak streamflow response.

For the first two streamflow events, the $I_{\mathrm{ASYM}}$ metric switches from strongly positive to close to zero during the event, implying that the rainfall field moved towards the outlet during the event; in other words, the rainfall cloud follows the overall water movement through the catchment and thereby leads to a stream response concentration. This might explain why these two

events are the only ones that show a pronounced single peak streamflow response. However, given the low number of observed events and the diversity of temporal patterns, these insights cannot be further used for a quantitative analysis.

## 4.2 Hydrologic response

### 4.2.1 Observed streamflow events

For 13 days (6 of the 48 rainfall events), the water stage sensor was disturbed by the proximity of a rock (see picture of the Figure S2 in Supplementary Material), resulting in missing streamflow data. For the remaining 42 rainfall events, a streamflow response was observed for 15 of them (see Table 2 and Table 3).

The fast streamflow volume during these events, $Q_{FAST,}$ shows a strong correlation with total rainfall and with $P_{SOUTH}$ (Figure 8a); however, the event on 24 July with only 8.0 mm of rain and 30.4 mm of fast streamflow falls far away from this
relationship, which further motivated the exclusion of this event from the analysis.

The 14 remaining events are distributed over the entire observation period, covering a wide range of streamflow conditions, which is reflected in the initial streamflow before each event, ranging from 7.9 mm in early July to 2.6 mm by mid-September (Table 3), with an almost linear decrease between the dates (correlation between initial streamflow and day of the year of - 0.90, see also Figure S3 in the Supplementary Material).

The correlation of this initial flow before events with $Q_{FAST}$ or with the runoff coefficient RC is extremely low (correlation of -0.02 and -0.05), which confirms our hypothesis that antecedent streamflow is not a good proxy for antecedent moisture.

The highest correlation between RC and antecedent precipitation occurs for a time span of 3 days preceding the streamflow event (0.67); this metric, called $W_{3 days}$, is thus retained as a proxy for antecedent moisture for further analysis. The role of initial wetness conditions can also be discussed more qualitatively by comparing a pair of rainfall events with very similar
spatial patterns and amounts (Figure 4). For the first event (24 August), the measured rainfall ranges from 6.2 mm to 11.8 mm, corresponding to 8.5 mm of rainfall over the catchment in 2 h 38 min. For the second event (29 August), the rainfall ranged between 5.4 mm and 11.4 mm, corresponding to 8.4 mm over the catchment during 1 h 14 min. Despite the similar total amount of rainfall and event duration (during the first event 76 % of the total rain happened for a duration similar to the second event), the first event shows a fast runoff volume of 7.4 mm, whereas for the second event the streamflow response is almost
invisible. This difference can be explained by the initial wetness conditions, with 29.5 mm of rainfall during the 3 days preceding the first event, compared to 12.4 mm for the second event.

### 4.2.2 Streamflow generation processes, RC and lag

The correlation analysis (Table 4) reveals a strong correlation between rainfall amounts and $Q_{FAST}$ (0.77, Table 4). This suggests that streamflow responses are triggered by saturation-excess, rather than by infiltration capacity-excess: If saturation
is exceeded, every unit of rainfall leads to a corresponding unit increase of streamflow, which in turn leads to a strong linear correlation between rainfall amounts and fast streamflow volumes. Furthermore, saturation-excess also implies that a longer

rainfall event leads to a higher streamflow response volume (once the saturation threshold is reached, all rainfall contributes to streamflow). This is confirmed by the high correlation (0.74) between the rainfall duration $P_{\text{DURATION}}$ and $Q_{\text{FAST}}$. If, on the contrary, the driving process was the exceedance of the soil infiltration capacity, then only rainfall intensities above the

capacity threshold would trigger a corresponding streamflow increase; small rainfall amounts would trigger almost no response. In this case (infiltration-excess), there would be no linear correlation between rainfall amounts or rainfall duration and streamflow amounts, but a strong correlation between fast streamflow amounts and high or maximum precipitation intensity; positive correlations between $Q_{\text{FAST}}$ and $P_{\text{max ALL}}$, $P_{\text{max NORTH}}$ or $P_{\text{max SOUTH}}$ are however all absent (values of -0.17, -0.16 and -0.08, Table 4). In addition, saturation-excess as a main driver of the fast streamflow response is further confirmed

by the clear threshold effect for the generation of streamflow as a function of total event rainfall (Figure 8); a streamflow response only occurs for total rainfall higher than 5 mm

This threshold effect supports the formulation of the lag time $\Delta_{\text{P/Q}}$ as the time between one third of the rainfall event volume and one third of the streamflow event volume, since a lag time between the starts of the events would here be misleading. Accordingly, the streamflow events show a relatively strong correlation (0.71, Table 4) between the RC and the lag $\Delta_{\text{P/Q}}$: we

observe a higher RC when the level of saturation increases; reaching such a higher level of saturation requires more time, which results in a longer lag before a significant amount of streamflow reaches the outlet.

We furthermore find a positive correlation between $I_{\text{ASYM}}$ and the lag $\Delta_{\text{P/Q}}$ (0.59, Table 4), which supports our initial assumption that negative $I_{\text{ASYM}}$ values (corresponding to rainfall concentrated on the northern part, close to the outlet) correspond to low lag times. However, the assumed negative correlation between RC and $I_{\text{ASYM}}$ (higher RC values for rainfall events with

negative $I_{\text{ASYM}}$ values) is not confirmed by the observed data (the correlation is 0.44, Table 4), thereby not confirming our hypothesis that rainfall on the northern catchment part (showing less water storage potential) leads to more fast streamflow. However, there is also a strong negative correlation between $\Delta_{\text{P/Q}}$ and the maximum rainfall intensity over 10 minutes, which is stronger for $P_{\text{max NORTH}}$ (-0.71, Table 4) than for $P_{\text{max SOUTH}}$ (-0.58). This probably reflects the fact that in the northern part of the catchment, there is a lack of soil storage capacity due to the large rock walls on the right stream side, which is not

compensated by the available soil storage on the left stream side, with ensuing Hortonian (infiltration-excess) streamflow generation processes becoming more important in the northern part than in the southern part of the catchment. This significant difference in streamflow generation processes is also visible in the drainage density, which is higher on the right stream side in the northern part than on the left stream side (Figure 1).

### 4.2.3    Dynamic stream network state

As discussed in 4.1.2, the rainfall distance metrics if computed with respect to the dry or the wet stream network state show very low correlations with the streamflow metrics. Accordingly, we attribute either the dry or the wet network state to each streamflow event as a function of the antecedent wetness $W_{3\text{ days}}$, which is used as a measure for the stream network expansion. In the following, we call these new distance metrics "pseudo-dynamic" since only two different states are observed. Setting a $W_{3\text{ days}}$ threshold to 20 mm to discriminate between the dry and the wet state yields correlations between $D_{\text{HILLS}}$ – pseudo-

dynamic and RC of -0.70 and between $D_{HILLS}$-pseudo-dynamic and $\Delta_{P/Q}$ of -0.66 (Table 5). $D_{STREAM}$ – pseudo-dynamic shows correlations of 0.53 and 0.60 with the RC and with the $\Delta_{P/Q}$, and we retain both pseudo-dynamic distances for further analysis. A sensitivity test showed that setting a $W_{3\,days}$ threshold of between 12 mm and 20 mm to discriminate between the dry and the wet state yields very similar results, and accordingly, we retain a threshold of 20 mm for $W_{3\,days}$ to compose the pseudo-dynamic network state. It should however be kept in mind that these pseudo-dynamic distance metrics represent simply a heuristic solution to overcome the absence of detailed stream network state observations before each event.

### 4.3 Identification of dominant hydrologic drivers via regression analysis

The above correlation analysis results in a range of potential explanatory variables for RC and $\Delta_{P/Q}$ referring to the rainfall amounts, maximum intensity and asymmetry, the pseudo-dynamic rainfall distance metrics and initial wetness conditions ($W_{3\,days}$). However, according to the correlation analysis, we retain the maximum rainfall intensities as explanatory variables only for $\Delta_{P/Q}$. The tested models, based on one or two explanatory variables, are summarized in Table 6 for RC and in Table 7 for $\Delta_{P/Q}$. The analysis is based on 14 events (after removing the 24 July event, subset #4 of Table 2) and the best models are selected based on their AICc ranking and coefficient of determination ($R^2$).

The best ranked model (in terms of AICc) for RC is a single predictor model using $D_{STREAMS}$ (pseudo-dynamic) as explanatory variable, which yields better results than using antecedent moisture $W_{3\,days}$ as a single predictor; it should be kept in mind here that the pseudo-dynamic distance metrics also embed information on antecedent moisture conditions (since $W_{3\,days}$ decides on the moisture state). However, the $R^2$ becomes considerably higher (0.75) using $P_{ALL}$ and $D_{STREAM}$ (pseudo-dynamic) as explanatory variables. Slightly less good results are obtained with $D_{HILLS}$ (pseudo-dynamic) as a single predictor or in combination with $P_{SOUTH}$. The fact that $D_{STREAM}$ (pseudo-dynamic) plays a prominent role to explain the RC might be surprising; a possible explanation lies in the fact that the length of instream flow paths is also a metric for runoff storage and exchange within the riparian area, especially in the southern part of the catchment.

For $\Delta_{P/Q}$, the best model (in terms of AICc) has the two explanatory variables $P_{max\,SOUTH}$ and $I_{ASYM}$ with a $R^2$ of 0.83 and is considerably better in terms of $R^2$ than any single predictor model. The best model including a distance metric is $P_{max,All}$ in combination with $D_{STREAM}$ ($R^2 = 0.78$), which underlines the prominent role of $D_{STREAM}$ (pseudo-dynamic) to explain the hydrologic response in this catchment.

### 4.4 Measurement network analysis

#### 4.4.1 Raingauge density analysis

During the observation period, 23 out of 48 events (subset #2, Table 2) were captured by the full network of 12 stations, measuring a total amount of rainfall of 120.7 mm. We tested what a partial rain gauge network (all possible combinations of networks composed with less than 12 stations) would record compared to the full rain gauge network of 12 stations taken as a

reference, using the Thiessen polygons method to interpolate the rainfall fields (since, as discussed earlier, the stochastic method cannot be applied to a small station number).

Figure 6a shows, in term of rain gauge density, the number of events having the total amount of rainfall $P_{ALL}$ overestimated or underestimated by a factor 2. We globally observe a misestimation inversely proportional to the rain gauge density, with up to 3 events overestimated by a factor 2 and 8 events underestimated by a factor 2 with the lowest rain gauge density of 0.07 rain

gauge per km² (1 rain gauge). It is necessary to reach 0.82 rain gauges per km² (11 rain gauges) to no longer have events misestimated by a factor 2. In presence of few rain gauges, Figure 6a also shows a strong tendency to underestimate rather than overestimate rainfall amounts. This can be explained by the fact that for a heterogeneous rainfall event, it is more likely to miss a localized important part of the rainfall field rather than to capture it.

Figure 6b presents in the same way the maximum error encountered on the maximum rainfall intensity over 10 minutes

$P_{MAX}$(10 min). We notice the expected inversely proportional trend, reducing the error if the rain gauge density increases. The figure also shows that in general a low rain gauge density tends to overestimate more than underestimate the $P_{MAX}$(10 min). This bias originates from the large footprint associated to each station in presence of a low rain gauge density, increasing the disparities between the observation points while interpolating the rainfall fields.

### 4.4.2     Optimum network identification

Based on the hydrologic driver analysis, we retain $P_{ALL}$, $P_{max,ALL}$, $I_{ASYM}$ and $D_{STREAM}$ (pseudo-dynamic) as key metrics for the optimal rain gauge network analysis. Figure 10 shows the best network configurations for 1 to 5 stations and the corresponding RMSE for the select reference metric for the network optimisation (one metric per line).

For a 1-station network, $P_{ALL}$ is best captured when the station is located in the middle of the catchment, while a 2-station network improves substantially the RMSE by arranging the measuring points between the northern and southern parts.

Additional stations still improve the RMSE, although to a lesser extent. With a 4-station and 5-station network, the stations tend to align along a north-south transect. For $I_{ASYM}$ and $P_{max,ALL}$, we see very similar evolution of the spatial patterns as for $P_{ALL}$ for increasing network sizes; for $P_{max,ALL}$, the RMSE continues however to considerably decrease with the number of stations, which is to be expected for this measure that is more sensitive to spatial-temporal variations of rainfall amount.

For $D_{STREAM}$ as a network optimisation metric, the optimal network configuration first selects stations at the extreme ends of

the stream network before organizing along a transect as for the other metrics, with one lateral station on the left stream side included in the 5-station network as for $P_{max,ALL}$ (the same) and for $I_{ASYM}$ (a different one).

Considering the small dataset underlying this analysis (23 events), the robustness of the best networks is assessed for two selected metrics (for the $P_{ALL}$ and $I_{ASYM}$) by re-computing the optimal network if between 1 and 3 events are removed from the dataset. Figure 11 shows how frequent a given configuration is identified as being the optimal solution for networks composed of 1 to 3 stations and clearly confirms the optimal solutions found previously.


### 4.4.3 Optimum network evaluation

To evaluate this optimum network analysis, we compare in a first step the RC and lag time $\Delta_{P/Q}$ obtained from the full stochastic rainfall field (median field) to the RC and $\Delta_{P/Q}$ values obtained from the best 1-station and 3-station networks and from the worst 3-station network (Figure 12). The corresponding rain gauge densities are 0.07 rain gauge per km² for a 1-station network, 0.15 rain gauge per km² for a 3-station network and 0.90 rain gauge per km² for the full network. For both the RC and $\Delta_{P/Q}$, the dispersion of the values obtained with the reduced rain gauge network decreases from the best 1-station network to the best 3-station network but remains sensibly the same for the worst 3-station network, underlining thereby that a 3 rain gauge can give could results conditional on a good location selection.

It is noteworthy that for the lag, even a 1-station network can reproduce this metric correctly for most of the events but can also be completely off (Figure 12). With the best 3-station rain gauge network, the RMSE with respect to the full stochastic rainfall field reduces from 23.18 to 8.12 compared to the best 1-station network.

In a second evaluation step of the identified optimum rain gauge network, we simulated the event-based streamflow response for the best 1-station network and the best and the worst 3-station network, to compare the result to the simulation with the original rainfall field and thereby obtain a validation on the entire streamflow dynamics rather than on RC or lag only (all simulations are available in Supplementary Material part 1). It is important to point out here that the semi-distributed hydrological model cannot reproduce all observed events equally well as shown by low correlation coefficients between observed and simulated streamflow in Figure 13. Even with the stochastic generation of rainfall fields, fast streamflow tends to be underestimated with the model; improving the simulation quality for all events would require an in-depth analysis of different subsurface flow mechanisms related also to snow melt and shallow-groundwater recharge, work that is ongoing in this catchment (Beria, 2020b).

Despite of this, we clearly see that the best 1-station network and the worst 3-station network considerably underperform with respect to the full network and that the best 3-station network yields a simulation performance close to the original rainfall field, confirming the results obtained for the summary streamflow response metrics RC and lag.

## 5 Discussion

### 5.1 Spatial heterogeneity of rainfall

One of the key identified metrics to characterize the spatial distribution of rainfall in relation to RC and lag prediction is $I_{ASYM}$. It splits the catchment into two parts, and aggregates rainfall observations into one value. Among the records showing a strong rainfall asymmetry, 7 out of the 8 events are too small to cause a detectable streamflow response (Figure 5), but one does create a streamflow response although it only rains over half of the 12 rain gauge stations. Despite of this absence of a strong asymmetry in the 14 rainfall events that cause a streamflow response, the regression analysis suggests that the spatial distribution might play an important role for the explanation of the lag time. The importance of this asymmetry predictor can

be related to the fact that it captures the key feature of the spatial catchment organisation in terms of distance to the outlet, drainage density and subsurface storage potential.

The second dominant metric of spatial rainfall distribution to predict the RC and the lag is $D_{STREAM}$ (pseudo-dynamic). This suggests that for this catchment, the rainfall distance to the outlet is the overall the dominant predictor for the analyzed streamflow response metrics.

It is noteworthy that this analysis could be affined by investigating different splitting geometries, e.g. by splitting the catchment into west and east parts, thereby separating the large slopes (west) from the steep slopes (east). This and similar spatial asymmetry metrics are case-specific as they rely on the particular geomorphology and topography of the catchment and are thus not directly applicable to other catchments. In particular $I_{ASYM}$ cannot be used as a tool to compare different catchments. The rainfall distance metrics to the stream network ($D_{HILLS}$) and along the stream network ($D_{STREAM}$) were designed here to overcome the limitations of the simple asymmetry measure. The prominent role of $D_{STREAM}$ - pseudo-dynamic to explain the lag time and RC underlines the importance of characterizing the spatial heterogeneity in terms of geomorphological distances to the actual stream network, which requires more detailed network expansion analyses in future studies.

We could expect that in that kind of steep environments, the residence time in hillslopes strongly dominates over residence times in the stream network (Nicotina et al., 2008); the fact that $D_{STREAM}$ outperforms here $D_{HILLS}$ for the prediction of RC and lag time may show that even in steep environments, with a priori fast instream processes and limited storage, the riparian area and related subsurface exchange processes could play a more prominent role. The fact that the travel distance in the stream network explains more of the RC variation than $D_{HILLS}$ might be an indirect effect: the longer the travel distance in the stream network, the more likely are delays due to exchange with groundwater in the riparian area. This implies that along-stream processes might need a better representation in rainfall-runoff models, even for small and steep catchments; to date, these processes are often ignored in rainfall-runoff hydrological models at this scale, or are represented with a simple constant velocity transport term (e.g. Schaefli et al., 2014).

However, future work on the role of water residence time in the stream network will necessarily require more detailed field data on the temporal evolution of the stream network. This will in addition open new perspectives to quantify how the stream network extension is imprinted in the streamflow response: in fact, as discussed by Rinaldo et al. (1995), the intrinsic fractal nature of the stream network is not transferred to the streamflow response and, accordingly, there is potential to infer the stream network extension from observed streamflow records, provided that we have high resolution rainfall data to disentangle the different effects. Finally, we would like to point out here that this result on the prominent role of travel time along the stream network opens interesting new analogies with urban hydrology, where introduction times to the network are typically short (Smith et al., 2013). Future work might show what methods from urban hydrology (Cristiano et al., 2017) could be transposed to the analysis of spatial rainfall variability in small alpine catchments.

## 5.2 Rain gauge network density

The selected metrics showed the importance and potential of a high density rain gauge network to capture rain events, and to investigate the dynamics of the hydrologic response. The rain gauge network analysis can then be used as a preliminary investigation to implement a permanent network, composed of fewer stations. The reliability of the study is directly dependent on the number of observed rainfall events, i.e. on deployment duration of the rain gauge network. Despite the small size of the catchment, there could potentially be storms that are not or only partially seen by the rain gauge network.

This possibility of missing localized events is highlighted by the event of July 24[th] (Section 4.1.1), which was considerably underestimated despite of the high density of the deployed network (1 station for 0.9 km² on average, maximal distance of 1,670 m from a point to a rain gauge). The best partial networks composed of 1, 2 or 3 stations (Section 4.4) give for this extremely localized event a total amount of rainfall respectively 12.0 mm, 9.4 mm and 9.2 mm, not far from the 10.6 mm measured with the full network, but these partial networks were trained on the dataset containing the particular event.

With only one station, there is a high risk of totally missing an event, whereas a 2-station network design measuring at least the northern and the southern part of the catchment would i) capture most of the events and ii) give a first estimation of the rainfall spatial distribution.

Overall, the network optimisation analysis with different metrics clearly suggests that to optimally reproduce the hydrologic response in terms of RC and $\Delta_{P/Q}$, we would need to implement at least a three station network in this catchment, organized along a north-south transect, with one of the stations being located in the remote southern part. The north-south organization can be explained by i) the shape of the catchment that also extends longitudinally or ii) a general tendency for rainfall events to move longitudinally, emphasizing the importance, for this case study, to capture spatial configuration of rainfalls over a north-south transect rather than over a west-east transect and iii) the general increasing trend of elevation along this transect.

## 6 Conclusion

Our analysis of the role of rainfall patterns for the streamflow response is one of the first data-based studies carried out at such a small scale in an Alpine environment. The detailed analysis of 48 events from one summer suggests that spatial rainfall patterns might play a key role to explain the hydrologic response in small Alpine catchments. The novelties of the study include the use of a low-cost rain gauge network to capture rainfall patterns, and the design of a data-based framework to analyze the rainfall-runoff response. The main conclusions from our analysis are:

- A high density rain gauge observation network is a major asset to identify critical areas that are influenced by local rainfall forcing, and give an estimation of the rainfall amount errors made by a partial network.
- A detailed analysis of the hydrological response as a function of rainfall patterns and geomorphology requires a rain gauge network specifically designed for this purpose in conjunction with detailed observations of the stream network expansion before events.
- Such a network should take into account the spatial distribution of distances to and along the stream network.

•    As shown here, even for small catchments the rainfall distance to the outlet along the stream network might play a key role to explain the hydrologic response. Accordingly, future hydrological modelling studies in small Alpine catchments should investigate the representation of instream transport and storage processes.

The analysis framework developed here is readily transferable to other settings, including natural or even urban catchments. Given the low cost of the deployed rainfall sensor network, the approach has potential for future detailed studies
in to-date sparsely gauged catchments.

*Data availability.* Rainfall and streamflow data used for this paper, and the MatLab code written to visualize the data are available online at https://doi.org/10.5281/zenodo.3946242.

*Code availability:* The stochastic space-time rain field generator of Benoit et al., (2018a) is freely available at: https://github.com/LionelBenoit/Local-rainfall-model

*Author contributions.* AM and BS conceived the ideas and designed methodology; AM, LB and HB collected the rainfall data; AM and LB analyzed the data; AM and BS led the writing of the manuscript. All authors contributed critically to the drafts
and gave final approval for publication.

*Competing interests.* Author BS is a member of the editorial board of the journal, but otherwise there are no competing interests that the authors are aware of.

*Acknowledgements.* The work of the authors is funded by the Swiss National Science Foundation (SNSF), grant number PP00P2\_1576

**Appendix A: Drop-counting rain gauge calibration and data correction**

Technical characteristics of the Pluvimate drop-counting rain gauges (see Section 3.1) are detailed in the work of Benoit et al. (2018a); for this study we extended the experimental tests to intensities up to 150 mm/h. It appears that for intensities up to 20

mm/h (99.88 % of the measured 2-min intensities during the 2018 observation period, see Figure A1) the linear relationship between drop count and rain intensity gives a good estimate (uncertainty below 5 %); beyond 20 mm/h the linear relationship underestimates the rainfall intensities, to reach 10 % of error at 60 mm/h and 15 % at 150 mm/h (Figure A1). For this study, rainfall intensities over 20 mm/h are corrected using a polynomial law based on the experimental measures.

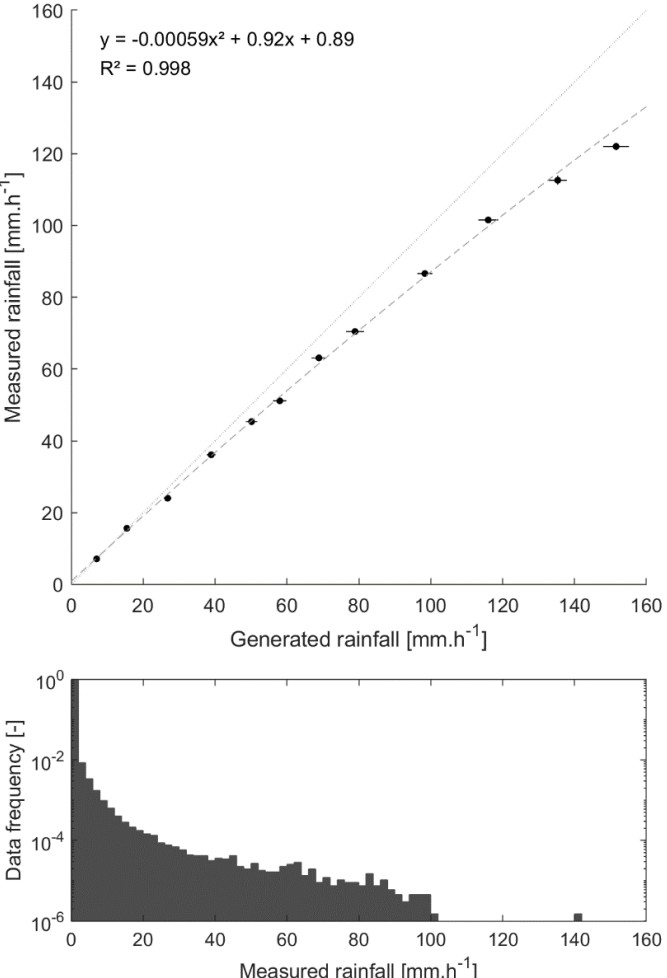

Figure A1. Calibration curve (on top) of the Pluvimate rain gauges based on experimental measures with controlled rainfall input, and (at the bottom) the data frequency measured in situ.

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

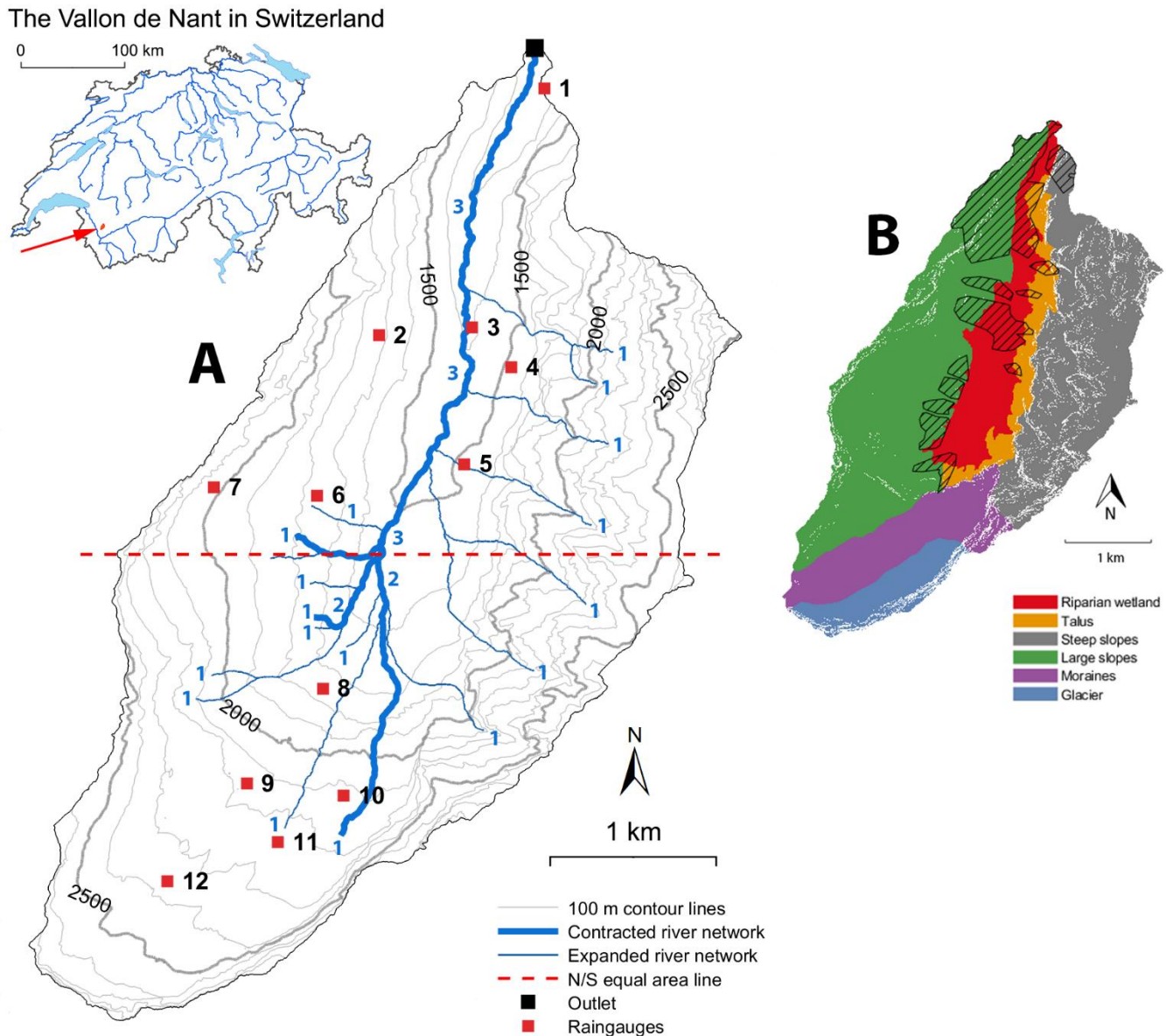

Figure 1. Map of the Vallon de Nant and location of the 12 rain gauges. The streamflow is measured on the main river at the outlet (46.25301 N / 7.10954 E in WGS84 coordinates). The red dashed line splits the catchment area into two parts of equal area. The small numbers next to the streams indicate the Strahler stream order (Strahler, 1957).

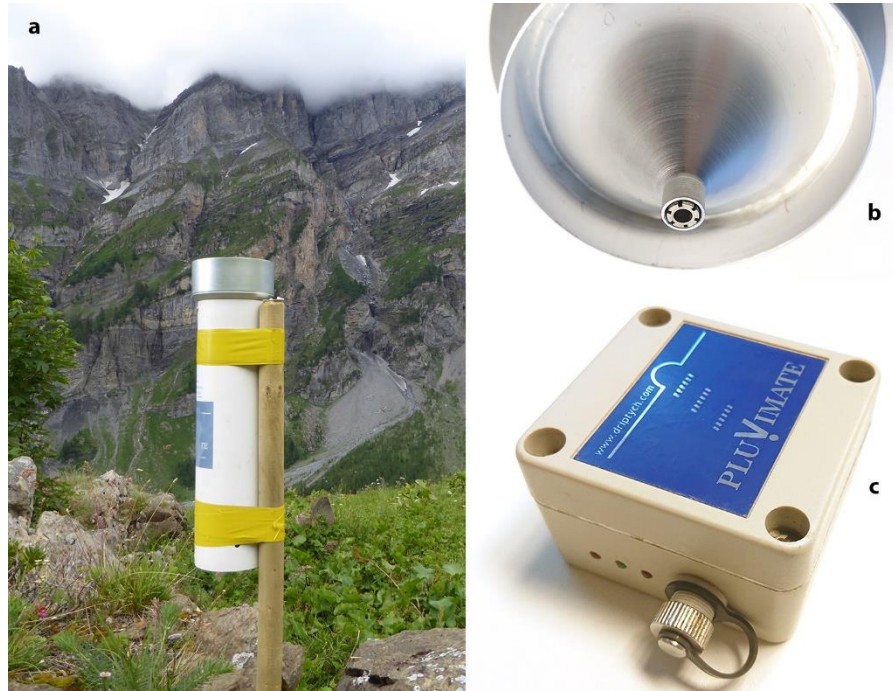

Figure 2.Drop-counting rain gauge used for rainfall measures. The Pluvimate is set-up vertically between 0.8 and 1.2 meters above the ground level (a). A tip at the end of the funnel (b) creates a calibrated drop of water that falls on the sensor, (c) which counts and records the number of drops during a given amount of time.

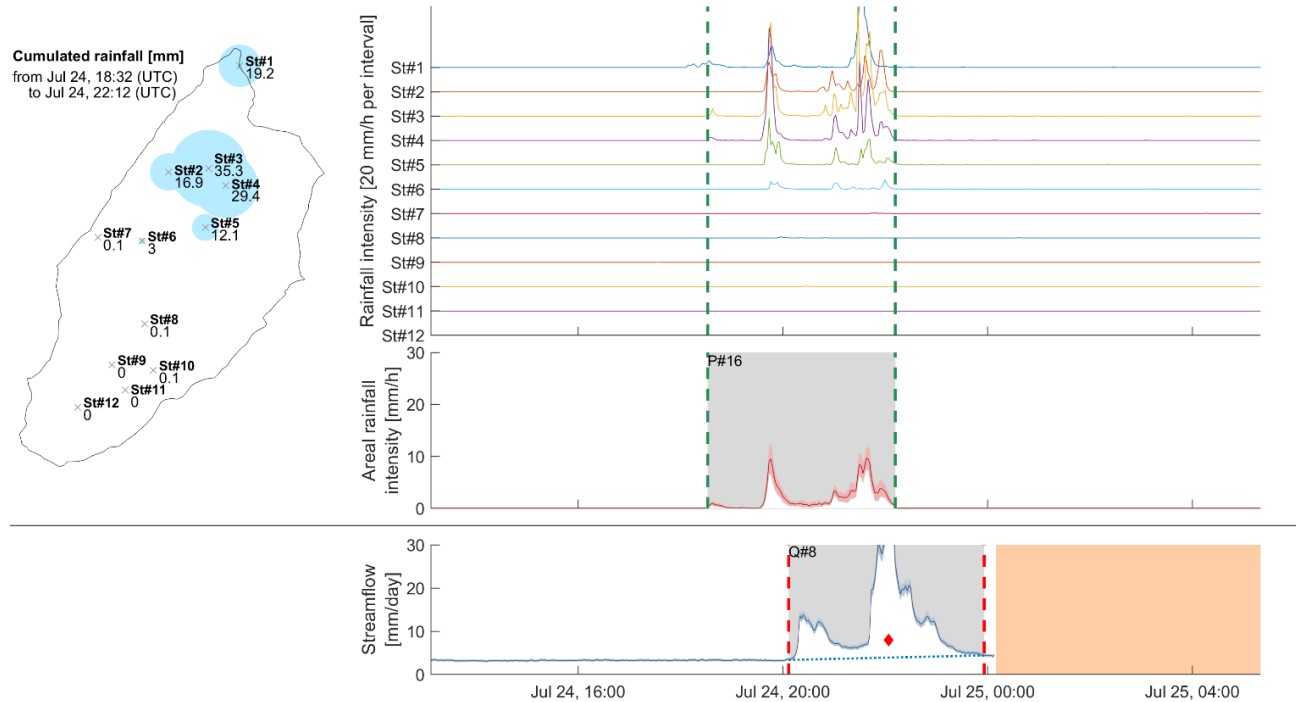

Figure 3. Summary of the recorded rainfall and streamflow for the rainfall event of July 24[th] 2018 at 6:32 PM (UTC).


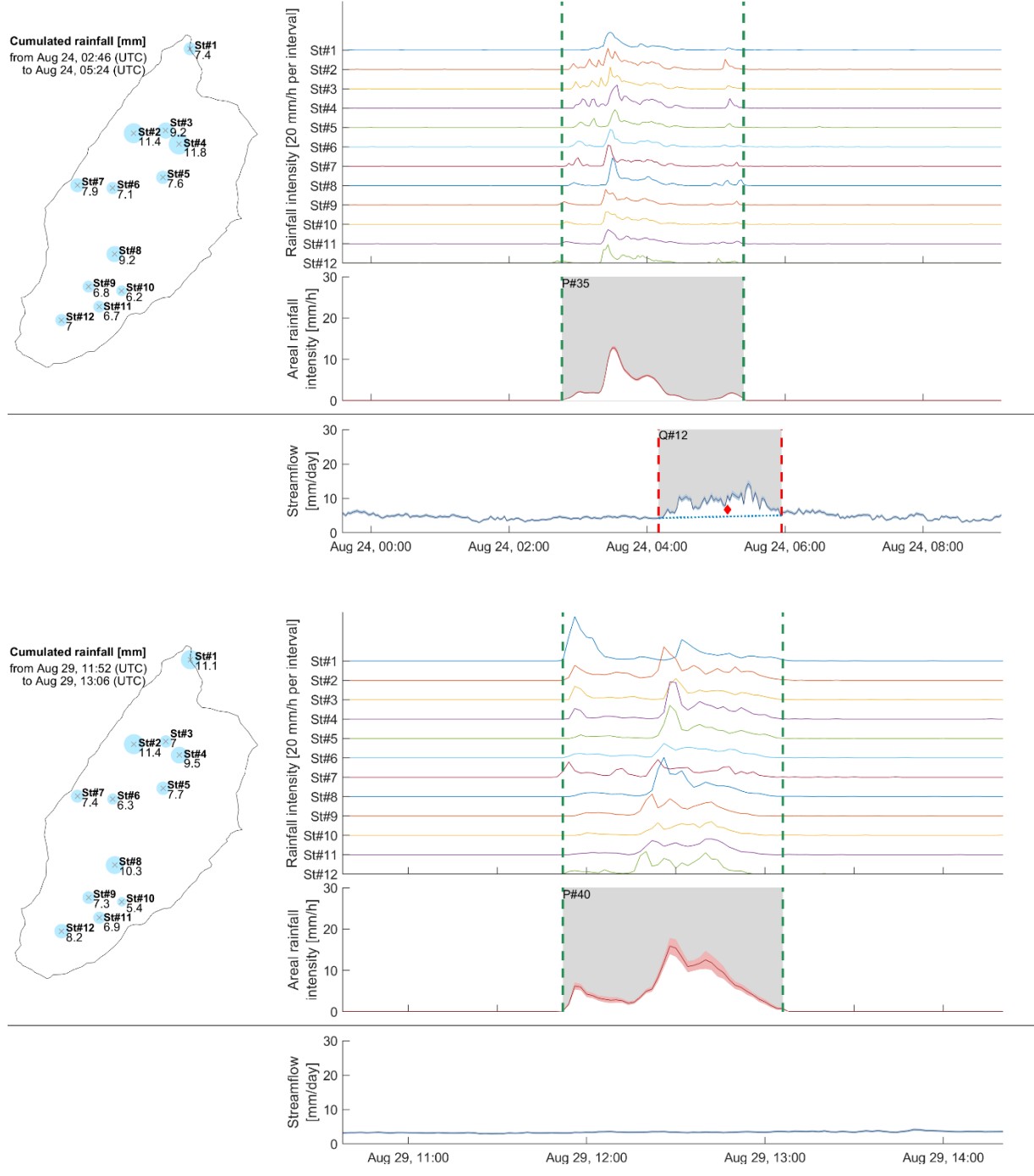

Figure 4. Summary of the recorded rainfall and streamflow for the rainfall events of August 24[th] 2018 at 2:46 AM (top) and August 29[th] 2018 at 11:52 AM (bottom).

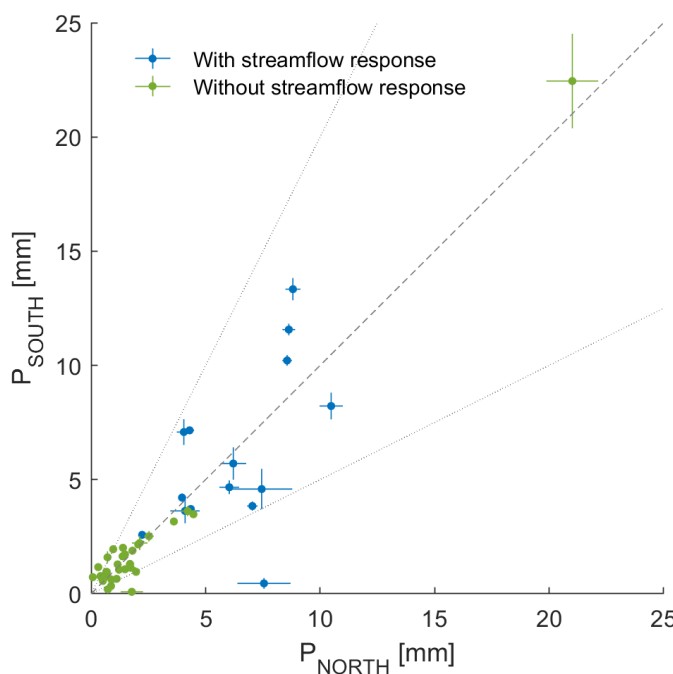

Figure 5. Scatterplot of the rainfall amounts over the northern and the southern parts of the catchment for all 48 rainfall events. The dotted lines show the 1/2 and 2/1 lines which correspond to twice more rainfall in one part of the catchment than in the other or to $|I_{ASYM}| > 0.33$. The highest event is an outlier (event of 6-Aug with 43.5 mm of rainfall in total): is flagged without streamflow response because the river stage measure was disturbed during this period.


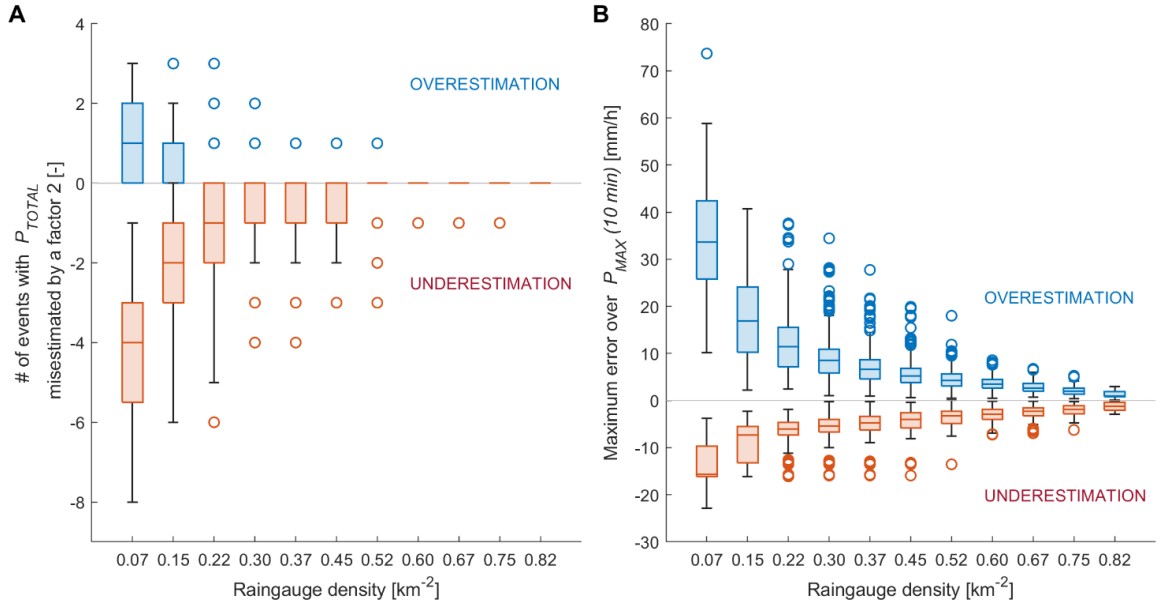


Figure 6. A) Number of rainfall events for which the total amount of rainfall is overestimated or underestimated by a factor 2, according to the rain gauge density, going from 0.07 to 0.82 rain gauges per km² (respectively 1 to 11 rain gauges within the catchment). B) Error on the maximum rainfall over 10 minutes $P_{MAX}$(10 min) according to the raingauge density. For each rain gauge density, all possible combinations of rain gauge networks are tested.

The reference value is estimated from the full 12-rain gauge network. The bottom and top of each boxes are the 25th and 75th percentiles of the sample, the middle line the sample median. The whiskers go up to 1.5 times the interquartile range; values beyond the whiskers (outliers) are marked with circles.

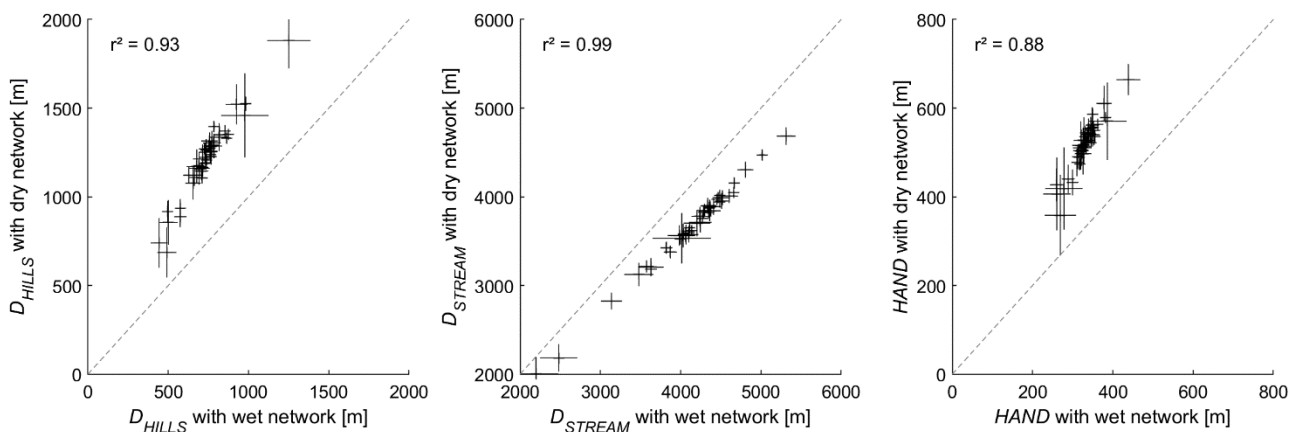

Figure 7. Scatterplots of the distance metrics for the dry network state versus the wet state, for all 48 rainfall events. The bars indicate the standard deviation obtained from the 20 rainfall field realisations. r² indicates the linear correlation.

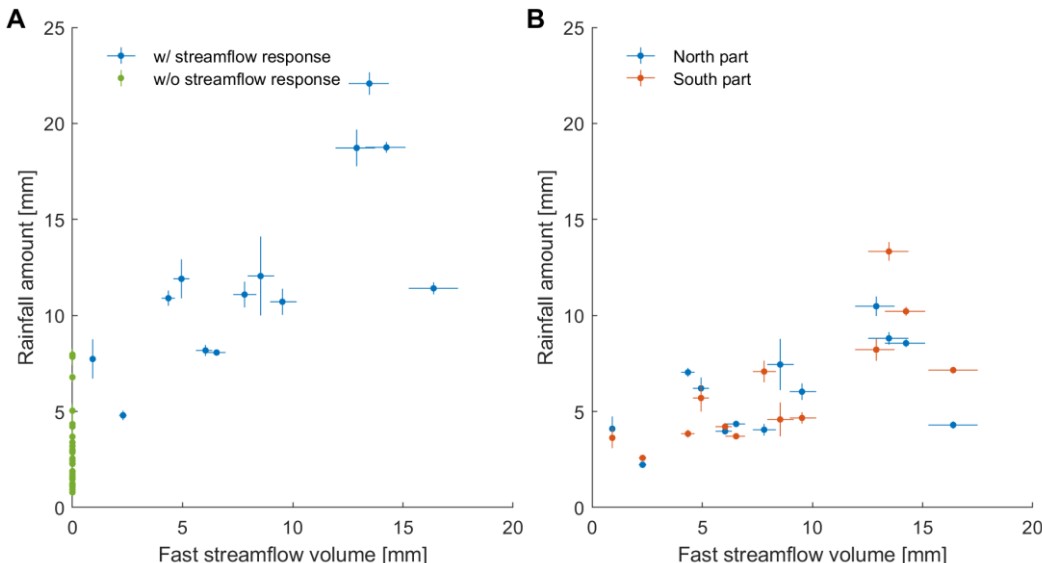

Figure 8. Scatterplots of A) total rainfall amounts versus fast streamflow (highlighting the threshold for streamflow response) and B) of rainfall amounts in the northern and the southern part against fast streamflow (for separation line, see Figure 1). The bars show the standard deviation of estimated rainfall (Section 3.2) and of streamflow (Section 2). The events of 24 July ($P_{ALL}$=8.0 mm, $Q_{FAST}$ = 30.4 mm) and of 6 Aug ($P_{ALL}$=43.5 mm, $Q$ not recorded) are out of the axes in A and in B.

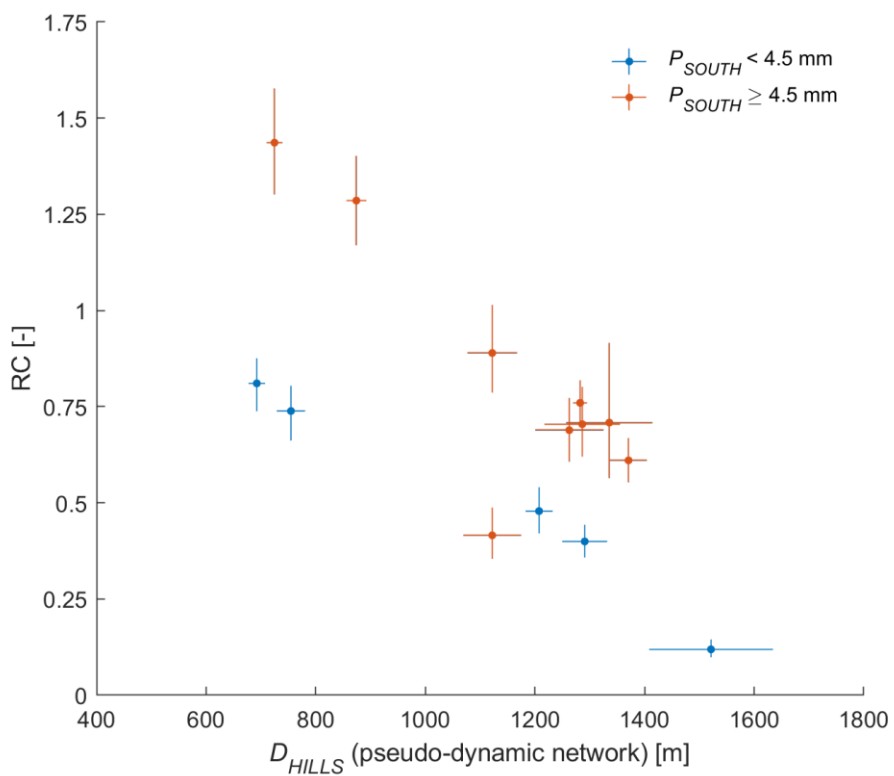

Figure 9. Runoff coefficient against $D_{HILLS}$, highlighting events with high rainfall amounts in the southern part, i.e. events with $P_{SOUTH}>4.5$ mm; the 24 July event with $3.02 <RC<4.85$ and $D_{HILLS}=740\pm140$ m has been discarded (see Section 4.1.1).

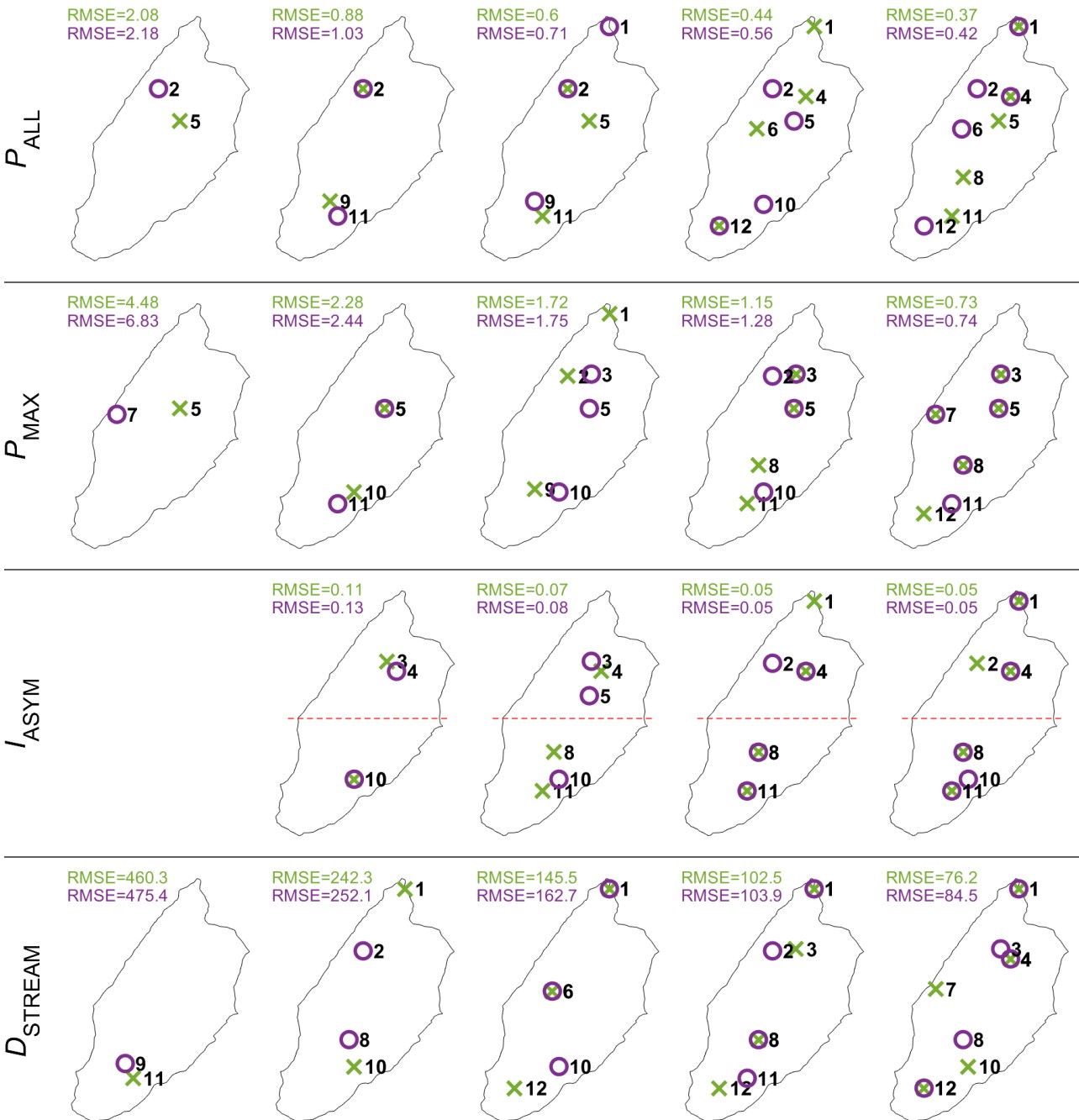

Figure 10. Best (green) and second best (purple) networks and associated RMSE values for 1 to 5 stations resulting from the minimization of the RMSE over 23 events for the $P_{ALL}$, $P_{MAX}$, $I_{ASYM}$ and $D_{STREAM}$. The red dashed line splits the catchments into two parts of equal area.

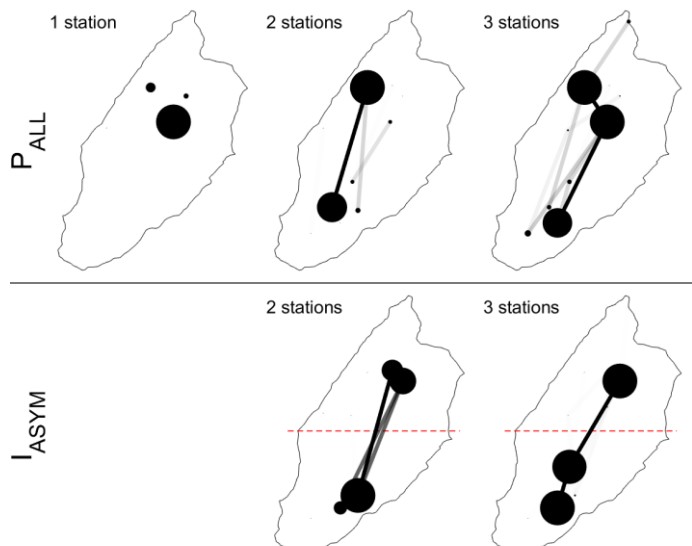

Figure 11. Sensitivity test over the best network from 1 to 3 stations, evaluated by removing from 1 to 3 events over the 23 events (2047 combinations) for the $P_{ALL}$ and $I_{ASYM}$. The result is presented graphically: larger dots and wider links represent configurations that are found more frequently than others over the different simulations. The red dashed line splits the catchments into two parts of equal area.

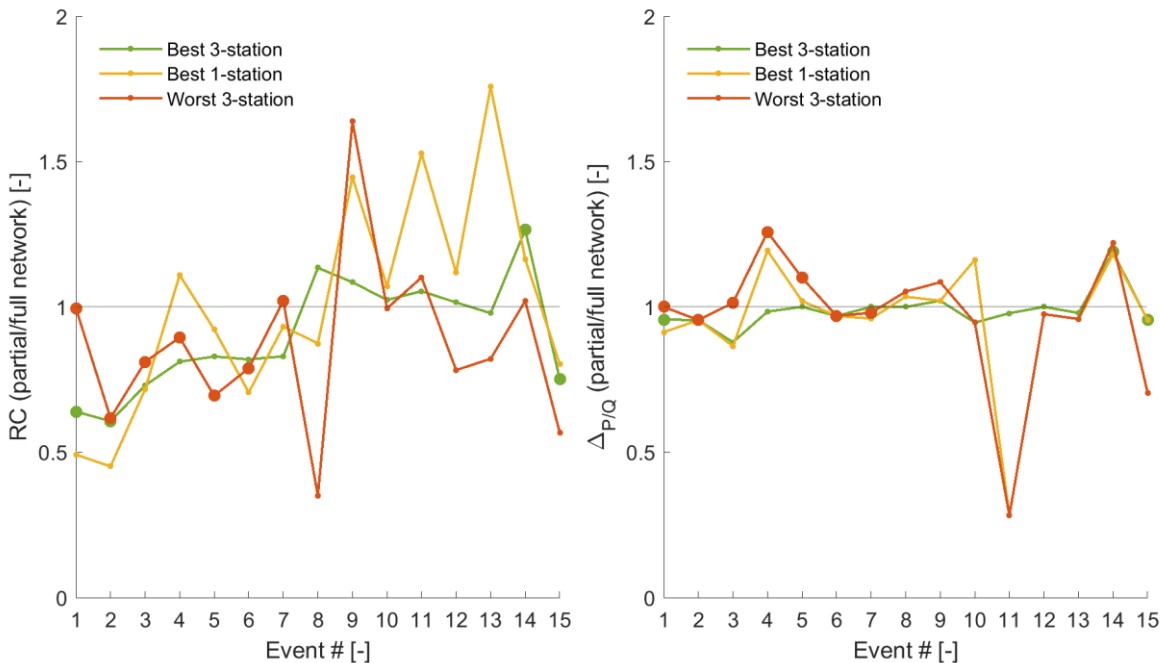

Figure 12. Comparison of streamflow response metrics ratios between a partial network (best 3-station, best 1-station and worst 3-station networks) and the full rain gauge network, using the RC (left) and lag time $\Delta$P/Q (right). The dataset is subset#4 of Table 2. Larger dots highlight events where events where only 2 of 3 stations were operational (see Section 4.1.1). The lines connect the events to improve readability.

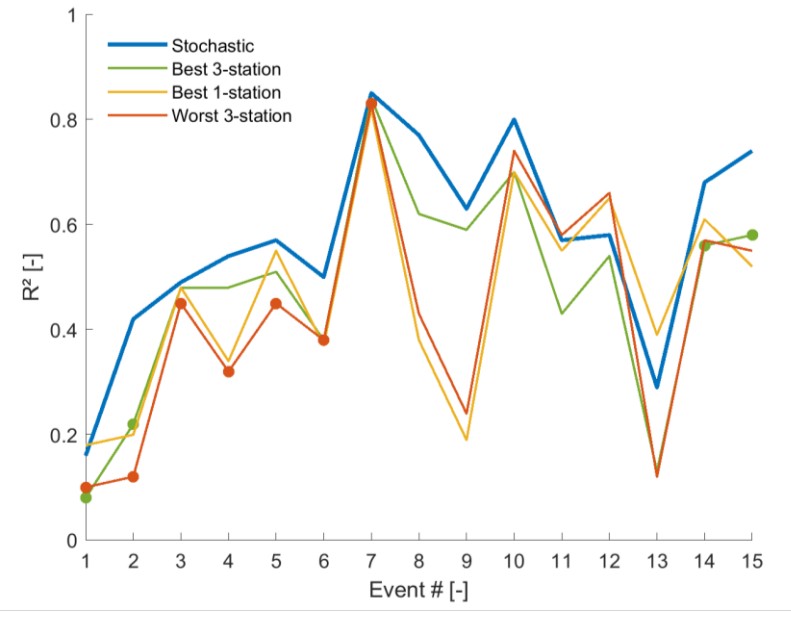


Figure 13. Analysis of 15 rainfall-runoff model events (subset #3, Table 2) with the correlation coefficient between simulated and observed streamflow for different rainfall fields inputs: the stochastic generation of rainfall fields based on all available rain gauge stations, the best 3-stations and the best 1-station network, and the worst 3-stations network. Larger dots highlight events where events where only 2 of 3 stations were operational (see Section 4.1.1).

The lines connect the events to improve readability.

Table 1. List of used metrics, with corresponding parameter name or abbreviation.

| Description | Notation, Unit |
| --- | --- |
| Rainfall interpolated over entire catchment | $P_{ALL}$, mm |
| Rainfall interpolated over north half of catchment | $P_{NORTH}$, mm |
| Rainfall interpolated over south half of catchment | $P_{SOUTH}$, mm |
| Rainfall event duration | $P_{DURATION}$, min |
| Maximum rainfall intensity over the event, $i$ = {ALL, NORTH, SOUTH} | $P_{max\ i}$, mm |
| Index of spatial asymmetry of rainfall | $I_{ASYM}$, - |
| Distance of rainfall spatial center of mass to stream network (along hillslopes) | $D_{HILLS}$, m |
| Distance of rainfall spatial center of mass to outlet along the stream network | $D_{STREAM}$, m |
| Mean height above the nearest drainage | $H_{HAND}$, m |
| Cumulated amount of rainfall for the last X days | $W_{X\ days}$, mm |
| Streamflow at the start of the streamflow event | $Q_{INIT}$, mm |
| Fast streamflow amount | $Q_{FAST}$, mm |
| Streamflow response event duration | $Q_{DURATION}$, min |
| Rainfall runoff coefficient | RC, - |
| Lag time between the first third of cumulated rainfall and the first third of cumulated fast streamflow | $\Delta_{P/Q}$, min |


Table 2. Summary of the different subsets of rainfall events used within this study. The streamflow response outlier event discarded in subset #4 corresponds to July 24th 2018.

| Subset | $P_{event}$ No. (1–48) | # of events |
|---|---|---|
| **#1**: $P_{event}$ recorded by at least 7 stations | | 48 |
| **#2**: $P_{event}$ recorded by 12 from 12 stations | | 23 |
| **#3**: $P_{event}$ w/ streamflow response (no streamflow observation) | | 15 |
| **#4**: $P_{event}$ w/ streamflow response (no outlier) | | 14 |

Table 3. List of recorded precipitation events with streamflow response (event series #3 of Table 2). Full details are available in the Supplementary Material.

| Date | $P_{DURATION}$ [min] | $Q_{DURATION}$ [min] | $\Delta_{P/Q}$ [min] | $P_{ALL}$ [mm] | $P_{NORTH}$ [mm] | $P_{SOUTH}$ [mm] | $P_{NORTH}/P_{ALL}$ [-] | $P_{SOUTH}/P_{ALL}$ [-] | $I_{ASYM}$ [-] | $W_{3\,days}$ [mm] | Stream network | $Q_{INIT}$ [mm] | $Q_{FAST}$ [mm] | RC [-] | $D_{HILLS}$ [m] | $D_{STREAM}$ [m] | $H_{HAND}$ [m] |
|---|---|---|---|---|---|---|---|---|---|---|---|---|---|---|---|---|---|
| 2-Jul | 42 | 44 | 24 | 7.7 | 4.1 | 3.6 | 0.53 | 0.47 | -0.06 | 3.2 | dry | 7.9 | 0.9 | 0.12 | 1521 | 4008 | 611 |
| 3-Jul | 40 | 135 | 23 | 12.1 | 7.4 | 4.6 | 0.62 | 0.38 | -0.24 | 12.7 | dry | 7.5 | 8.5 | 0.71 | 1336 | 3842 | 550 |
| 5-Jul | 224 | 309 | 71 | 8.2 | 4.0 | 4.2 | 0.49 | 0.51 | 0.03 | 29.8 | wet | 6.0 | 6.0 | 0.74 | 755 | 4374 | 350 |
| 6-Jul | 478 | 587 | 65 | 20.2 | 8.6 | 11.6 | 0.43 | 0.57 | 0.15 | 40.3 | wet | 5.8 | 25.9 | 1.29 | 874 | 4450 | 355 |
| 14-Jul | 358 | 302 | 49 | 18.7 | 10.5 | 8.2 | 0.56 | 0.44 | -0.12 | 0.0 | dry | 4.5 | 12.9 | 0.69 | 1263 | 3574 | 554 |
| 15-Jul | 136 | 281 | 33 | 10.7 | 6.0 | 4.7 | 0.56 | 0.44 | -0.13 | 18.9 | dry | 5.5 | 9.5 | 0.89 | 1122 | 3377 | 528 |
| 20-Jul | 288 | 228 | 49 | 18.8 | 8.6 | 10.2 | 0.46 | 0.54 | 0.09 | 3.4 | dry | 4.8 | 14.2 | 0.76 | 1282 | 3823 | 541 |
| 24-Jul | 220 | 229 | 45 | 8.0 | 7.5 | 0.5 | 0.94 | 0.06 | 0.02 | 12.2 | dry | 3.1 | 30.4 | 3.78 | 740 | 2184 | 419 |
| 14-Aug | 204 | 152 | 47 | 11.1 | 4 | 7.1 | 0.37 | 0.64 | 0.27 | 10.2 | dry | 4.0 | 7.8 | 0.70 | 1286 | 4305 | 540 |
| 17-Aug | 152 | 109 | 38 | 11.9 | 6.2 | 5.7 | 0.52 | 0.48 | -0.04 | 17.5 | dry | 3.2 | 4.9 | 0.42 | 1122 | 3780 | 490 |
| 23-Aug | 388 | 237 | 47 | 22.1 | 8.8 | 13.3 | 0.40 | 0.60 | 0.20 | 5.4 | dry | 2.4 | 13.5 | 0.61 | 1371 | 3756 | 563 |
| 24-Aug | 158 | 107 | 40 | 8.1 | 4.4 | 3.7 | 0.54 | 0.46 | -0.08 | 29.5 | wet | 4.1 | 6.5 | 0.81 | 692 | 4114 | 320 |
| 29-Aug | 72 | 116 | 48 | 4.8 | 2.2 | 2.6 | 0.46 | 0.54 | 0.07 | 12.4 | dry | 3.0 | 2.3 | 0.48 | 1207 | 3526 | 524 |
| 01-sept | 628 | 341 | 101 | 11.4 | 4.3 | 7.2 | 0.38 | 0.63 | 0.25 | 20.4 | wet | 3.4 | 16.4 | 1.44 | 725 | 4487 | 331 |
| 13-sept | 370 | 59 | 45 | 10.9 | 7.0 | 3.8 | 0.65 | 0.35 | -0.29 | 0.0 | dry | 2.6 | 4.4 | 0.40 | 1291 | 3594 | 556 |

Table 4. Correlations between rainfall metrics and hydrologic response metrics for event series #4 of Table 2. Absolute values equal or higher than 0.60 are in bold.

| | $P_{ALL}$ [mm] | $P_{NORTH}$ [mm] | $P_{SOUTH}$ [mm] | $P_{max\ ALL}$ [mm.h$^{-1}$] | $P_{max\ NORTH}$ [mm.h$^{-1}$] | $P_{max\ SOUTH}$ [mm.h$^{-1}$] | $I_{ASYM}$ [-] | $W_{3\ days}$ [mm] | $Q_{INIT}$ [mm] | $Q_{FAST}$ [mm] | $P_{DURATION}$ [min] | $Q_{DURATION}$ [min] | $\Delta_{P/Q}$ [min] |
|---|---|---|---|---|---|---|---|---|---|---|---|---|---|
| $P_{ALL}$ [mm] | - | | | | | | | | | | | | |
| $P_{NORTH}$ [mm] | **0.89** | - | | | | | | | | | | | |
| $P_{SOUTH}$ [mm] | **0.94** | **0.69** | - | | | | | | | | | | |
| $P_{max\ ALL}$ [mm.h$^{-1}$] | 0.01 | 0.19 | -0.12 | - | | | | | | | | | |
| $P_{max\ NORTH}$ [mm.h$^{-1}$] | 0.09 | 0.33 | -0.11 | **0.96** | - | | | | | | | | |
| $P_{max\ SOUTH}$ [mm.h$^{-1}$] | 0.19 | 0.19 | 0.16 | **0.87** | **0.78** | - | | | | | | | |
| $I_{ASYM}$ [-] | 0.25 | -0.20 | 0.55 | -0.42 | -0.56 | -0.06 | - | | | | | | |
| $W_{3\ days}$ [mm] | -0.19 | -0.30 | -0.09 | -0.22 | -0.27 | -0.23 | 0.18 | - | | | | | |
| $Q_{INIT}$ [mm] | -0.13 | 0.00 | -0.21 | 0.52 | 0.54 | 0.27 | -0.28 | 0.26 | - | | | | |
| $Q_{FAST}$ [mm] | **0.77** | 0.58 | **0.80** | -0.17 | -0.16 | -0.08 | 0.43 | 0.33 | -0.01 | - | | | |
| $P_{DURATION}$ [min] | 0.56 | 0.38 | **0.62** | -0.59 | -0.52 | -0.48 | 0.44 | 0.14 | -0.43 | **0.74** | - | | |
| $Q_{DURATION}$ [min] | 0.56 | 0.39 | **0.61** | -0.27 | -0.27 | -0.17 | 0.42 | 0.52 | 0.11 | **0.89** | **0.64** | - | |
| $\Delta_{P/Q}$ [min] | 0.13 | -0.11 | 0.29 | **-0.71** | **-0.71** | -0.58 | 0.59 | 0.41 | -0.33 | 0.52 | **0.81** | **0.60** | - |
| RC [-] | 0.31 | 0.13 | 0.40 | -0.25 | -0.29 | -0.22 | 0.44 | **0.65** | -0.05 | **0.81** | **0.67** | **0.80** | **0.72** |

Table 5. Correlations between distance metrics for rainfall events with streamflow response (series #4 of Table 2). Absolute values equal or higher than 0.60 are in bold. Correlations for all rainfall events are available in the Supplementary Material.

| | River network | $D_{HILLS}$ Wet | $D_{HILLS}$ Dry | $D_{STREAM}$ Wet | $D_{STREAM}$ Dry | $H_{HAND}$ Wet | $H_{HAND}$ Dry | $D_{HILLS}$ Pseudo-dynamic | $D_{STREAM}$ Pseudo-dynamic | $H_{HAND}$ Pseudo-dynamic |
|---|---|---|---|---|---|---|---|---|---|---|
| $D_{HILLS}$ | *Wet* | - | | | | | | | | |
| $D_{HILLS}$ | *Dry* | **0.96** | - | | | | | | | |
| $D_{STREAM}$ | *Wet* | 0.59 | **0.61** | - | | | | | | |
| $D_{STREAM}$ | *Dry* | 0.54 | 0.53 | **0.99** | - | | | | | |
| $H_{HAND}$ | *Wet* | **0.91** | **0.93** | 0.51 | 0.44 | - | | | | |
| $H_{HAND}$ | *Dry* | **0.75** | **0.89** | 0.40 | 0.28 | **0.90** | - | | | |
| $D_{HILLS}$ | *Pseudo-dynamic* | 0.42 | 0.45 | 0.08 | 0.04 | 0.51 | 0.49 | - | | |
| $D_{STREAM}$ | *Pseudo-dynamic* | 0.32 | 0.31 | **0.75** | **0.77** | 0.18 | 0.09 | -0.57 | - | |
| $H_{HAND}$ | *Pseudo-dynamic* | 0.26 | 0.30 | -0.05 | -0.10 | 0.40 | 0.42 | **0.98** | **-0.68** | - |
| *RC* | | -0.20 | -0.21 | 0.10 | 0.13 | -0.28 | -0.28 | **-0.70** | 0.53 | **-0.70** |
| $\Delta_{P/Q}$ | | -0.10 | -0.05 | 0.21 | 0.21 | -0.13 | -0.06 | **-0.66** | **0.60** | **-0.68** |


Table 6. List of the tested predictors for the RC with a pure quadratic regression, and corresponding statistics: root mean square error (RMSE), coefficient of determination (R²), variance of residuals (var. residuals), p-value, corrected Akaike criterion (AICc) and AICc ranking. The acceptable p-values (≤0.05) and first 3 ranks are highlighted. The analysis is over the 14 events of series #4 of Table 2.

| Predictor 1 | Predictor 2 | RMSE | R² | var. residuals | p-value | AICc | rank AICc |
|---|---|---|---|---|---|---|---|
| $P_{ALL}$ | - | 0.34 | 0.14 | 0.10 | 0.44 | -24.96 | 17 |
| $P_{NORTH}$ | - | 0.36 | 0.02 | 0.11 | 0.88 | -23.20 | 18 |
| $P_{SOUTH}$ | - | 0.31 | 0.28 | 0.08 | 0.17 | -27.44 | 12 |
| $I_{ASYM}$ | - | 0.33 | 0.22 | 0.09 | 0.25 | -26.37 | 16 |
| $W_{3\,days}$ | - | 0.27 | 0.48 | 0.06 | **0.03** | -31.90 | 7 |
| $D_{HILLS}$ (pseudo-dynamic) | - | 0.26 | 0.52 | 0.06 | **0.02** | -33.00 | **3** |
| $D_{STREAM}$ (pseudo-dynamic) | - | 0.23 | 0.61 | 0.04 | **0.01** | -36.13 | **1** |
| $P_{ALL}$ | $I_{ASYM}$ | 0.33 | 0.35 | 0.07 | 0.36 | -19.88 | 19 |
| $P_{NORTH}$ | $I_{ASYM}$ | 0.34 | 0.29 | 0.08 | 0.50 | -18.53 | 21 |
| $P_{SOUTH}$ | $I_{ASYM}$ | 0.33 | 0.35 | 0.07 | 0.37 | -19.84 | 20 |
| $W_{3\,days}$ | $I_{ASYM}$ | 0.25 | 0.62 | 0.04 | **0.05** | -27.38 | 13 |
| $D_{HILLS}$ (pseudo-dynamic) | $I_{ASYM}$ | 0.23 | 0.68 | 0.04 | **0.03** | -29.55 | 9 |
| $D_{STREAM}$ (pseudo-dynamic) | $I_{ASYM}$ | 0.25 | 0.62 | 0.04 | **0.05** | -27.30 | 14 |
| $P_{ALL}$ | $D_{HILLS}$ (pseudo-dynamic) | 0.22 | 0.70 | 0.03 | **0.02** | -30.65 | 8 |
| $P_{NORTH}$ | $D_{HILLS}$ (pseudo-dynamic) | 0.26 | 0.60 | 0.05 | 0.06 | -26.76 | 15 |
| $P_{SOUTH}$ | $D_{HILLS}$ (pseudo-dynamic) | 0.21 | 0.74 | 0.03 | **0.01** | -32.80 | 4 |
| $W_{3\,days}$ | $D_{HILLS}$ (pseudo-dynamic) | 0.24 | 0.65 | 0.04 | **0.04** | -28.34 | 11 |
| $P_{ALL}$ | $D_{STREAM}$ (pseudo-dynamic) | 0.20 | 0.75 | 0.03 | **0.01** | -33.18 | **2** |
| $P_{NORTH}$ | $D_{STREAM}$ (pseudo-dynamic) | 0.21 | 0.74 | 0.03 | **0.01** | -32.55 | 5 |
| $P_{SOUTH}$ | $D_{STREAM}$ (pseudo-dynamic) | 0.21 | 0.74 | 0.03 | **0.01** | -32.46 | 6 |
| $W_{3\,days}$ | $D_{STREAM}$ (pseudo-dynamic) | 0.24 | 0.67 | 0.04 | **0.03** | -29.10 | 10 |


Table 7 As Table 6 but for the lag $\Delta_{P/Q}$.

| Predictor 1 | Predictor 2 | RMSE | R² | var. residuals | p-value | AICc | rank AICc |
|---|---|---|---|---|---|---|---|
| $P_{\text{max ALL}}$ | - | 13.07 | 0.64 | 144.52 | **0.00** | 76.99 | **3** |
| $P_{\text{max NORTH}}$ | - | 12.70 | 0.66 | 136.56 | **0.00** | 76.20 | **2** |
| $P_{\text{max SOUTH}}$ | - | 16.52 | 0.43 | 231.05 | **0.05** | 83.56 | 11 |
| $I_{\text{ASYM}}$ | - | 17.25 | 0.37 | 251.75 | 0.08 | 84.76 | 13 |
| $W_{\text{3 days}}$ | - | 19.83 | 0.17 | 332.65 | 0.35 | 88.66 | 19 |
| $D_{\text{HILLS}}$ (pseudo-dynamic) | - | 16.28 | 0.44 | 224.27 | **0.04** | 83.14 | 10 |
| $D_{\text{STREAM}}$ (pseudo-dynamic) | - | 13.39 | 0.62 | 151.71 | **0.00** | 77.67 | 4 |
| $P_{\text{max ALL}}$ | $I_{\text{ASYM}}$ | 11.10 | 0.79 | 85.35 | **0.00** | 78.72 | 5 |
| $P_{\text{max NORTH}}$ | $I_{\text{ASYM}}$ | 12.89 | 0.71 | 115.01 | **0.02** | 82.89 | 8 |
| $P_{\text{max SOUTH}}$ | $I_{\text{ASYM}}$ | 10.06 | 0.83 | 70.06 | **0.00** | 75.95 | **1** |
| $W_{\text{3 days}}$ | $I_{\text{ASYM}}$ | 15.86 | 0.57 | 174.17 | 0.08 | 88.70 | 20 |
| $D_{\text{HILLS}}$ (pseudo-dynamic) | $I_{\text{ASYM}}$ | 12.97 | 0.71 | 116.52 | **0.02** | 83.07 | 9 |
| $D_{\text{STREAM}}$ (pseudo-dynamic) | $I_{\text{ASYM}}$ | 13.83 | 0.67 | 132.39 | **0.03** | 84.86 | 15 |
| $P_{\text{max ALL}}$ | $D_{\text{HILLS}}$ (pseudo-dynamic) | 14.18 | 0.65 | 139.25 | **0.03** | 85.57 | 17 |
| $P_{\text{max NORTH}}$ | $D_{\text{HILLS}}$ (pseudo-dynamic) | 13.95 | 0.67 | 134.65 | **0.03** | 85.10 | 16 |
| $P_{\text{max SOUTH}}$ | $D_{\text{HILLS}}$ (pseudo-dynamic) | 16.57 | 0.53 | 190.15 | 0.12 | 89.93 | 21 |
| $W_{\text{3 days}}$ | $D_{\text{HILLS}}$ (pseudo-dynamic) | 15.65 | 0.58 | 169.50 | 0.07 | 88.32 | 18 |
| $P_{\text{max ALL}}$ | $D_{\text{STREAM}}$ (pseudo-dynamic) | 11.40 | 0.78 | 89.99 | **0.01** | 79.46 | 6 |
| $P_{\text{max NORTH}}$ | $D_{\text{STREAM}}$ (pseudo-dynamic) | 11.55 | 0.77 | 92.36 | **0.01** | 79.82 | 7 |
| $P_{\text{max SOUTH}}$ | $D_{\text{STREAM}}$ (pseudo-dynamic) | 13.37 | 0.69 | 123.70 | **0.02** | 83.91 | 12 |
| $W_{\text{3 days}}$ | $D_{\text{STREAM}}$ (pseudo-dynamic) | 13.82 | 0.67 | 132.18 | **0.03** | 84.84 | 14 |