# Peer review of "Benefits from high density rain gauge observations for hydrological response analysis in a small alpine catchment"

_Hydrology and Earth System Sciences, 2020_

## Author Comment (AC1) · 7 Aug 2020

This manuscript is the resubmission of the manuscript https://doi.org/10.5194/hess-2019-683 which has been entirely revised following the propositions of improvement of the reviewers. The modifications and additions can be summarized as follows:

- We introduce a confidence interval on the rating curve to account for errors on the salt gauging measures (Section 2). This error is reflected in the computations and related figures.

- The interpolation of the rainfall fields changed from the Thiessen method to a

stochastic interpolation method (section 3.2.1). The standard deviation on the rainfall estimates is shown on the related figures and propagates through the computations involving rainfall measures.

- We use the rainfall maximum intensity over 10 minutes as a proxy of saturation-excess vs. capacity-excess infiltration (section 4.2.2).

- We tested the height above the nearest drainage (HAND) metric (Section 3.2.2) as an additional geomorphological distance measure.

- To account for the dynamic of the stream network expansion state (Section 4.2.3), the stream network varies between its maximum extent (wet conditions) and its minimum extent (dry conditions). The related geomorphological distance metrics (e.g. DHILLS DSTREAM, DHAND) are computed for both network extents. A composite network is also introduced, mixing both: the network extent is chosen (at the event scale) based on the initial catchment wetness conditions, based on antecedent precipitation.

- Most of the figures and tables in the main text and in the supplementary material have been updated to account for the above changes.

- The data and MatLab code associated has been updated on Zenodo to add the rainfall and streamflow uncertainty estimates https://doi.org/10.5281/zenodo.3946242.

Finally, we would like to point out here that during the first submission to HESS, the discussion arose whether a hydrological model could shed more light on how hetero-geneous rainfall events impact the streamflow generation dynamics (link to answer to reviewer 2: https://hess.copernicus.org/preprints/hess-2019-683/hess-2019-683-AC2-supplement.pdf). While we considered the use of a simple model in a virtual experi-ment (with generated rainfall and streamflow values), we decided that adding such a
modelling part would go beyond the key objective of this paper, which is a data-based analysis of the rainfall-streamflow dynamics of this catchment and the role of distributed rainfall observations. The analysis framework and results presented in the manuscript submitted here underline that for this (and possibly similarly small catchments), knowing the localization of rainfall is of key importance to understand the rainfall-runoff response. The results shed new light on the value of established geomorphological distance measures and their role for understanding the streamflow response. Furthermore, our work highlights the importance of observing the extension and retraction of the river network, which is rarely reported in similar studies.

---

## Referee Comment (RC1) · Anonymous Referee #1 · 20 Aug 2020

This was the first time I was involved as a reviewer for this manuscript. The aim is to identify the best rain gauge network setup for runoff predictions. The topic is suitable for the journal and of interest for the community; the manuscript is well-written. However, I do not recommend a publication at its current stage. There are a few major comments listed below, which have to be addressed before the manuscript can be recommended for publication. More specific comments and some technical corrections follow afterwards. My overall recommendation of the manuscript is major revision.

Major comments:

1. The title states "value of high density rain gauge observations for . . . hydrology". I'm

struggling with this holistic formulation. Indeed, the value is "only" (please don't get me wrong here) based on prediction of RC and deltaP/Q. While a realistic estimate of these characteristics is valuable, the uncertainties resulting from the final network with 3 rain gauges for these two criteria is not shown and should be added in a later version of the manuscript. In general, I'm missing the runoff peak as important characteristic in the manuscript. Maybe the authors can involve it/comment on it why it was not considered. Also, although the analysis is designed mainly for discharge estimation, results should be also interpreted in terms of rainfall (e.g. resulting areal rainfall (extremes) for different rain gauge network densities, spatial rainfall characteristics,…).

2. Based on the comment before, the impact of the rain gauge network densities (and rain gauge locations) on the runoff is not analysed. In the additionally uploaded comment the main author states a rainfall-runoff modelling would go beyond the scope of the study. I do not agree with that and recommend this modelling approach to analyse the impact on the resulting runoff itself instead on single runoff statistics. To attribute the spatial rainfall variability, a distributed rainfall-runoff model would be the best solution.

3. Also, I was wondering why is there not a consistent number of events analysed throughout the manuscript. I understand that there are always measuring issues and maybe some observations are questionable, but then please remove them at the beginning. There could be one number of rainfall events considered and one subset of them for discharge analysis, but at the current state results from different subsections cannot be compared with each other due to the different populations of considered events.

Specific comments:

L25-27 It should be mentioned here again that this issue is related to mountainous areas and is not a problem in general.

Fig. 2 I don't see the additional worth of showing Fig. 2 and recommend to leave it out,

especially since it is included in the supplement as Fig S2 as well.

L90 "average elevation" Please change to mean or median, depending on how you determined the "average" value.

L117-118 The construction of the rating curve is not interesting for the manuscript and can be left out, also the elements regarding its construction in the supplement.

L154-155 The term interpolation is not suitable in my opinion due to the rainfall generation mechanisms behind. I suggest "areal rainfall is generated after Bernoit et al. (2018a) by constraining actual observations at rain gauge locations". The authors should give a less brief explanation, since in the cited manuscript different versions are applied for rainfall generation (three versions due to different covariance models) and it remains unclear for the reader, which model is used for the current study. Why did the authors choose this rainfall generation instead of a regionalization approach as kriging (maybe with altitude as additional information), inverse distance weighting or Thiessen polygons. The latter is chosen later in the manuscript nevertheless due to computational efforts, so why not for the whole study? Was it the authors intention to add an uncertainty analysis

L154-163 The authors should bring this argument in context with the catchment concentration time.

L165-166 The location of the line chosen for the splitting of the catchment seems to be chosen arbitrary. Would a line constructed perpendicular to the main flow direction of the river (or even better, not a straight line but following the lines perpendicular to the isohypses to separate flows exactly) lead to more representable results, since the catchment is then split into a real upper and lower part? Or (thinking the other way around) does it not matter at all and the splitting line could be also drawn from South to North as long as both parts have the same area?

L211-215 I suggest to move this paragraph to the beginning of section 3.3.2

L217-218 The authors declare volume and lag time as "the two key characteristics of streamflow reaction". I do not agree with that. The most important characteristic is peak flow, followed by volume and then lag time and flatting behaviour. Even if all characteristics are considered equal important, the authors should state why peak is not considered in the study. If there were attempts to include peaks which did not work, the authors should state so as "lessons learned" in the manuscript.

L219-221 Is this criterion developed by the authors or should a reference be cited in this context? How was 1/3 chosen as threshold? This value should be catchment-dependent in my opinion, or not? Please clarify.

L222 Why is this criterion "1/3 of the rainfall amount" more robust than "start of the rainfall event", although both starting points are linear correlated?

L275 Same differences lead to higher asymmetry values for smaller values. To avoid a misinterpretation ("Interestingly . . .") Pnorth and Psouth could be normalized by the mean event rainfall amount. This would provide deeper insights, especially since larger differences between both parts cannot be seen in the current approach if they occur for events with high rainfall amounts.

L323-327. I cannot follow the argumentation here. Please explain in detail how you achieve this conclusion and consider at least one or two sentences for each argument.

L330 "to reach a higher RC. . ." Please rephrase, the manuscript is about observations, not modelling.

L341 composites: If there is a differentiation into wet and dry state, how do the authors achieve only one value for each criterion? Are two values estimated (for wet and dry) and then the arithmetic mean is mentioned? Please clarify!

L351-355 It would be nice to have a table with all criteria , where it is stated which one was removed (and why) and which ones were kept. Maybe the information can be added to Table 5 or 6?!

L354 Again, it feels as the number of considered events changes among all subsections.

L380 What is the reason for IASYM preference in the Southern part? Due to the steeper areas? I would have estimated Northern part, since the hydrograph would have already been smoothed when originated in the South. Please try to find physical explanations to your results.

Technical corrections:

General: Please double-check the abbreviation for "meter above sea level"; I have only seen "m a.s.l." and "m asl" so far, but not "m asl."

L155 Benoit et al. 2018 <- a or b? I assume a.

Eq 2, 3, 4 I'm a bit confused what rainfall characteristic is used as input for these equations. Is every raster cell with rainfall used (so I understood it from the text) or only the centre of the rainfall events (as mentioned in Table 1)?

L63 "overlooked" -> ignored

Eq. 2, 3, 4 The term in the numerator should be put in brackets (Eq. 2: "P(..)dHills" -> "(P(..)dHills)")

L195 DHAND is not a distance as indicated by the D, and in the text the variable is introduced with HAND. I suggest to stick to HAND throughout the manuscript to avoid confusions with the other two "real" distances".

L202 Section 3.5 includes no network extent description. Is it missing in the manuscript?

L268 317.8 mm – Is it areal rainfall amount sum or sum over all stations?

L268-269 please provide also the mean values, not only the highest and lowest values, so that the reader get a "feeling" for the rainfall events.

L275 again, please don't use the term average, use mean or median to be more concise. Since Iasym can be positive and negative, the median of its absolute values would be worth to show instead of just the mean, since positive and negative values are levelling out each other.

Fig. 5 and 6 For a logical order the figures should show the rainfall events first, followed by the discharge plot.

L279 "One strongly asymmetric and high intensity event" -> "One strong asymmetric and very intense event"

L283 A volume can't be fast (check also for later occurrences...)

L288 In the sentence before authors mention that the number of events under consideration are reduced by "1", but here again 48 events are studied (also in the following subsections).

L289 The authors should state what wet and dry networks are. I found it later in the caption of Table 1 in S1, but it would lead to clarifications here. Also, the Table 1 in S1 should be shown in the manuscript, since the written part in Section 4.1.2 is more confusing than explaining for me.

Fig9 "events without reaction are not shown" belongs to part b), not a). Please correct the caption. General: Maybe I missed it, but which temporal resolution was used to calculate the correlation (and other criteria)? 2min as this is the resolution of the rain gauge? Or are values aggregated up to e.g. 1h? This has a high impact on the values of the correlation coefficient.

L339 "absence of correlation". Correlation cannot be absent. Better to speak of low correlation or provide absolute values.

L384-386 "is assessed", "is evaluated" – two verbs, please rephrase the sentence.

L402 "what we previously thought"? What was the hypothesis of the authors before?

L421 "three station network" It would be nice to provide the resulting density here as well as "(general) recommendation".

---

## Referee Comment (RC2) · Anonymous Referee #2 · 27 Aug 2020

I already reviewed the initial version of this paper, which aims at highlighting the values of high density rain gauges networks for hydrological purposes in a small catchment of mountainous areas. I still believe that the topic is interesting and relevant for the community. It furthermore has other potential applications in urban areas which are also small and quickly reactive catchments where rainfall variability has strong consequences.

The minor difficulties with regards to the presentation and understanding of the paper have been corrected. Results are now better presented with the new figures. However, the main point was not addressed, i.e. the fact that the authors aims at showing

the importance of grasping the spatio-temporal variability of the rainfall process in the prediction of flows, but the chosen indicators are only event based averages. Furthermore the main rainfall variability (which is at the core of the paper) indicator used is too simplistic since it is basically an asymmetry indicator on the total depth splitting the catchment in two. So I still think that indicators actually accounting for the spatio-temporal variability of the rainfall and hydrologic response should be implemented to actually address the stated topic of the paper. Implementing them requires major modifications of the paper. I guess that this would enable to highlight more precisely the importance of dense networks of rainfall measurement devices.

Specific comments:

- l. 110 – 115 : "The actual extent of the stream network is based on observations during dry and wet periods during Summer 2017 and its exact path was calculated using the Swiss digital elevation model at a resolution of 2 m (swissALTI3D, 2012)." I think that the intrinsic fractal nature of river networks should be mentioned and discussed. The concept of variable network used after also seems interesting.

- l. 150-151 : "Some additional artefacts were recorded, probably generated by strong winds creating resonance. These periods have been manually removed from the data". It should clarified how the data was selected for being removed and what portion was removed.

- 1. 154-157 : It is a great improvement to use this stochastic procedure. Nevertheless, I believe that more details on the interpolation procedure are needed. It should be clarified how the 20 samples are used (computing the error bars in 8-10)?

- Eq. 1 on I_ASYM. As already mentioned, it seems a too simplistic indicator to grasp spatio-temporal variability of the rainfall process. An initial simple suggestion could for instance be to look for the temporal evolution of I_ASYM during an event. But other indicators are needed.

[Figure]

- l. 212-215 : the explanation on why not using streamflow variations (notably peak flow) is not very convincing. If the purpose is to investigate the importance of spatio-temporal variability, I guess studying the temporal variability of the simulated stream-flow is needed.

---

## Author Comment (AC2) · 28 Sep 2020

This was the first time I was involved as a reviewer for this manuscript. The aim is to identify the best rain gauge network setup for runoff predictions. The topic is suitable for the journal and of interest for the community; the manuscript is well-written. How-ever, I do not recommend a publication at its current stage. There are a few major comments listed below, which have to be addressed before the manuscript can be recommended for publication. More specific comments and some technical corrections follow afterwards. My overall recommendation of the manuscript is major revision.

Thank you very much for agreeing to review our paper and for the time spent to formulate all the constructive comments and corrections below.

1a - The title states "value of high density rain gauge observations for… hydrology". I'm struggling with this holistic formulation.

We propose to change the title from "*On the value of high density rain gauge observations for small Alpine headwater catchment hydrology*" to "*Even event-scale hydrological response characterization benefits from high density rain gauge observations*".

1b - Indeed, the value is "only" (please don't get me wrong here) based on prediction of RC and deltaP/Q. While a realistic estimate of these characteristics is valuable, the uncertainties resulting from the final network with 3 rain gauges for these two criteria is not shown and should be added in a later version of the manuscript.

It is true that the uncertainties resulting from the final network of 3-station raingauges is not shown. We propose to add in "4.4 Measurement network analysis" two figures showing i) the RC (Figure 1) and ii) the lag time $\Delta_{P/Q}$ (Figure 2) by comparing the values obtained from the best 1-station or 3-station raingauge network vs. the reference value calculated from the full raingauge network.

As the stochastic method for generating rainfall fields cannot be used with a number of points as low as 1 or 3 stations, we performed the computations using the Thiessen polygons methods and consequently no error bars are associated to these plots. Nevertheless, the Figure 3 compares the two methods (stochastic vs Thiessen polygons) when the RC and the lag time $\Delta_{P/Q}$ are computed from the full raingauge network.

We observe for both the RC (Figure 1) and $\Delta_{P/Q}$ (Figure 2) a lower dispersion of values while increasing the density of the raingauge network. With a 3-station raingauge network the error on the RC (RMSE = 0.186) drops below the error obtained by comparing 2 different interpolation methods (RMSE = 0.256), giving a good confidence to the Thiessen polygons method used for this calculation. In the same way, for $\Delta_{P/Q}$ the error with a 3-station network (RMSE = 8.12) is lower than the error obtain with the model comparison (RMSE = 13.22).

On Figure 2, the dispersion of $\Delta_{P/Q}$ is originally low. Even with a 1-station network the lag can be reproduced correctly for most of the events but can also be completely wrong for one of them. Outliers are still observed with the 3-station raingauge network though, even if the error gets lower (RMSE reduced from 23.18 to 8.12).

[Figure]

*Figure 1. Comparison of RC whether it is calculated from the full raingauge network or from a partial, considering the best 1-station and 3-station network. The dataset is based on the 15 rain events associated to a river reaction.*

[Figure]

*Figure 2. Difference of lag time $\Delta_{P/Q}$ obtained from a partial network (1-station and 3-station network) and the full network.*

[Figure]

*Figure 3. Comparison of the RC (left) and lag time $\Delta_{P/Q}$ (right) calculated using the full raingauge network, but with a different rainfall field interpolation method (Thiessen polygons vs. stochastic).*

1c -In general, I'm missing the runoff peak as important characteristic in the manuscript. Maybe the authors can involve it/comment on it why it was not considered.

The Figure 4 shows the hydrograph of the 15 rainfall events generating a river reaction. The runoff peak identification is straightforward for 5 of them (Q event #1, #2, #6, #14 and #15), but for 8 of them (Q event #3, #4, #5, #7, #9, #10, #12 and #13) the flatty shape makes the exercise very uncertain. As well for the Q events #8 and #11 showing a double peak, the shape of the hydrograph itself is then explained more by the fluctuations of the rainfall amounts than by the dynamic of the hydrological processes. This statement led us to use event-scale metrics for the hydrological response. We will add the Figure 4 to the supplementary material to illustrate our comment.

[Figure]

*Figure 4. River quickflow for 15 rainfall events causing a noticeable river reaction. The length of events is normalized.*

1d - Also, although the analysis is designed mainly for discharge estimation, results should be also interpreted in terms of rainfall (e.g. resulting areal rainfall (extremes) for different rain gauge network densities, spatial rainfall characteristics…).

We agree that these metrics are finally not fully exploited. We propose to add a short analysis on the impact of the raingauge density over i) the number of misestimated events and ii) the maximum rainfall intensities. The figures and text below will follow the existing paragraph of section 4.1.1 after changing its title from "Amounts and asymmetry" to the more general formulation of "Rainfall characteristics" (please read our answer to the point 3 of this review for further details concerning subsets). The references here are pointing to the table and figures of this document:

Relying on the rainfall events subset #2 composed of 23 rainfall events recorded by the full raingauge network (see Table 1), we tested what a partial raingauge network (all possible combinations of networks composed with less than 12 stations) would record, compared to the full raingauge network of 12 stations taken as a reference. The Figure 5 shows, in term of raingauge density, the number of events having the total amount of rainfall $P_{\text{TOTAL}}$ overestimated or underestimated by a factor 2. We globally observe a misestimation inversely proportional to the raingauge density, with up to 3 events overestimated and 8 events underestimated with the lowest raingauge density of 0.07 raingauge per km² (1 raingauge). It is necessary to reach 0.82 raingauges per km² (11 stations) to no longer have events misestimated by a factor 2. We also observe, with few raingauges, a strongest trend to underestimate than overestimate events. The invoked reason is that facing a heterogeneous event for which a good spatial resolution of the rainfall field is needed, it is statistically more probable to miss the localized important part of the rainfall field than capture it.

[Figure]

*Figure 5. Number of rainfall events for which the total amount of rainfall is overestimated or underestimated by a factor 2, according to the raingauge density, going from 0.07 raingauges per km² (1 raingauge within the catchment) to 0.82 raingauges per km² (11 raingauges). For each raingauge density, all possible combination of raingauge network is tested. The reference value is estimated from the full 12-raingauge network. The bottom and top of each boxes are respectively the 25th and 75th percentiles of the sample. The line in the middle of each box is the sample median. The whiskers go up to 1.5 times the interquartile range away from the bottom or top of the boxes and values beyond are marked as outliers with circles.*

The Figure 6 presents in the same way the maximum error encountered on the maximum rainfall intensity over 10 minutes $P_{MAX}(10\ min)$. We logically notice an inversely proportional trend, minimizing the error while the raingauge density increases. The figure also shows that in general a low raingauge density tend to overestimate more than underestimate the $P_{MAX}(10\ min)$. This bias originates the large footprint associated to a low raingauge density, increasing the disparities between the measuring points while interpolating the rainfall fields.

[Figure]

*Figure 6. Error on the maximum rainfall over 10 minutes according to the raingauge density. For each raingauge density, all possible combinations of raingauge network is tested. The reference value is estimated from the full 12-raingauge network.*

2 - Based on the comment before, the impact of the rain gauge network densities (and rain gauge locations) on the runoff is not analysed. In the additionally uploaded comment the main author states a rainfall-runoff modelling would go beyond the scope of the study. I do not agree with that and recommend this modelling approach to analyse the impact on the resulting runoff itself instead on single runoff statistics. To attribute the spatial rainfall variability, a distributed rainfall-runoff model would be the best solution.

Following this suggestion and the suggestion of the referee #2, we implemented an event-based modelling approach (note: the following answer is identical to the answer 8b given to the referee #2). The runoff response of Vallon de Nant to rainfall forcing is modeled by a semi-distributed model. This model first simulates the mobilization of water at the sub-catchment scale (here 25 sub-catchments are defined over the Vallon de Nant) using a SCS runoff curve number approach. Next, stream discharge is obtained by convoluting the resulting hillslope responses with a travel path distribution derived from the stream network geometry (Schaefli et al., 2013). In the current version (to be refined) the subcatchments and the stream network geometry are identified using *TopoToolbox* (https://topotoolbox.wrrdpress.com) (Figure 7)[1], in which travel paths correspond to the distance between the bottom part of each sub-catchment and the catchment outlet. In this model we focus on the fast response (i.e. runoff) of the catchment, and baseflow (defined here as the average discharge during the 30 min preceding event start) is subtracted from the actual discharge prior to runoff modeling. For calibration, the model is run using the mean of the 20 stochastic rainfall realizations as reference input; it is then calibrated against observed runoff (i.e. discharge - baseflow) through likelihood maximization assuming that the model residuals are normally distributed (e.g. Schaefli et al., 2007). After calibration the event-based runoff model is applied to the different network configurations to test how rain gauge network geometry influences the simulated runoff response. As the stochastic rainfall interpolation cannot be performed with a number of observation points as low as 3 stations (or less), we use the Thiessen polygons method to interpolate the rainfall fields from the 1 to 3-station raingauge network. The results of this model are all shown in the Appendix at the end of this document.

What we can say at this stage is that this kind of typical conceptual event-based hydrological model cannot reproduce all observed events equally well (Appendix 1). This would require in-depth analysis of different subsurface flow mechanisms related also to snow melt and shallow-groundwater recharge, work that is ongoing in this catchment. What is clear is that the simulations with the worst 1 station network are completely off. In exchange, the simulations with the best 1-station, 2-station or 3-stations network is always close to the simulations obtained with the stochastic rainfall fields, which underlines the value of the station network selection methodology in the submitted paper. The analysis furthermore shows that an ill-placed weather station can result in completely erroneous runoff simulation, whereas a network of at least 3 stations results in much better runoff simulations. This conclusion would not have been possible without the high density network observations. However, this model experiment cannot shed further light on the value of the high density networks as the ability of the model to reproduce streamflow responses is not good enough for clear conclusions. This cannot be easily solved with another conceptual model (we tried already other conceptual model structures, e.g. Benoit, 2020) nor with any "out-of-the-shelf" model, which do not exist for high alpine headwater catchments. The development of a fully distributed high resolution (e.g. 10 m x 10m) physical model with the inference of distributed model parameter fields is beyond the reach of this study.

In any case, we can try to include some key results from the modelling study in the revised version.
* * *
[1] An automatic identification of subcatchments corresponding to a manually identified stream network (i.e. identified in the field) is non trivial; solution to be found.

[Figure]

*Figure 7. Map of the Vallon the Nant showing the 25 subcatchments and the stream network geometry used for the modelization.*

3 - Also, I was wondering why is there not a consistent number of events analysed throughout the manuscript. I understand that there are always measuring issues and maybe some observations are questionable, but then please remove them at the beginning. There could be one number of rainfall events considered and one subset of them for discharge analysis, but at the current state results from different subsections cannot be compared with each other due to the different populations of considered events.

Indeed, we have different subsets of events. Among i) the initial 48 rainfall events that were recorded by at least 7 raingauges, which is enough for computing the spatial rainfall pattern metrics (Section 3.2.2), we choose ii) the subset of 23 rainfall events recorded by the full network of 12 raingauges to evaluate the best partial network (Section 3.5); the iii) 15 rainfall events associated with a streamflow reaction was reduced to iv) 14 events after discarding the July 24th event identified as an outlier. This last subset is used for computations involving the hydrological response. Although we are often referring to the July 24th outlier event and recall in the figure captions that this particular event is out of the axis limits, it is not included in our computations. We will make this clearer in the article by adding the above sentences and the synthetic Table 1 that summarizes and numbers the different subsets used within this study. However, we stick to the idea of keeping datasets as large as possible in order to increase the statistical robustness of our results.

*Table 1. Summary of the different subsets of rainfall events ($P_{event}$) used within this study.*

[Figure]

4 - L25-27 It should be mentioned here again that this issue is related to mountainous areas and is not a problem in general.

Indeed, it is useful to precise it in this sentence too in order to avoid confusion. We will mention it.

5 - Fig. 2 I don't see the additional worth of showing Fig. 2 and recommend to leave it out, especially since it is included in the supplement as Fig S2 as well.

We agree that the weir picture of the Figure 2 of the paper rather has an illustrative role and is not essential in a hydrology paper. As the Figure S2 in the supplementary material also fulfills this aim, the Figure 2 will be removed from the main part of the article.

6 - L90 "average elevation" Please change to mean or median, depending on how you determined the "average" value.

We agree to use "mean" rather than "average", it will be corrected in the revised version of the paper.

7 - L117-118 The construction of the rating curve is not interesting for the manuscript and can be left out, also the elements regarding its construction in the supplement.

We agree. The details concerning the rating curve construction and error estimate will be move to an appendix.

8a - L154-155 The term interpolation is not suitable in my opinion due to the rainfall generation mechanisms behind. I suggest "areal rainfall is generated after Benoit et al. (2018a) by constraining actual observations at rain gauge locations". The authors should give a less brief explanation, since in the cited manuscript different versions are applied for rainfall generation (three versions due to different covariance models) and it remains unclear for the reader, which model is used for the current study.

Thanks, we will add some more details in an appendix.

8b - Why did the authors choose this rainfall generation instead of a regionalization approach as kriging (maybe with altitude as additional information), inverse distance weighting or Thiessen polygons. The latter is chosen later in the manuscript nevertheless due to computational efforts, so why not for the whole study? Was it the authors intention to add an uncertainty analysis.

Indeed, we choose to use the stochastic approach for the valuable estimation of the errors it provides (in response to the first round of reviews during the first submission; the original manuscript used Thiessen only). The Thiessen method also used throughout this paper fills the weak points of the first method, namely i) the computation time, which is very short using the Thiessen method and allows to explore within a reasonable amount of time all the possible combinations of raingauge networks for their optimization, and ii) to calculate rainfall fields with a low number of raingauges. For this last point the stochastic method require at least 5 stations to capture correctly the spatial and temporal rainfall characteristics.

Concerning the altitude effect on rainfall, we do not observe any trend ($R^2 = 0.06$) between the cumulated rainfall per station and the altitude. This information will be added in "4.1.1 Rainfall characteristics" (please see 1d concerning the title change of this section). The Figure 8 below is shown as illustration purpose for this document.

[Figure]

*Figure 8. Cumulated rainfall at each raingauge vs. altitude of each raingauge.*

9 - L154-163 The authors should bring this argument in context with the catchment concentration time.

This statement probably refers to "Using the interpolated rainfall fields, rainfall events were identified as rainy periods separated by at least 90 minutes without rain. This inter-event duration was selected based on the observed delay between rainfall onset and streamflow response for the large event recorded on August 23rd (detailed in the part 2 of supplementary material); the streamflow reaction to the first half-hour of this rainfall event was caused only by rainfall in the southern half of the catchment (stations 8 to 12)."

Following e.g. Dingman (2002), "the time of concentration Tc [is] defined as the time it takes for water to travel from the hydraulically most distant part of the contributing area to the outlet". It is difficult to determine in practice due to unknown flow paths. That is exactly why we choose the event on August 23$^{rd}$, which happened far away from the outlet as an indicator for this travel time to the outlet. We will make this explanation clearer and add a reference to the concentration time.

10 - L165-166 The location of the line chosen for the splitting of the catchment seems to be chosen arbitrary. Would a line constructed perpendicular to the main flow direction of the river (or even better, not a straight line but following the lines perpendicular to the isohypses to separate flows exactly) lead to more representable results, since the catchment is then split into a real upper and lower part? Or (thinking the other way around) does it not matter at all and the splitting line could be also drawn from South to North as long as both parts have the same area?

Indeed, the choice of a west-east line splitting the catchment into 2 parts of equal area is arbitrary. We agree that a line splitting the catchment by crossing perpendicularly the river (or the close solution of following a line perpendicular to the isohypses) are better solutions, as they are a more general answer to this problem of splitting the catchment into two areas, whatever is its orientation. We believe though that the easy geometric solution we choose for the computations would not give, in this particular case, fundamentally different results.

We tried the solution of splitting the area following a given isohypse (around 2023 m asl, to split the catchment into two parts of equal area) but we finally discard this option later while introducing $D_{HILLS}$, which is a redundant metrics ($R^2 = 0.76$) that gives a more accurate description of the localization of the rainfall within the catchment.

However, we did not consider splitting the catchment following a north/south (or some titled line to follow the general orientation of the catchment) line. We would not do it in purpose of testing the relevance of the orientation of the splitting line, but because we think it could actually be a good choice. Such a line would roughly separate the steep slopes on the east side from the grassy slopes on the west side (see Figure 2 of the article). For this case study this choice would interestingly isolate geomorphological distinct areas. We unfortunately did not have time to explore this solution, but a sentence will be added in "3.2.2 Spatial rainfall pattern metrics" to criticize our choice of the north/south catchment splitting and the potential solutions to be considered.

11 - L211-215 I suggest to move this paragraph to the beginning of section 3.3.2

This comment refers to the statement on how baseflow is separated ("The beginning and the end of the streamflow response determine the initial and final baseflow, respectively; the streamflow volume above the line connecting these two points is considered here as fast runoff.") and we think that it is an integral part of event identification. We will rename the subsection "3.3.1 Event identification" to "3.3.1 Event and quickflow identification".

12 - L217-218 The authors declare volume and lag time as "the two key characteristics of streamflow reaction". I do not agree with that. The most important characteristic is peak flow, followed by volume and then lag time and flatting behaviour. Even if all characteristics are considered equal important, the authors should state why peak is not considered in the study. If there were attempts to include peaks which did not work, the authors should state so as "lessons learned" in the manuscript.

In agreement with our answer to the point 1c of this document, we will give more details about the reasons that led us to discard the peak flow.

13 - L219-221 Is this criterion developed by the authors or should a reference be cited in this context? How was 1/3 chosen as threshold? This value should be catchment-dependent in my opinion, or not? Please clarify.

This comment refers to the runoff coefficient. We will add a reference to a classic textbook (Dingman, 2002).

Classical lag time definitions are e.g. the lag between the start of the effective rainfall (the one that creates a reaction) and peak flow. As discussed above, the concept of peak flow is difficult to apply to our observed events. A classically used alternative is the centroid-lag (lag between the centroids of rainfall and of streamflow), which is a useful to characterize the response time (e.g. Dingman, 2002). Given the varying shape of our hydrographs, we empirically tested different lag formulations; the lag between 1/3 of rainfall and 1/3 of streamflow gives the best results in the regression analysis. We will clarify this in the revised version.

14 - L222 Why is this criterion "1/3 of the rainfall amount" more robust than "start of the rainfall event", although both starting points are linear correlated?

Thanks for pointing this out. The formulation was not well chosen; the start of the rainfall event will not contain information about the actual start of effective rainfall (the rainfall that creates a reaction), which depends on antecedent storage conditions. We will reformulate (see also comment 13).

15 - L275 Same differences lead to higher asymmetry values for smaller values. To avoid a misinterpretation ("Interestingly…") Pnorth and Psouth could be normalized by the mean event rainfall amount. This would provide deeper insights, especially since larger differences between both parts cannot be seen in the current approach if they occur for events with high rainfall amounts.

We agree. We will modify the Table 2 of the paper (and the supplementary material) by adding two columns with $P_{NORTH}/P_{ALL}$ and $P_{SOUTH}/P_{ALL}$. The Table 2 of the paper will be modified as the Table 2 below.

Table 2. Modified table of the "List of recorded precipitation events with streamflow reaction (in 2018)".

| Date | $P_{DURATION}$ [min] | $Q_{DURATION}$ [min] | $\Delta_{P/Q}$ [min] | $P_{ALL}$ [mm] | $P_{NORTH}$ [mm] | $P_{SOUTH}$ [mm] | $P_{NORTH}/P_{ALL}$ [-] | $P_{SOUTH}/P_{ALL}$ [-] | $W_{3\,days}$ [mm] | $Q_{UNIT}$ [mm] | $Q_{FAST}$ [mm] | RC [-] | $I_{ASYM}$ [-] | $D_{HILLS}$ [m] | $D_{STREAM}$ [m] | HAND [m] |
|---|---|---|---|---|---|---|---|---|---|---|---|---|---|---|---|---|
| 2-Jul | 42 | 44 | 24 | 7.7 | 4.1 | 3.6 | 0.53 | 0.47 | 3.2 | 7.9 | 0.9 | 0.12 | -0.06 | 1521 | 4008 | 611 |
| 3-Jul | 40 | 135 | 23 | 12.1 | 7.4 | 4.6 | 0.62 | 0.38 | 12.7 | 7.5 | 8.5 | 0.71 | -0.24 | 1336 | 3842 | 550 |
| 5-Jul | 224 | 309 | 71 | 8.2 | 4.0 | 4.2 | 0.49 | 0.51 | 29.8 | 6.0 | 6.0 | 0.74 | 0.03 | 755 | 4374 | 350 |
| 6-Jul | 478 | 587 | 65 | 20.2 | 8.6 | 11.6 | 0.43 | 0.57 | 40.3 | 5.8 | 25.9 | 1.29 | 0.15 | 874 | 4450 | 355 |
| 14-Jul | 358 | 302 | 49 | 18.7 | 10.5 | 8.2 | 0.56 | 0.44 | 0.0 | 4.5 | 12.9 | 0.69 | -0.12 | 1263 | 3574 | 554 |
| 15-Jul | 136 | 281 | 33 | 10.7 | 6.0 | 4.7 | 0.56 | 0.44 | 18.9 | 5.5 | 9.5 | 0.89 | -0.13 | 1122 | 3377 | 528 |
| 20-Jul | 288 | 228 | 49 | 18.8 | 8.6 | 10.2 | 0.46 | 0.54 | 3.4 | 4.8 | 14.2 | 0.76 | 0.09 | 1282 | 3823 | 541 |
| 24-Jul | 220 | 229 | 45 | 8.0 | 7.5 | 0.5 | 0.94 | 0.06 | 12.2 | 3.1 | 30.4 | 3.78 | 0.02 | 740 | 2184 | 419 |
| 14-Aug | 204 | 152 | 47 | 11.1 | 4 | 7.1 | 0.37 | 0.64 | 10.2 | 4.0 | 7.8 | 0.70 | 0.27 | 1286 | 4305 | 540 |
| 17-Aug | 152 | 109 | 38 | 11.9 | 6.2 | 5.7 | 0.52 | 0.48 | 17.5 | 3.2 | 4.9 | 0.42 | -0.04 | 1122 | 3780 | 490 |
| 23-Aug | 388 | 237 | 47 | 22.1 | 8.8 | 13.3 | 0.40 | 0.60 | 5.4 | 2.4 | 13.5 | 0.61 | 0.20 | 1371 | 3756 | 563 |
| 24-Aug | 158 | 107 | 40 | 8.1 | 4.4 | 3.7 | 0.54 | 0.46 | 29.5 | 4.1 | 6.5 | 0.81 | -0.08 | 692 | 4114 | 320 |
| 29-Aug | 72 | 116 | 48 | 4.8 | 2.2 | 2.6 | 0.46 | 0.54 | 12.4 | 3.0 | 2.3 | 0.48 | 0.07 | 1207 | 3526 | 524 |
| 01-sept | 628 | 341 | 101 | 11.4 | 4.3 | 7.2 | 0.38 | 0.63 | 20.4 | 3.4 | 16.4 | 1.44 | 0.25 | 725 | 4487 | 331 |
| 13-sept | 370 | 59 | 45 | 10.9 | 7.0 | 3.8 | 0.65 | 0.35 | 0.0 | 2.6 | 4.4 | 0.40 | -0.29 | 1291 | 3594 | 556 |

16 - L323-327. I cannot follow the argumentation here. Please explain in detail how you achieve this conclusion and consider at least one or two sentences for each argument.

The corresponding original text will be complemented as follows:

"The strong correlation between rainfall amounts and $Q_{FAST}$ (0.77, Table 3) suggests that streamflow reactions are triggered by saturation-excess, rather than by infiltration capacity-excess: [NEW] If saturation is exceeded, every unit of rainfall will lead to a corresponding unit increase of streamflow, hence a strong linear correlation to rainfall amounts. Furthermore, saturation-excess also implies that a longer rainfall event leads to more streamflow reaction (once the saturation threshold is reached, all rainfall contributes to streamflow). If, on the contrary, the driving process was the exceedance of infiltration capacity, then only rainfall intensities above the capacity threshold would trigger a corresponding streamflow increase, small rainfall amounts would trigger almost no reaction. In this case (infiltration-excess), there would be no linear correlation between rainfall amounts or rainfall duration and streamflow amounts, but a strong correlation between streamflow amounts and high or maximum precipitation intensity. [end NEW] [REFORMULATE] Saturation-excess as a main driver of the fast streamflow response is confirmed by i) the absence of correlation between maximum rainfall intensity over 10 minutes and the RC (Table 3) [end REFORMULATE], ii) the strong correlation between rainfall duration and $Q_{FAST}$ (0.73) and iii) by the clear threshold effect for the generation of streamflow as a function of total event rainfall (Figure 9); a streamflow reaction only occurs for total rainfall higher than 5 mm".

17 - L330 "to reach a higher "RC" Please rephrase, the manuscript is about observations, not modelling.

We will rephrase "to reach a higher RC, we need a higher level of saturation [...]" by "we observe a higher RC when the level of saturation increases [...]".

18 - L341 composites: If there is a differentiation into wet and dry state, how do the authors achieve only one value for each criterion? Are two values estimated (for wet and dry) and then the arithmetic mean is mentioned? Please clarify!

Instead of analyzing all the events with an identical "dry" or "wet" network extent all along, for the composite network the "dry" or "wet" state of the network is chosen each time at the beginning of each event. The network extent is based on the initial wetness conditions, by looking at the total amount of precipitations that fell during the 3 previous days before the beginning of the event. If this amount is over or equal to the threshold of 20 mm of rainfall, we use the "wet" network; below this threshold we pick the dry network. Thus, the estimated value

is calculated once, using one or the other of the networks. The process will be more detailed into the revised version of the article.

19 - L351-355 It would be nice to have a table with all criteria, where it is stated which one was removed (and why) and which ones were kept. Maybe the information can be added to Table 5 or 6?!

Few criteria are retained at the end of the regression analysis. We believe it will be clearer to detail more the criteria which are kept and those which are removed, and the associated reasons, directly in the text. We will improve this part accordingly.

20 - L354 Again, it feels as the number of considered events changes among all subsections.

Please refer to our answer to the point 3 of this review for details.

21 - L380 What is the reason for IASYM preference in the Southern part? Due to the steeper areas? I would have estimated Northern part, since the hydrograph would have already been smoothed when originated in the South. Please try to find physical explanations to your results.

This refers to the statement "And for a single station network, the metric $I_{ASYM}$ prefers a station location in the southern part rather than in the northern part."

This statement is wrong, thank you for pointing out. It must be a legacy effect and we apologize. We will remove i) this sentence and ii) the plot of the best 1-station network for $I_{ASYM}$ in the Figure 11 of the article, as of course $I_{ASYM}$ cannot be defined for a single station.

22 - General: Please double-check the abbreviation for "meter above sea level"; I have only seen "m a.s.l." and "m asl" so far, but not "m asl."

Thank you for this observation, it will be corrected using "m asl" throughout the paper.

23 - L155 Benoit et al. 2018 <- a or b? I assume a.

It is 2018a indeed, it will be corrected.

24 - Eq 2, 3, 4 I'm a bit confused what rainfall characteristic is used as input for these equations. Is every raster cell with rainfall used (so I understood it from the text) or only the centre of the rainfall events (as mentioned in Table 1)?

Thank you for pointing out that this part is not so clear. Within these 3 equations we use $P(i,j,t)$, the rainfall amount previously calculated using the stochastic method (section 3.2.1) for each of the 10 x 10 meters grid cell (referenced by $i$ and $j$) at each 2-minute time step $t$. The rainfall characteristic and space-time resolution will be specified in the text introducing the first equation.

This remark also reveals that the descriptions of $D_{HILLS}$, $D_{STREAM}$ and $HAND$ in the Table 1 of the paper must be corrected by removing the "mean" at the beginning of each definition (e.g. for $D_{HILLS}$: "Mean distance of rainfall spatial center of mass to stream network (along hillslopes)" becomes "Distance of rainfall spatial center of mass to stream network (along hillslopes)").

25 - L163 "overlooked" -> ignored

Thanks, it will be corrected.

26 - Eq. 2, 3, 4 The term in the numerator should be put in brackets (Eq. 2: "P(..)dHills" -> "(P(..)dHills)")

Thank you for pointing out this oversight, it will be corrected.

27 - L195 DHAND is not a distance as indicated by the D, and in the text the variable is introduced with HAND. I suggest to stick to HAND throughout the manuscript to avoid confusions with the other two "real" distances".

We agree that the D is confusing. We will stick to the HAND abbreviation throughout the article.

28 - L202 Section 3.5 includes no network extent description. Is it missing in the manuscript?

The network extent is briefly introduced in the section 2, but a description of the composite network is missing. A description will be added in the section 3.4 Rainfall-streamflow response characterization, and the reference L202 will be corrected accordingly.

29 - L268 317.8 mm – Is it areal rainfall amount sum or sum over all stations?

We will specify that the value of 317.8 mm is the areal rainfall amount.

30 - L268-269 please provide also the mean values, not only the highest and lowest values, so that the reader get a "feeling" for the rainfall events.

We agree. The mean values (6.6 mm for the rainfall amount and 2h47 for the rainfall event duration) will be added to the revised version of the paper.

31 - L275 again, please don't use the term average, use mean or median to be more concise. Since Iasym can be positive and negative, the median of its absolute values would be worth to show instead of just the mean, since positive and negative values are levelling out each other.

We agree again that the "average" word is not adapted, and it will be outlawed from the manuscript. Indeed, in this case using the median value of $I_{ASYM}$ (0.025) is better than using the mean. It will be corrected.

32 - Fig. 5 and 6 For a logical order the figures should show the rainfall events first, followed by the discharge plot.

We agree with this point. The figures will be corrected accordingly.

33 - L279 "One strongly asymmetric and high intensity event" -> "One strong asymmetric and very intense event"

Thank you for the suggestion, it will be modified.

34 - L283 A volume can't be fast (check also for later occurrences…)

Thank you, this occurrence and the others will be corrected.

35 - L288 In the sentence before authors mention that the number of events under consideration are reduced by "1", but here again 48 events are studied (also in the following subsections).

Indeed, it is confusing. The line it is referred to at the end of 4.1.1 "This event and its streamflow reaction are excluded from further analysis" will be replaces by "This event and its streamflow reaction are excluded from further analysis involving the hydrological response".

36 - L289 The authors should state what wet and dry networks are. I found it later in the caption of Table 1 in S1, but it would lead to clarifications here. Also, the Table 1 in S1 should be shown in the manuscript, since the written part in Section 4.1.2 is more confusing than explaining for me.

We will include a reference to the states (shown in Figure 1) at the beginning of 4.1.2 Stream network distance metrics. Their extent is first introduced in Section 2.

37a - Fig9 "events without reaction are not shown" belongs to part b), not a). Please correct the caption.

Thanks, it will be corrected.

37b - General: Maybe I missed it, but which temporal resolution was used to calculate the correlation (and other criteria)? 2min as this is the resolution of the rain gauge? Or are values aggregated up to e.g. 1h? This has a high impact on the values of the correlation coefficient.

The temporal resolution of times series used for correlation calculations is 2 minutes. The correlation between events is done at the event-scale. Occurrences will be checked and detailed throughout the article.

38 - L339 "absence of correlation". Correlation cannot be absent. Better to speak of low correlation or provide absolute values.

Thanks, we will provide values and improve the formulation.

39 - L384-386 "is assessed", "is evaluated" – two verbs, please rephrase the sentence.

Thanks. The sentence will be corrected by "Considering the small dataset underlying this analysis (23 events), the robustness of the best networks is assessed for two selected metrics (for the $P_{ALL}$ and $I_{ASYM}$) by re-computing the optimal network when between 1 and 3 events are removed from the dataset".

40 - L402 "what we previously thought"? What was the hypothesis of the authors before?

This statement refers to "The fact that $D_{STREAM}$ outperforms here $D_{HILLS}$ for the prediction of RC and lag time is an interesting result: it underlines that even in steep environments, with a priori fast instream processes and limited storage, the riparian area and related subsurface exchange processes could play a more prominent role than what we previously thought".

We will reformulate. The classic hypothesis (e.g. Nicotina et al., 2008) is that in steep environments, the travel time in hillslopes strongly dominates over travel times in the stream network, because instream velocities are very fast compared to travel times in hillslopes. The fact that the travel distance in the stream network explains nevertheless more of the RC variation than $D_{HILLS}$ might be an indirect effect: the longer the travel distance in the stream network, the more likely are delays due to exchange with groundwater in the riparian area. This will be explained in the revised version.

41 - L421 "three station network" It would be nice to provide the resulting density here as well as "(general) recommendation".

A 3-station network corresponds to 0.22 raingauges per km². This value will be added, and we will check throughout the article if such corresponding density values must be added as well.

REFERENCES

Benoit, L. (2020). High Resolution Stochastic Modelling of Local Rain Fields (PhD Thesis). University of Lausanne, Faculty of Geosciences and Environment, Institute of Earth Surface Dynamics, Lausanne, Switzerland.

Dingman, S.L. Physical Hydrology. 2nd Edition, Prentice Hall, Upper Saddle River, 2002.

Nicótina, L., Alessi Celegon, E., Rinaldo, A., and Marani, M.: On the impact of rainfall patterns on the hydrologic response, Water Resour Res, 44, W12401, 10.1029/2007WR006654, 2008.

Schaefli, B., Talamba, D.B., Musy, A., 2007. Quantifying hydrological modeling errors through a mixture of normal distributions. J. Hydrol 332, 303–315. https://doi.org/ 10.1016/j.jhydrol.2006.07.005.

**APPENDIX 1: Model results for all of the 15 events**

[Figure]

| | |
|---|---|
| —— | Full network (12 gauges) - stoch. interp. |
| —— | Best sub-network (3 gauges) - Thiessen |
| – – – | Best sub-network (2 gauges) - Thiessen |
| ·········· | Best sub-network (1 gauge) - Thiessen |
| —— | Worth sub-network (3 gauges) - Thiessen |
| – – – | Worth sub-network (2 gauges) - Thiessen |
| ·········· | Worth sub-network (1 gauges) - Thiessen |

Q event #1 (July 2nd)

On each figure the Y-axis of each hydrograph is in m³/s.

- The black curve is the observed streamflow.
- The 20 blue curves correspond the simulated streamflow based on the 20 possible rainfall fields from the stochastic interpolation method (12-station raingauge network).
- The plain, dashed and dotted red lines are resp. the simulated streamflow using the best 1-station (station #5), 2-station (stations #2 and #9) and 3-station (stations #2, #5 and #11) raingauge network, using the Thiessen polygons interpolation method.
- The plain, dashed and dotted purple lines are resp. the simulated streamflow using the worst 1-station (station #1), 2-station (stations #1 and #3) and 3-station (stations #1, #3 and #4) raingauge network, using the Thiessen polygons interpolation method.

[Figure]

Q event #2 (July 3rd)

[Figure]

Q event #3 (July 5$^{th}$)

[Figure]

Q event #4 (July 6$^{th}$)

Q event #5 (July 14th)

[Figure]

Q event #6 (July 15th)

[Figure]

Q event #7 (July 20th)

[Figure]

Q event #8 (July 24th)

[Figure]

Q event #9 (August 14$^{th}$)

[Figure]

Q event #10 (August 17$^{th}$)

[Figure]

Q event #11 (August 23$^{rd}$)

[Figure]

Q event #12 (August 24th)

[Figure]

Q event #13 (August 29th)

[Figure]

Q event #14 (September 1st)

[Figure]

Q event #15 (September 13th)

---

## Author Comment (AC3) · 28 Sep 2020

Answer to Referee #2

1) I already reviewed the initial version of this paper, which aims at highlighting the values of high density rain gauges networks for hydrological purposes in a small catchment of mountainous areas. I still believe that the topic is interesting and relevant for the community. It furthermore has other potential applications in urban areas which are also small and quickly reactive catchments where rainfall variability has strong consequences.

Thank you very much for agreeing to review our paper a second time. Indeed, we did not mention urban hydrology in the new version. We will add references to this topic in the introduction (e.g. Cristiano et al., 2017) and discuss it in the conclusion.

2) The minor difficulties with regards to the presentation and understanding of the paper have been corrected. Results are now better presented with the new figures. However, the main point was not addressed, i.e. the fact that the authors aims at showing the importance of grasping the spatio-temporal variability of the rainfall process in the prediction of flows, but the chosen indicators are only event based averages.

First of all, we would like to point out that we accidentally used the formulation "runoff prediction" rather than "runoff coefficient prediction" in the abstract. This having said, we did not mean to pretend that we study the predictability of streamflow but well the relation between rainfall field characteristics and runoff event characteristics, which has indeed a long tradition in hydrology as a basis for model development and comparative hydrology (Merz et al., 2006).

We believe that a focus on the scale of runoff response events is fully justified. We use for our event-based analysis descriptors of two fundamental properties of runoff events, which refer to the time until a response occurs and the magnitude of the response; we choose here the runoff coefficient and a lag time.

We agree that these average streamflow event properties hide other interesting aspects of the hydrological response namely referring to the shape of the response and including e.g. the occurrence of double-peak response. A further detailed analysis of such double-peaked events versus single peaked events is however not possible for this small data set (see Figure showing all streamflow events at the end of this document)..

One additional descriptor of hydrograph shape could be what Tarasova et al. (2018) call the runoff event time scale, i.e. the ratio of runoff volume in mm and the runoff event peak flow in mm/day (see also Gaál et al., 2015). This descriptor could potentially shed light on the mixture of fast and slow runoff response processes. The identification of peak runoff is however extremely challenging for this case study because the moment of peak flow occurrence is often not well defined (see Figure 4 and our response to the point 8 of this document).

Another descriptor for this mixture could be the rising time of an event, quantifying the time to peak (normalized by the event duration). We refrained however from using descriptors requiring a precise quantification of event duration since the identification of an exact start and end time remains extremely challenging (Tarasova et al., 2018) and could largely affect the results in presence of a relatively small data set.

We would be happy to receive any further suggestions to find an additional runoff response shape descriptor.

Finally, this comment can also be read as a critic regarding the fact that we only use average indicators for the rainfall characterization. Please refer to our response to comment 7 for an answer to this point.

3) Furthermore, the main rainfall variability (which is at the core of the paper) indicator used is too simplistic since it is basically an asymmetry indicator on the total depth splitting the catchment in two. So I still think that indicators actually accounting for the spatiotemporal variability of the rainfall and hydrologic response should be implemented to actually address the stated topic of the paper. Implementing them requires major

modifications of the paper. I guess that this would enable to highlight more precisely the importance of dense networks of rainfall measurement devices.

We agree $I_{ASYM}$ is a simple indicator to capture the key rainfall field properties for the hydrological response. In other studies and namely in urban hydrology such an indicator is typically based e.g. on the variogram (Berne et al., 2004) or on the spatial moments of rainfall (Zoccatelli et al., 2011;Mei et al., 2014) of continuously observed rainfall fields (radar images).

The asymmetry indicator is just one of the indicators used in the study, along with the geomorphological distances, which correspond to the above first order spatial moments, albeit decomposed according to hillslope and stream network flow distances. We did not explicitly mention the link to the spatial rainfall moments since we were not aware of the link before writing this comment.

The link to rainfall spatial moments opens new perspectives: the first order moment captures the location of the rainfall centroid, the $2^{nd}$ order moment would assess the dispersion of the rainfall field relative to its mean location. We will test the usefulness of the $2^{nd}$ order moment for the revised version.

Furthermore, we would like to emphasize here that compared to e.g. the rainfall width function (see Figure 7 in the public discussion of the earlier version https://hess.copernicus.org/preprints/hess-2019-683/hess-2019-683-AC2-supplement.pdf), the asymmetry indicator has the potential to efficiently discriminate between different event types since it shows a considerable variability between the recorded events. We will provide additional evidence (figures in the Supplementary material) to underline this point.

Please refer also to our response to comment 7 below, referring to a similar topic.

4) l. 110 – 115: "The actual extent of the stream network is based on observations during dry and wet periods during Summer 2017 and its exact path was calculated using the Swiss digital elevation model at a resolution of 2 m (swissALTI3D, 2012)." I think that the intrinsic fractal nature of river networks should be mentioned and discussed. The concept of variable network used after also seems interesting.

Thanks for this interesting comment. As discussed by Rinaldo et al. (1995), the fractality of the basin shape response of the stream network extent is not  transferred to the dynamics of the hydrological response, in other words, as Rinaldo et al. put it "one can distinguish from the hydrologic response whether the basin is large or not simply from the regularity of the gauged record (the larger the basin the smoother the gauged trace)."

Our catchment (13.4 km²) is, however, likely not large enough to have a significantly different filtering effect for different network extents. This having said, it would be extremely tempting to develop a method to infer the network extent from the runoff event characteristics. We will mention this exciting outlook in the discussion of the revised version.

5) l. 150-151: "Some additional artefacts were recorded, probably generated by strong winds creating resonance. These periods have been manually removed from the data". It should clarified how the data was selected for being removed and what portion was removed.

We will give more details in the revised version. The removed periods are identified as artefacts because some stations were recording a signal representing a very strong and highly variable rainfall over hours, happening during periods that did obviously not have any actual rainfall. We checked therefore MeteoSwiss RADAR data (that, at least qualitatively, could not have miss such a major event) for confirmation. The weather stations recording wind within the catchment have shown a high wind velocity during these periods, leading to the suggestion of the resonance effect of the device. These details will be added to the revised version of the paper.

6) 1. 154-157: It is a great improvement to use this stochastic procedure. Nevertheless, I believe that more details on the interpolation procedure are needed. It should be clarified how the 20 samples are used (computing the error bars in 8-10)?

The rainfall fields are spatially interpolated from the point measurements. The stochastic model generates 20 different realizations that respect the spatial and temporal structure of rainfall measured by the raingauge network. The amount of precipitation is then spatially averaged, giving 20 possible values for each time step. The mean and standard deviation are computed at each time step based on these 20 realizations, and later at the event scale. These details will be added to the revised version of the paper.

7) Eq. 1 on I_ASYM. As already mentioned, it seems a too simplistic indicator to grasp spatio-temporal variability of the rainfall process. An initial simple suggestion could for instance be to look for the temporal evolution of I_ASYM during an event. But other indicators are needed.

The $I_{ASYM}$ indicator is specific to the elongated catchment shape, and the dimensions of the two defined areas match in our view the temporal scale (event scale) considered in this case. It complements the spatial moments based on geomorphological distance ($D_{HILLS}$ and $D_{STREAM}$).

The temporal evolution of $I_{ASYM}$ (see Figure 1) shows indeed interesting patterns. For individual events, it sheds further light on what might have caused e.g. a double peak response shape.

For the double peak runoff event #8 (see Figure 4), corresponding to the double rainfall peak event #16 (see Figure 1) stationary on the northern part of the catchment ($I_{ASYM}$ highly negative throughout the event), we can conclude that the double peak runoff response is due to its location within the northern part of the catchment, remaining close to the outlet during its entire duration. The geomorphological distances (see event #16 on Figure 2 for $D_{HILLS}$ and Figure 3 for $D_{STREAM}$) shows indeed the stationarity of the rainfall field center relatively to the stream network, but not its location within the catchment.

Unfortunately, some of the rainfall events with very clear temporal evolution of $I_{ASYM}$ did not give any significant runoff response (e.g. rainfall event #5 and #21).

A comparison of the $I_{ASYM}$ and the geomorphological distances evolution during these events will shed further light on these events (to be done for the revised version). We will add at least a qualitative discussion of the temporal evolution of the rainfall indicators and how they are linked to the observed runoff responses. The $I_{ASYM}$, $D_{HILLS}$ (computed using wet and dry networks) and $D_{STREAM}$ (computed using wet and dry networks) for all the 48 rainfall events are in Appendix 1 of this document.

[Figure]

Figure 1. Evolution of rainfall intensity and $I_{ASYM}$ for the 15 rainfall events (P event) associated with a river reaction (Q event).

[Figure]

Figure 2. Evolution of $D_{HILLS}$ for the 15 rainfall events (P event) associated with a river reaction (Q event).

[Figure]

*Figure 3. Evolution of $D_{STREAM}$ for the 15 rainfall events (P event) associated with a river reaction (Q event).*

8a) l. 212-215 : the explanation on why not using streamflow variations (notably peak flow) is not very convincing.

The timing of peak value is difficult to identify. As visible on the Figure 4 this metric can be used for few events only (Q event #1, #2, #6, #14 and #15). Thus, the hydrograph shape of the observed events is explained more by the fluctuations of the rainfall amounts than by the dynamic of the hydrological processes as is e.g. clearly visible for rainfall event#16 (double peak rainfall event) and runoff response event #8 (double peak runoff response). This observation led us to use event-scale metrics for the hydrological response. In the revised version, we will clearly make this point (based on observed data) at the start of the presentation of the developed methodology and the Figure 4 will be added to the supplementary material.

[Figure]

*Figure 4. River quickflow for 15 rainfall events (P event) causing a noticeable river reaction (Q event). The length of events is normalized.*

8b) If the purpose is to investigate the importance of spatiotemporal variability, I guess studying the temporal variability of the simulated streamflow is needed.

Following this suggestion and the suggestion of the referee #1, we implemented an event-based modelling approach (note: the following answer is identical to the answer 2 given to the referee #1). The runoff response of Vallon de Nant to rainfall forcing is modeled by a semi-distributed model. This model first simulates the mobilization of water at the sub-catchment scale (here 25 sub-catchments are defined over the Vallon de Nant) using a SCS runoff curve number approach. Next, stream discharge is obtained by convoluting the resulting hillslope responses with a travel path distribution derived from the stream network geometry (Schaefli et al., 2013). In the current version (to be refined) the subcatchments and the stream network geometry are identified using *TopoToolbox* (https://topotoolbox.wrrdpress.com) (Figure 5)[1], in which travel paths correspond to the distance between the bottom part of each sub-catchment and the catchment outlet. In this model we focus on the fast response (i.e. runoff) of the catchment, and baseflow (defined here as the average discharge during the 30 min preceding event start) is subtracted from the actual discharge prior to runoff modeling. For calibration, the model is run using the mean of the 20 stochastic rainfall realizations as reference input; it is then calibrated against observed runoff (i.e. discharge - baseflow) through likelihood maximization assuming that the model residuals are normally distributed (e.g. Schaefli et al., 2007). After calibration the event-based runoff model is applied to the different network configurations to test how rain gauge network geometry influences the simulated runoff response. As the stochastic rainfall interpolation cannot be performed with a number of observation points as low as 3 stations (or less), we use the Thiessen polygons method to interpolate the rainfall fields from the 1 to 3-station raingauge network. The results of this model are all shown in the Appendix 2 at the end of this document.

What we can say at this stage is that this kind of typical conceptual event-based hydrological model cannot reproduce all observed events equally well (Appendix 1). This would require in-depth analysis of different subsurface flow mechanisms related also to snow melt and shallow-groundwater recharge, work that is ongoing in this catchment. What is clear is that the simulations with the worst 1 station network are completely off. In exchange, the simulations with the best 1-station, 2-station or 3-stations network is always close to the simulations obtained with the stochastic rainfall fields, which underlines the value of the station network selection methodology in the submitted paper. The analysis furthermore shows that an ill-placed weather station can result in completely erroneous runoff simulation, whereas a network of at least 3 stations results in much better runoff simulations. This conclusion would not have been possible without the high density network observations. However, this model experiment cannot shed further light on the value of the high density networks as the ability of the model to reproduce streamflow responses is not good enough for clear conclusions. This cannot be easily solved with another conceptual model (we tried already other conceptual model structures, e.g. Benoit, 2020) nor with any "out-of-the-shelf" model, which do not exist for high alpine headwater catchments. The development of a fully distributed high resolution (e.g. 10 m x 10m) physical model with the inference of distributed model parameter fields is beyond the reach of this study.

In any case, we can try to include some key results from the modelling study in the revised version.
* * *
[1] An automatic identification of subcatchments corresponding to a manually identified stream network (i.e. identified in the field) is non trivial; solution to be found.

[Figure]

*Figure 5. Map of the Vallon the Nant showing the 25 subcatchments and the stream network geometry used for the modelization.*

REFERENCES

Addor, N., Nearing, G., Prieto, C., Newman, A. J., Le Vine, N., and Clark, M. P.: A Ranking of Hydrological Signatures Based on Their Predictability in Space, Water Resources Research, 54, 8792-8812, 10.1029/2018WR022606, 2018.

Berne, A., Delrieu, G., Creutin, J.-D., and Obled, C.: Temporal and spatial resolution of rainfall measurements required for urban hydrology, Journal of Hydrology, 299, 166-179, https://doi.org/10.1016/j.jhydrol.2004.08.002, 2004.

Cristiano, E., ten Veldhuis, M. C., and van De Giesen, N.: Spatial and temporal variability of rainfall and their effects on hydrological response in urban areas - a review, Hydrol Earth Syst Sc, 21, 3859-3878, 10.5194/hess-21-3859-2017, 2017.

Gaál, L., Szolgay, J., Kohnová, S., Hlavčová, K., Parajka, J., Viglione, A., Merz, R., and Blöschl, G.: Dependence between flood peaks and volumes: a case study on climate and hydrological controls, Hydrological Sciences Journal, 60, 968-984, 10.1080/02626667.2014.951361, 2015.

Mei, Y., Anagnostou, E. N., Stampoulis, D., Nikolopoulos, E. I., Borga, M., and Vegara, H. J.: Rainfall organization control on the flood response of mild-slope basins, Journal of Hydrology, 510, 565-577, https://doi.org/10.1016/j.jhydrol.2013.12.013, 2014.

Merz, R., Blöschl, G., and Parajka, J.: Spatio-temporal variability of event runoff coefficients, Journal of Hydrology, 331, 591-604, https://doi.org/10.1016/j.jhydrol.2006.06.008, 2006.

Rinaldo, A., Vogel, G. K., Rigon, R., and Rodriguez-Iturbe, I.: Can One Gauge the Shape of a Basin, Water Resour Res, 31, 1119-1127, Doi 10.1029/94wr03290, 1995.

Tarasova, L., Basso, S., Zink, M., and Merz, R.: Exploring Controls on Rainfall-Runoff Events: 1. Time Series-Based Event Separation and Temporal Dynamics of Event Runoff Response in Germany, Water Resources Research, 54, 7711-7732, 10.1029/2018WR022587, 2018.

Tarasova, L., Merz, R., Kiss, A., Basso, S., Blöschl, G., Merz, B., Viglione, A., Plötner, S., Guse, B., Schumann, A., Fischer, S., Ahrens, B., Anwar, F., Bárdossy, A., Bühler, P., Haberlandt, U., Kreibich, H., Krug, A., Lun, D., Müller-Thomy, H., Pidoto, R., Primo, C., Seidel, J., Vorogushyn, S., and Wietzke, L.: Causative classification of river flood events, WIREs Water, 6, e1353, 10.1002/wat2.1353, 2019.

Zoccatelli, D., Borga, M., Viglione, A., Chirico, G. B., and Blöschl, G.: Spatial moments of catchment rainfall: rainfall spatial organisation, basin morphology, and flood response, Hydrol. Earth Syst. Sci., 15, 3767-3783, 10.5194/hess-15-3767-2011, 2011.

**APPENDIX 1: $I_{ASYM}$, $D_{HILLS}$ and $D_{STREAM}$ for all 48 rainfall events**

[Figure]

*Figure 6. Evolution of rainfall intensity and $I_{ASYM}$ for the 48 rainfall events (P event).*

[Figure]

*Figure 7. Evolution of $D_{HILLS}$ (computed with the "wet" network) for the 48 rainfall events (P event).*

[Figure]

*Figure 8. Evolution of $D_{HILLS}$ (computed with the "dry" network) for the 48 rainfall events (P event).*

[Figure]

*Figure 9. Evolution of DSTREAM (computed with the "wet" network) for the 48 rainfall events (P event).*

[Figure]

*Figure 10. Evolution of DSTREAM (computed with the "dry" network) for the 48 rainfall events (P event).*

**APPENDIX 2: Model results for all of the 15 events**

[Figure]

Q event #1 (July 2$^{nd}$)

On each figure the Y-axis of each hydrograph is in m$^3$/s.

- The black curve is the observed streamflow.
- The 20 blue curves correspond the simulated streamflow based on the 20 possible rainfall fields from the stochastic interpolation method (12-station raingauge network).
- The plain, dashed and dotted red lines are resp. the simulated streamflow using the best 1-station (station #5), 2-station (stations #2 and #9) and 3-station (stations #2, #5 and #11) raingauge network, using the Thiessen polygons interpolation method.
- The plain, dashed and dotted purple lines are resp. the simulated streamflow using the worst 1-station (station #1), 2-station (stations #1 and #3) and 3-station (stations #1, #3 and #4) raingauge network, using the Thiessen polygons interpolation method.

[Figure]

Q event #2 (July 3$^{rd}$)

[Figure]

Q event #3 (July 5th)

[Figure]

Q event #4 (July 6th)

[Figure]

Q event #5 (July 14th)

[Figure]

Q event #6 (July 15th)

[Figure]

Q event #7 (July 20th)

[Figure]

Q event #8 (July 24th)

[Figure]

Q event #9 (August 14th)

[Figure]

Q event #10 (August 17th)

[Figure]

Q event #11 (August 23rd)

[Figure]

Q event #12 (August 24th)

[Figure]

Q event #13 (August 29th)

[Figure]

Q event #14 (September 1st)

[Figure]

Q event #15 (September 13th)

---

## Author Comment (AC4) · 17 Jan 2021

Dear Editor and Reviewers,

Thank you for your detailed comments and suggestions about our manuscript now entitled "*Even event-scale hydrological response characterization benefits from high density rain gauge observations*".
The paper has been revised accordingly. Please find hereafter the details of the changes in the form of an item-by-item response (in green) to your comments (in black). If our corrections are direct implementations of your remarks, the answer to the comment is simply 'Ok'. Please note that the line numbers indicated hereafter refer to the ones of the track-change revised manuscript attached below the answers. To avoid some confusion, letters will be used to refer to the figures from this author-reply document and the usual figure numbers will be used to refer to figures from the track-change version. To not unduly increase the length of this rebuttal, we refer at times to the public discussion.

We hope that our responses address all the raised concerns,

Best regards,

Anthony Michelon, Lionel Benoit, Harsh Beria, Natalie Ceperley, Bettina Schaefli

**Responses to the comments of Reviewer #1:**

1a - The title states "value of high density rain gauge observations for… hydrology". I'm struggling with this holistic formulation.

*The title has been changed from "On the value of high density rain gauge observations for small Alpine headwater catchment hydrology" to "Even event-scale hydrological response characterization benefits from high density rain gauge observations".*

1b - Indeed, the value is "only" (please don't get me wrong here) based on prediction of RC and deltaP/Q. While a realistic estimate of these characteristics is valuable, the uncertainties resulting from the final network with 3 rain gauges for these two criteria is not shown and should be added in a later version of the manuscript.

*We initially proposed in the public discussion to add two figures showing i) the RC (Figure A) and ii) the lag time $\Delta_{P/Q}$ (Figure B) by comparing the values obtained from the best 1-station or 3-station raingauge network vs. the reference value calculated from the full raingauge network. These two scatter plots are shown below, but we finally found that these results were more visible as a polar plots. These two plots are gathered in the Figure 12 of the manuscript in "4.4.3 Optimum network evaluation", and they show the RC and lag time $\Delta_{P/Q}$ calculated from the best 1-station and 3-station network compared to the full raingauge network (L534-543).*

*As the stochastic method for generating rainfall fields cannot be used with a number of points as low as 1 or 3 stations, we performed the computations using the Thiessen polygons methods and consequently no error bars are associated to these plots. Nevertheless, the Figure C compares the two methods (stochastic vs Thiessen polygons) when the RC and the lag time $\Delta_{P/Q}$ are computed from the full raingauge network.*

*We observe for both the RC (Figure A) and $\Delta_{P/Q}$ (Figure B) a lower dispersion of values while increasing the density of the raingauge network. With a 3-station raingauge network the error on the RC (RMSE = 0.186) drops below the error obtained by comparing 2 different interpolation methods (RMSE = 0.256), giving a good confidence to the Thiessen polygons method used for this calculation. In the same way, for $\Delta_{P/Q}$ the error with a 3-station network (RMSE = 8.12) is lower than the error obtained with the model comparison (RMSE = 13.22).*

*On the Figure C, the dispersion of $\Delta_{P/Q}$ is originally low. Even with a 1-station network the lag can be reproduced correctly for most of the events but can also be completely wrong for one of them. Outliers are still observed with the 3-station raingauge network though, even if the error gets lower (RMSE reduced from 23.18 to 8.12).*

[Figure]

*Figure A. Comparison of RC whether it is calculated from the full raingauge network or from a partial, considering the best 1-station and 3-station network. The dataset is based on the 15 rain events associated to a river reaction.*

[Figure]

*Figure B. Difference of lag time $\Delta_{P/Q}$ obtained from a partial network (1-station and 3-station network) and the full network.*

[Figure]

*Figure C. Comparison of the RC (left) and lag time $\Delta_{P/Q}$ (right) calculated using the full raingauge network, but with a different rainfall field interpolation method (Thiessen polygons vs. stochastic).*

1c - In general, I'm missing the runoff peak as important characteristic in the manuscript. Maybe the authors can involve it/comment on it why it was not considered.

This point has been clarified in the public discussion.
(https://editor.copernicus.org/index.php?_mdl=msover_md&_jrl=13&_lcm=oc108lcm109w&_acm=g et_comm_sup_file&_ms=87052&c=189590&salt=1072061062177638648) and we added the figure showing the peak flows to the Supplementary Material (Figure S5).

1d - Also, although the analysis is designed mainly for discharge estimation, results should be also interpreted in terms of rainfall (e.g. resulting areal rainfall (extremes) for different rain gauge network densities, spatial rainfall characteristics…).

Accordingly to our answer in the public discussion, the impact of the raingauge density over i) the maximum rainfall intensities and ii) the number of misestimated events has been added in the new subsection "4.4.1 Raingauge density analysis" (L498-514) coming with the Figure 7 (new).

2 - Based on the comment before, the impact of the rain gauge network densities (and rain gauge locations) on the runoff is not analysed. In the additionally uploaded comment the main author states a rainfall-runoff modelling would go beyond the scope of the study. I do not agree with that and

recommend this modelling approach to analyse the impact on the resulting runoff itself instead on single runoff statistics. To attribute the spatial rainfall variability, a distributed rainfall-runoff model would be the best solution.

Accordingly to our answer in the public discussion, we added a modelling component to this paper; the model is discussed in the public discussion (https://editor.copernicus.org/index.php?_mdl=msover_md &_jrl=13&_lcm=oc108lcm109w&_acm=get_comm_sup_file&_ms=87052&c=189590&salt=1072061 06217763861 48).
Corresponding modifications of the paper are i) at the end of the introduction (L92-94), ii) presenting the model used in the method part "3.6 Rainfall-runoff model" (L335-350), iii) in the results section in "4.4.3 Optimum network evaluation" (L544-555), iv) with the Figure 15 summarizing the results of the different simulations and v) in the Supplementary Material part 1 with the Figure S9 (map of subcatchments), Figure S10 (the results of all simulations per event) and Figure S11 (the results of simulations per event, cumulated over time).

3 - Also, I was wondering why is there not a consistent number of events analysed throughout the manuscript. I understand that there are always measuring issues and maybe some observations are questionable, but then please remove them at the beginning. There could be one number of rainfall events considered and one subset of them for discharge analysis, but at the current state results from different subsections cannot be compared with each other due to the different populations of considered events.

The different subsets used through this study are visually clarified by the table 2 (new) and introduced in the main text when discussing of the rainfall events subsets are brought up for the first time in "4.1.1 Areal rainfall and asymmetry" and in the same way with streamflow events in "4.2.1 Observed streamflow events". This table is also referred to several times later throughout the paper when using different subsets.

4 - L25-27 It should be mentioned here again that this issue is related to mountainous areas and is not a problem in general.

Ok (L30).

5 - Fig. 2 I don't see the additional worth of showing Fig. 2 and recommend to leave it out, especially since it is included in the supplement as Fig S2 as well.

The figure showing a picture of the hydrological station (previously Figure 2) has been removed from the paper. The Figure S2 in the Supplementary Material part 1 fulfills the aim of illustrating the measurement site.

6 - L90 "average elevation" Please change to mean or median, depending on how you determined the "average" value.

Ok. The misuse of "average" has been changed to "mean" when needed throughout the whole paper.

7 - L117-118 The construction of the rating curve is not interesting for the manuscript and can be left out, also the elements regarding its construction in the supplement.

The details about the rating curve have been moved from the main text to the caption of Figure S2 and Figure S3 in the Supplementary Material part 1.

8a - L154-155 The term interpolation is not suitable in my opinion due to the rainfall generation mechanisms behind. I suggest "areal rainfall is generated after Benoit et al. (2018a) by constraining actual observations at rain gauge locations". The authors should give a less brief explanation, since in the cited manuscript different versions are applied for rainfall generation (three versions due to different covariance models) and it remains unclear for the reader, which model is used for the current study.

The explanation of the rainfall generation has been revised and extended to make the gridding process clearer (L170-177). The reference to 'interpolation' is kept for the general gridding process only but has been removed from the description of the stochastic model. In practice, the stochastic model used in the present study corresponds to the model version C in Benoit et al. 2018a because it is the best suited version for high resolution data. This is now clearly specified in the revised manuscript. In addition, we provide an open-source MatLab implementation of the model to help interested readers better grasp each step of the gridding process.

8b - Why did the authors choose this rainfall generation instead of a regionalization approach as kriging (maybe with altitude as additional information), inverse distance weighting or Thiessen polygons. The latter is chosen later in the manuscript nevertheless due to computational efforts, so why not for the whole study? Was it the authors intention to add an uncertainty analysis.

Indeed, we choose the stochastic rainfall generation for the valuable estimation of the errors it provides. The Thiessen method also used throughout this paper fills the weak points of the stochastic approach, namely i) the computation time, which is very short using the Thiessen method and allows to explore within a reasonable amount of time all the possible combinations of raingauge networks for their optimization, and ii) to calculate rainfall fields even with a low number of raingauges. For this last point the stochastic method require at least 5 stations to capture correctly the spatial and temporal rainfall characteristics.

Concerning the altitude effect on rainfall, we do not observe any trend ($R^2 = 0.06$) between the cumulated rainfall per station and the altitude. This information has been added (L356-357) in "4.1. Areal rainfall and asymmetry". The Figure below is shown as illustration purpose for this document

[Figure]

*Figure D. Cumulated rainfall at each raingauge vs. altitude of each raingauge.*

9 - L154-163 The authors should bring this argument in context with the catchment concentration time.

We clarify the justification about inter-event time chosen to separate 2 consecutive events, introducing the catchment's response time and the recent paper of Beven (2020) that extensively clarify this concept. This justification reads as: "Accordingly, we assume that this event gives a rough estimate of the catchment's response time (Beven, 2020) i.e. of the time required until the entire catchment contributes to the streamflow response, including the delay caused by runoff transfer to the stream network and from there to the outlet from the hydrologically most distant parts of the catchment. The 90 minutes were therefore selected to maximize the chances of observing a distinct streamflow response for two distinct consecutive rainfall events." (L183-186)

10 - L165-166 The location of the line chosen for the splitting of the catchment seems to be chosen arbitrary. Would a line constructed perpendicular to the main flow direction of the river (or even better, not a straight line but following the lines perpendicular to the isohypses to separate flows exactly) lead to more representative results, since the catchment is then split into a real upper and lower part? Or (thinking the other way around) does it not matter at all and the splitting line could be also drawn from South to North as long as both parts have the same area?

The possible geometries of the splitting line used to compute $I_{ASYM}$ is now discussed in "5.1 Spatial heterogeneity of rainfall". This justification reads as: "It is noteworthy that this analysis could be affined by investigating different splitting geometries, e.g. by splitting the catchment into west and east parts, thereby separating the large slopes (west) from the steep slopes (east).This and similar spatial asymmetry metrics are case-specific as they rely on the particular geomorphology and topography of the catchment and are thus not directly applicable to other catchments. In particular $I_{ASYM}$ cannot be used as a tool to compare different catchments." (L570-573)

Furthermore, we also justify the use of the current north/south splitting line when introducing it for the first time in "3.2.2 Spatial rainfall pattern metrics", discussing the Strahler stream order. This justification reads as "This heuristic splitting into two parts is interesting here due to i) the elongated catchment shape and furthermore ii) the clearly distinct stream network organisation in the upper (southern) part of the catchment with more branching than in the northern part (reflected in the Strahler stream order that does not further increase in the norther part, see Figure 1). Accordingly, we assume the rainfall events falling exclusively on one or the other part of the catchment lead to a distinct streamflow response, with a faster and stronger response for events falling on the northern part (closer to outlet, steeper hillslopes, less storage potential than for the southern part)." (L196-201)

11 - L211-215 I suggest to move this paragraph to the beginning of section 3.3.2

This comment refers to the statement on how baseflow is separated ("The beginning and the end of the streamflow response determine the initial and final baseflow, respectively; the streamflow volume above the line connecting these two points is considered here as fast runoff.") and we think that it is an integral part of event identification. However, we changed the subsection title "3.3.1 Event identification" to "3.3.1 Identification of streamflow events and fast runoff".

12 - L217-218 The authors declare volume and lag time as "the two key characteristics of streamflow reaction". I do not agree with that. The most important characteristic is peak flow, followed by volume and then lag time and flatting behaviour. Even if all characteristics are considered equal important, the authors should state why peak is not considered in the study. If there were attempts to include peaks which did not work, the authors should state so as "lessons learned" in the manuscript.

L234-239: we corrected the beginning of section 3.3.2. It now read as: "The key metrics to characterize the streamflow response are the peak flow, the fast streamflow volume, the lag time elapsed between rainfall and streamflow response, and the flatting behaviour. For technical reasons we discarded the peak flow (see section 3.3.1**Erreur ! Source du renvoi introuvable.**) and consequently the flatting behaviour. We use the fast streamflow volume through the runoff coefficient (RC), which is obtained by dividing the fast runoff volume by the total rainfall for the given event. The lag time […]" (L262-266).

And at the end of the section 3.3.1: "It is noteworthy that we do not use peak streamflow to characterize streamflow events, for two reasons: i) given the small size of the catchment and the complex temporal distribution of rain intensities, the streamflow response has rarely a single, well identifiable peak (all events are plotted in Figure S5 in Supplementary Material Part 1); ii) peak streamflow identification is further complicated by the noise in the stage recordings."

13 - L219-221 Is this criterion developed by the authors or should a reference be cited in this context? How was 1/3 chosen as threshold? This value should be catchment-dependent in my opinion, or not? Please clarify.

L241-244: we clarified how the 1/3 threshold on rainfall and streamflow reaction was selected. It now reads as: "Since the start of excess rainfall is not known, the concept of peak flow is difficult to apply to our observed events (Section 3.3.1) and given the varying shape of our hydrographs, we empirically tested different lag formulations; the lag between 1/3 of the rainfall event volume and 1/3 of the streamflow event volume gives the best results in the regression analysis, and is therefore retained. It is noted $\Delta_{P/Q}$ in the following." (L271-275).

14 - L222 Why is this criterion "1/3 of the rainfall amount" more robust than "start of the rainfall event", although both starting points are linear correlated?

The formulation was not well chosen and has been reformulated along with the sentence of the previous point (L271-275).

15 - L275 Same differences lead to higher asymmetry values for smaller values. To avoid a misinterpretation ("Interestingly…") Pnorth and Psouth could be normalized by the mean event rainfall amount. This would provide deeper insights, especially since larger differences between both parts cannot be seen in the current approach if they occur for events with high rainfall amounts.

We added the columns $P_{NORTH}/P_{ALL}$ and $P_{SOUTH}/P_{ALL}$ in the Table 3.

16 - L323-327. I cannot follow the argumentation here. Please explain in detail how you achieve this conclusion and consider at least one or two sentences for each argument.

The paragraph has been reformulated. It now reads as: "The correlation analysis (Table 4) reveals a strong correlation between rainfall amounts and $Q_{FAST}$ (0.77, **Erreur ! Source du renvoi introuvable.**). This suggests that streamflow responses are triggered by saturation-excess, rather than by infiltration capacity-excess: If saturation is exceeded, every unit of rainfall leads to a corresponding unit increase of streamflow, which in turn leads to  a strong linear correlation between rainfall amounts and fast streamflow volumes. Furthermore, saturation-excess also implies that a longer rainfall event leads to a higher streamflow response volume (once the saturation threshold is reached, all rainfall contributes to streamflow). This is confirmed by the high correlation (0.74) between the rainfall duration $P_{DURATION}$ and $Q_{FAST}$. If, on the contrary, the driving process was the exceedance of the soil infiltration capacity, then only rainfall intensities above the capacity threshold would trigger a corresponding streamflow increase; small rainfall amounts would trigger almost no response. In this case (infiltration-excess), there would be no linear correlation between rainfall amounts or rainfall duration and streamflow amounts, but a strong correlation between fast streamflow amounts and high or maximum precipitation intensity; positive correlations between $Q_{FAST}$ and $P_{max\ ALL}$, $P_{max\ NORTH}$ or $P_{max\ SOUTH}$ are however all absent (values of -0.17, -0.16 and -0.08, Table 4). In addition, saturation-excess as a main driver of the fast streamflow response is further confirmed  by the clear threshold effect for the generation of streamflow as a function of total event rainfall (**Erreur ! Source du renvoi introuvable.**); a streamflow response only occurs for total rainfall higher than 5 mm." (L429-443)

17 - L330 "to reach a higher "RC" Please rephrase, the manuscript is about observations, not modelling.

We rephrased "to reach a higher RC, we need a higher level of saturation [...]" by "we observe a higher RC when the level of saturation increases [...]" (L447).

18 - L341 composites: If there is a differentiation into wet and dry state, how do the authors achieve only one value for each criterion? Are two values estimated (for wet and dry) and then the arithmetic mean is mentioned? Please clarify!

First, we changed the name "composite network" to "pseudo-dynamic network" in the entire document. A column has been added to Table 3 to show which network (dry/wet) is used for each of the 15 rainfall events having a streamflow reaction, and the missing explanation of the network extent and pseudo-dynamic network calculation is now explained in "3.4.1 Pseudo-dynamic stream network extent". It reads as:

"In absence of exact observations of the stream network extent before the start of each streamflow event, we propose here to use a pseudo-dynamic stream network extent which assigns the dry or the wet network to each streamflow. The network state is chosen based on a measure of the initial catchment wetness conditions." (L284-285)

"This correlation analysis yields an optimum antecedent wetness indicator corresponding to the rainfall over the 3 days preceding the start of a rainfall event, noted $W_{3days}$. Using this indicator, the pseudo-dynamic network extent is obtained by assigning the dry network state to rainfall events that have $W_{3days}$ < 20 mm and the wet network state to rainfall events that show $W_{3days} \geq 20$ mm. This threshold of 20 mm is selected by maximizing the correlation coefficient between $D_{HILLS}$ and RC (see Section **Erreur ! Source du renvoi introuvable.**)." (L95-299).

19 - L351-355 It would be nice to have a table with all criteria, where it is stated which one was removed (and why) and which ones were kept. Maybe the information can be added to Table 5 or 6?!

We discuss in this section ("4.3 Identification of dominant hydrologic drivers via regression analysis") only the best models. Among all the models tested through this regression analysis (combining models having one or two explanatory variables), the selection is exclusively based on AICc ranking and R². The rejection of the models having a lower rank is therefore not detailed in the text or in the tables. We made the model selection method clearer in the text and it now reads as: "The tested models, based on one or two explanatory variables, are summarized in **Erreur ! Source du renvoi introuvable.** for RC and in **Erreur ! Source du renvoi introuvable.** for $\Delta_{P/Q}$. The analysis is based on 14 events (after removing the 24 July event, subset #4 of Table 2) and the best models are selected based on their AICc ranking and coefficient of determination (R²)." (L480-482)

20 - L354 Again, it feels as the number of considered events changes among all subsections.

This has been clarified (see also point 3 above).

21 - L380 What is the reason for IASYM preference in the Southern part? Due to the steeper areas? I would have estimated Northern part, since the hydrograph would have already been smoothed when originated in the South. Please try to find physical explanations to your results.

Thanks for pointing this out. Due to a legacy effect the sentence "And for a single station network, the metric $I_{ASYM}$ prefers a station location in the southern part rather than in the northern part" is false and has been removed, as well as the plot of the best 1-station network for $I_{ASYM}$ in the figure 12.

22 - General: Please double-check the abbreviation for "meter above sea level"; I have only seen "m a.s.l." and "m asl" so far, but not "m asl."

The abbreviations of meter above sea level have been corrected from "m asl." to "m asl" throughout the whole paper.

23 - L155 Benoit et al. 2018 <- a or b? I assume a.

Ok (it is "2018a" indeed).

24 - Eq 2, 3, 4 I'm a bit confused what rainfall characteristic is used as input for these equations. Is every raster cell with rainfall used (so I understood it from the text) or only the centre of the rainfall events (as mentioned in Table 1)?

The rainfall characteristics and space-time resolution used into the Equations 2, 3 and 4 have been clarified. The text now reads as "[…] where $i$ and $j$ are the coordinates of rainfall location within the grid, $P(i,j,t)$ is the rainfall amount previously calculated using the stochastic method (section 3.2.1) for each of the 10 x 10 meters grid cell at each 2-minute time step $t$, and $d_{HILLS}(i,j)$ is the distance of this grid cell to the nearest stream network grid cell (following the line of steepest descent in the 2 x 2 m DEM (swissALTI3D, 2012))." (L217-220)

25 - L163 "overlooked" -> ignored

Ok.

26 - Eq. 2, 3, 4 The term in the numerator should be put in brackets (Eq. 2: "P(..)dHills" -> "(P(..)dHills)")

The equations formulation (missing brackets) have been corrected for $D_{HILLS}$, $D_{STREAM}$ and $H_{HAND}$ (Equations 2, 3 and 4, respectively).

27 - L195 DHAND is not a distance as indicated by the D, and in the text the variable is introduced with HAND. I suggest to stick to HAND throughout the manuscript to avoid confusions with the other two "real" distances".

We now use the abbreviation $H_{HAND}$ instead of $D_{HAND}$ (and not $HAND$ like we initially proposed, to avoid variable names made from several letters) throughout the whole paper, figures and associated documents.

28 - L202 Section 3.5 includes no network extent description. Is it missing in the manuscript?

The missing explanation of the network extent is now explained in "3.4.1 Pseudo-dynamic stream network extent" (L278-299).

29 - L268 317.8 mm – Is it areal rainfall amount sum or sum over all stations?

We specified (L355) that the value of 317.8 mm is the areal rainfall amount.

30 - L268-269 please provide also the mean values, not only the highest and lowest values, so that the reader get a "feeling" for the rainfall events.

The mean values have been added (L356).

31 - L275 again, please don't use the term average, use mean or median to be more concise. Since Iasym can be positive and negative, the median of its absolute values would be worth to show instead of just the mean, since positive and negative values are levelling out each other.

Indeed, in this case using the median value of $I_{ASYM}$ (0.025) is better than using the mean, it has been corrected (L363-364). Also, the misuses of "average" has been corrected when needed throughout the whole paper.

32 - Fig. 5 and 6 For a logical order the figures should show the rainfall events first, followed by the discharge plot.

The figures showing rainfall events records (Figure 3 and Figure 4 in the main text, and all the figures of the Supplementary Material part 2) have been rearranged to have rainfall data above streamflow data.

33 - L279 "One strongly asymmetric and high intensity event" -> "One strong asymmetric and very intense event"

Ok.

34 - L283 A volume can't be fast (check also for later occurrences…)

Ok.

35 - L288 In the sentence before authors mention that the number of events under consideration are reduced by "1", but here again 48 events are studied (also in the following subsections).

Indeed, it is confusing. The line it is referred to "This event and its streamflow reaction are excluded from further analysis" has been replaced by "This event and its streamflow reaction are excluded from further analysis involving the hydrological response" (L373-374).

36 - L289 The authors should state what wet and dry networks are. I found it later in the caption of Table 1 in S1, but it would lead to clarifications here. Also, the Table 1 in S1 should be shown in the manuscript, since the written part in Section 4.1.2 is more confusing than explaining for me.

The dry and wet networks are now introduced in "2 Study area". It now reads as: "The actual extent of the stream network is based on observations during Summer 2017 (dry and wet periods) and its exact path was calculated using the Swiss digital elevation model at a resolution of 2 m (swissALTI3D, 2012)." (L126-127)

It is detailed later on in "3.4.1 Pseudo-dynamic stream network extent" and it reads as "The extent of the stream network evolves as a function of the catchment wetness conditions. Its minimal and maximal extent (**Erreur ! Source du renvoi introuvable.**) are determined manually by identifying the uppermost points of the catchment where streamflow was observed in the field during summer baseflow (minimum extent, called *dry* state) and during summer high flow (maximum extent, called *wet* state)."

37a - Fig9 "events without reaction are not shown" belongs to part b), not a). Please correct the caption.

Ok.

37b - General: Maybe I missed it, but which temporal resolution was used to calculate the correlation (and other criteria)? 2min as this is the resolution of the rain gauge? Or are values aggregated up to e.g. 1h? This has a high impact on the values of the correlation coefficient.

The temporal resolution of times series used for correlation calculations is 2 minutes. The correlation between events is done at the event-scale. This is now clarified in the text.

38 - L339 "absence of correlation". Correlation cannot be absent. Better to speak of low correlation or provide absolute values.

Ok.

39 - L384-386 "is assessed", "is evaluated" – two verbs, please rephrase the sentence.

Ok. It now reads as "Considering the small dataset underlying this analysis (23 events), the robustness of the best networks is assessed for two selected metrics (for the $P_{ALL}$ and $I_{ASYM}$) by re-computing the optimal network if between 1 and 3 events are removed from the dataset." (L329-331)

40 - L402 "what we previously thought"? What was the hypothesis of the authors before?

We reformulated the explanation about the outperformance of $D_{STREAM}$ over $D_{HILLS}$ for the prediction of RC and lag time. It now reads as: "We could expect that in that kind of steep environments, the residence time in hillslopes strongly dominates over residence times in the stream network (Nicotina et al., 2008); the fact that $D_{STREAM}$ outperforms here $D_{HILLS}$ for the prediction of RC and lag time may show that even in steep environments, with a priori fast instream processes and limited storage, the riparian area and related subsurface exchange processes could play a more prominent role. The fact that the travel distance in the stream network explains more of the RC variation than $D_{HILLS}$ might be an indirect effect: the longer the travel distance in the stream network, the more likely are delays due to exchange with groundwater in the riparian area." (L578-584)

41 - L421 "three station network" It would be nice to provide the resulting density here as well as "(general) recommendation".

The results are now also presented in term of raingauge density in the figure 7 and in the text referring to (L505-506) and later in the text L536-537.

**Responses to the comments of Reviewer #2:**

1) I already reviewed the initial version of this paper, which aims at highlighting the values of high density rain gauges networks for hydrological purposes in a small catchment of mountainous areas. I still believe that the topic is interesting and relevant for the community. It furthermore has other potential applications in urban areas which are also small and quickly reactive catchments where rainfall variability has strong consequences.

A reference to urban hydrology (Cristiano et al., 2017) has been added into the introduction. It now reads as "While our analysis focuses here on a small natural headwater catchment, it is noteworthy that the developed rainfall monitoring and data analysis framework might also be of interest for urban hydrology, which deals with similar questions regarding how spatial rainfall patterns, runoff generation processes and flow network geometry lead to peak flows in urban drainage systems (for a review, see the work of Cristiano et al., 2017)." (L77-81)

2) The minor difficulties with regards to the presentation and understanding of the paper have been corrected. Results are now better presented with the new figures. However, the main point was not addressed, i.e. the fact that the authors aims at showing the importance of grasping the spatio-temporal variability of the rainfall process in the prediction of flows, but the chosen indicators are only event based averages.

First of all, we would like to point out that we accidentally used the formulation "runoff prediction" rather than "runoff coefficient prediction" in the abstract. We removed the misleading formulation "runoff prediction" from the abstract, that has also been adapted to the changes made to the paper. This having said, we did not mean to pretend that we study the predictability of streamflow but well the relation between rainfall field characteristics and runoff event characteristics, which has indeed a long tradition in hydrology as a basis for model development and comparative hydrology (Merz et al., 2006).

We believe that a focus on the scale of runoff response events is fully justified. We use for our event-based analysis descriptors of two fundamental properties of runoff events, which refer to the time until a response occurs and the magnitude of the response; we choose here the runoff coefficient and a lag time.

We agree that these average streamflow event properties hide other interesting aspects of the hydrological response namely referring to the shape of the response and including e.g. the occurrence of double-peak response. A further detailed analysis of such double-peaked events versus single peaked events is however not possible for this small data set (see Figure showing all streamflow events at the end of this document).

One additional descriptor of hydrograph shape could be what Tarasova et al. (2018) call the runoff event time scale, i.e. the ratio of runoff volume in mm and the runoff event peak flow in mm/day (see also Gaál et al., 2015). This descriptor could potentially shed light on the mixture of fast and slow runoff response processes. The identification of peak runoff is however extremely challenging for this case study because the moment of peak flow occurrence is often not well defined (see **Erreur ! Source du renvoi introuvable.** and our response to the point 8 of this document).

Another descriptor for this mixture could be the rising time of an event, quantifying the time to peak (normalized by the event duration). We refrained however from using descriptors requiring a precise quantification of event duration since the identification of an exact start and end time remains extremely challenging (Tarasova et al., 2018) and could largely affect the results in presence of a relatively small data set.

3) Furthermore, the main rainfall variability (which is at the core of the paper) indicator used is too simplistic since it is basically an asymmetry indicator on the total depth splitting the catchment in two. So I still think that indicators actually accounting for the spatiotemporal variability of the rainfall and hydrologic response should be implemented to actually address the stated topic of the paper. Implementing them requires major modifications of the paper. I guess that this would enable to highlight more precisely the importance of dense networks of rainfall measurement devices.

We agree that $I_{ASYM}$ is a simple indicator to capture the key rainfall field properties for the hydrological response. In other studies and namely in urban hydrology such an indicator is typically based e.g. on the variogram or on the spatial moments of rainfall with continuously observed rainfall fields (radar images). We added this comment in "3.2.2 Spatial rainfall pattern metrics" and it reads as: "Spatial rainfall patterns are classically characterized with geostatistical tools, including variograms (Berne et al., 2004) or with spatial moments of rainfall (Smith et al., 2002;Zoccatelli et al., 2011;Mei et al., 2014), in particular in presence of observed rainfall fields, e.g. from radar images. Here we propose to use more hydrological-process oriented metrics that explicitly account for known features of the catchment and the stream network." (L190-194)

The asymmetry indicator is just one of the indicators used in the study, along with the geomorphological distances, which corresponds to the above first order spatial moments, albeit decomposed according to hillslope and stream network flow distances. As we answered in the public discussion, we tried the second order moment of distance metrics, but it does not show any noticeable correlation with a rainfall or streamflow metric. We added this result in the text: "It is noteworthy that these two metrics, $D_{HILLS}$ and $D_{STREAM}$ correspond to the aforementioned first order spatial rainfall moments, albeit decomposed according to hillslope and stream network distances, similar to what was proposed by Zoccatelli et al., 2015 in their analytical framework to quantify the smoothing of spatial rainfall organisation effects by channel residence time. It would be tempting to use also higher order rainfall moments; however, no significant correlation could be found the retained streamflow metrics." (L229-233)

Finally, we also added the section "4.1.3 Temporal evolution of rainfall metrics", please see our answer to the point 7 below.

4) l. 110 – 115: "The actual extent of the stream network is based on observations during dry and wet periods during Summer 2017 and its exact path was calculated using the Swiss digital elevation model at a resolution of 2 m (swissALTI3D, 2012)." I think that the intrinsic fractal nature of river networks should be mentioned and discussed. The concept of variable network used after also seems interesting.

We implemented at the end of "5.1 Spatial heterogeneity of rainfall" a comment about the fractality of the river network. It reads as: "However, future work on the role of water residence time in the stream network will necessarily require more detailed field data on the temporal evolution of the stream network. This will in addition open new perspectives to quantify how the stream network extension is imprinted in the streamflow response: in fact, as discussed by Rinaldo et al. (1995), the intrinsic fractal nature of the stream network is not transferred to the streamflow response and, accordingly, there is potential to infer the stream network extension from observed streamflow records, provided that we have high resolution rainfall data to disentangle the different effects." (L587-592)

5) l. 150-151: "Some additional artefacts were recorded, probably generated by strong winds creating resonance. These periods have been manually removed from the data". It should clarified how the data was selected for being removed and what portion was removed.

We clarified how raw rainfall data were selected and some parts removed. It now reads as: "Additional artefacts were recorded, probably generated by strong winds creating resonance. Some stations in fact recorded very strong and highly variable rainfall over several hours during periods with high wind velocity but during days without any observed rainfall in the combined MeteoSwiss radar-rain gauge

data (Sideris et al., 2014). Four periods (over 4 different days) have been manually removed from the data." (L164-167)

6) 1. 154-157: It is a great improvement to use this stochastic procedure. Nevertheless, I believe that more details on the interpolation procedure are needed. It should be clarified how the 20 samples are used (computing the error bars in 8-10)?

We added details about the stochastic procedure and error bar computation. It now reads as: "Before further analysis, the rainfall amounts measured by each station were interpolated to a 10 by 10 m grid at a 2 min time step using a high-resolution stochastic approach developed by Benoit et al. (2018a). In a nutshell, it generates an ensemble of stochastic space-time rain fields constrained by the actual observations at the rain gauge locations. The resulting ensemble (here composed of 20 realizations) can be used to analyze spatial rainfall uncertainty or to construct a single rainfall estimator. Following Benoit et al. (2018a), a non-separable and asymmetric covariance function was used to perform the simulations, which allows modelling rainfall advection and diffusion observed in the raw data. Areal rainfall time series are calculated for each of the 20 realization, and from these a single time series (mean and standard deviation) of the areal rainfall." (L170-177)

7) Eq. 1 on I_ASYM. As already mentioned, it seems a too simplistic indicator to grasp spatio-temporal variability of the rainfall process. An initial simple suggestion could for instance be to look for the temporal evolution of I_ASYM during an event. But other indicators are needed

We added figures showing the evolution of $D_{STREAM}$, $D_{HILLS}$ and $I_{ASYM}$ in the supplementary Material, and added in the text a qualitative discussion of the temporal evolution of the rainfall metrics in "4.1.3 Temporal evolution of rainfall metrics". It reads as:

"We computed the temporal evolution of the rainfall metrics to unravel potential temporal evolution patterns in $I_{ASYM}$, $D_{HILLS}$ and $D_{STREAM}$ and their relation to the streamflow response (full results are available in the Supplementary Material part 1). The temporal evolution of the two distance metrics is overall rather flat with no clear fluctuation patterns. There is only one event with a pronounced temporal trend for $D_{HILLS}$ (Q event #1).

For $I_{ASYM}$, some events show interesting temporal patterns. For example, during the double peak runoff of **Erreur ! Source du renvoi introuvable.**, $I_{ASYM}$ shows an almost constant negative value suggesting that the corresponding double peak rainfall event remained stationary on the northern part of the catchment over its entire duration and therefore caused the double peak streamflow response.

For the first two streamflow events, the $I_{ASYM}$ metric switches from strongly positive to close to zero during the event, implying that the rainfall field moved towards the outlet during the event; in other words, the rainfall cloud follows the overall water movement through the catchment and thereby leads to a stream response concentration. This might explain why these two events are the only ones that show a pronounced single peak streamflow response. However, given the low number of observed events and the diversity of temporal patterns, these insights cannot be further used for a quantitative analysis." (L391-404)

8a) l. 212-215: the explanation on why not using streamflow variations (notably peak flow) is not very convincing.

This point has been clarified in the public discussion. (https://editor.copernicus.org/index.php?_mdl=msover_md&_jrl=13&_lcm=oc108lcm109w&_acm=get_comm_sup_file&_ms=87052&c=189590&salt=10720610621776386148) and we added the figure showing the peak flows to the Supplementary Material (Figure S5).

8b) If the purpose is to investigate the importance of spatiotemporal variability, I guess studying the temporal variability of the simulated streamflow is needed.

We decided to add a modelling component to this paper; the model is discussed in the public discussion (https://editor.copernicus.org/index.php?_mdl=msover_md&_jrl=13&_lcm=oc108lcm109w&_acm=get_comm_sup_file&_ms=87052&c=189590&salt=10720610621776386148).

Corresponding modifications of the paper are i) at the end of the introduction (L92-94), ii) presenting the model used in the method part "3.6 Rainfall-runoff model" (L335-350), iii) in the results section in "4.4.3 Optimum network evaluation" (L544-555), iv) with the Figure 15 summarizing the results of the different simulations and v) in the Supplementary Material part 1 with the Figure S9 (map of subcatchments), Figure S10 (the results of all simulations per event) and Figure S11 (the results of simulations per event, cumulated over time.

[revised manuscript text omitted]

**3.3.1    Identification of streamflow events and fast runoff**

> **Commenté [AM16]:** Renamed as stated in the answer to Reviewer #1 – Point 11

The beginning and the end of each streamflow event are identified manually using a data visualization tool (developed in

250 MathWorks MatLab 2017a, see Figure 3 and Figure 4). This choice of a visual expertise was made based on the observation that automatic identification of streamflow events would require almost a case-by-case filtering and parametrization, and thus would not be generalizable. This is partly related to a potentially high signal-to-noise ratio for river stage recordings during sediment transport events, a phenomenon potentially very important after a strong streamflow variation. The result of this visual identification for each streamflow event is displayed in the part 2 of Supplementary Material.

255 The beginning and the end of the streamflow response determine the initial and final baseflow; the streamflow volume above the line connecting these two points is considered here as fast runoff. It is noteworthy that we do not use peak streamflow to characterize streamflow events, for two reasons: i) given the small size of the catchment and the complex temporal distribution of rain intensities, the streamflow response has rarely a single, well identifiable peak (all events are plotted in Figure S5 in Supplementary Material Part 1); ii) peak streamflow identification is further complicated by the noise
260 in the stage recordings.

**Commenté [AM17]:** Modified according to Reviewer #1, point 12

**3.3.2 Streamflow metrics**

The key metrics to characterize the streamflow response are the peak flow, the fast streamflow volume, the lag time elapsed between rainfall and streamflow response, and the flatting behaviour. For technical reasons we discarded the peak flow (see section 3.3.1) and consequently the flatting behaviour. We use the
265 fast streamflow volume through the runoff coefficient (RC), which is obtained by dividing the fast runoff volume by the total rainfall for the given event.  The lag time  is usually defined as the
270  elapsed time between the start of  excess rainfall (the part of rainfall that causes the streamflow response) and the peak flow (McCuen, 2009). Since the start of excess rainfall is not known, the concept of peak flow is difficult to apply to our observed events (Section 3.3.1) and given the varying shape of our hydrographs, we empirically tested different lag formulations; the lag between 1/3 of the rainfall event volume and 1/3 of the streamflow event volume gives the best results in the regression analysis, and is therefore
275 retained. It is noted $\Delta_{P/Q}$ in the following.

**Commenté [AM18]:** Modified according to Reviewer #1, point 12

**Commenté [AM19]:** Details about the 1/3 threshold added according to Reviewer #1, point 13

**3.4 Rainfall-streamflow response characterization**

**3.4.1  Pseudo-dynamic stream network extent**

The extent of the stream network evolves as a function of the catchment wetness conditions. Its minimal and maximal extent (Figure 1) are determined manually by identifying the
280 uppermost points of the  catchment where streamflow  was observed in the field during summer baseflow (minimum extent, called *dry* state) and during summer high flow (maximum extent, called *wet* state).
In absence of exact observations of the stream network extent before the start of each streamflow event, we propose here to
285 use a pseudo-dynamic stream network extent which assigns the dry or the wet to each streamflow. The network state is chosen

**Commenté [AM20]:** Added according to Reviewer #1 – point 36

**Commenté [AM21]:** Added according to Reviewer #1 – point 18

based on a measure of the initial catchment wetness conditions, which is known to be the major variable explaining the dynamics of the hydrological response to different rainfall events (Penna et al., 2011;Rodriguez-Blanco et al., 2012), in particular through the creation of runoff thresholds (Zehe et al., 2005;Tromp-van Meerveld and McDonnell, 2006). Many studies use the baseflow before the start of a streamflow event as an indicator for the antecedent wetness conditions of the catchment. For snow-influenced catchments with a highly seasonal streamflow regime, this indicator might not reflect the actual wetness conditions. Hence, we rather quantify initial wetness conditions in terms of antecedent rainfall, i.e. using the cumulative rainfall (in mm) that occurred during a period from 1 to 5 days before a given rainfall event. The actual time span is selected based on a correlation analysis between antecedent rainfall over 1 to 5 days and the retained streamflow metrics (Section 4.2.1 and following).

This correlation analysis yields an optimum antecedent wetness indicator corresponding to the rainfall over the 3 days preceding the start of a rainfall event, noted $W_{3days}$. Using this indicator, the pseudo-dynamic network extent is obtained by assigning the dry network state to rainfall events that have $W_{3days} < 20$ mm and the wet network state to rainfall events that show $W_{3days} \geq 20$ mm. This threshold of 20 mm is selected by maximizing the correlation coefficient between $D_{HILLS}$ and RC (see Section 0).

**Commenté [AM22]:** Added according to Reviewer #1 – point 18

**3.4.2 Regression analysis**

We analyze the relationships between the spatial distribution of rainfall and the hydrological response based on a correlation analysis between the spatial rainfall pattern metrics (Section 3.2.2) and the streamflow metrics (Section 3.3.2) at the event scale, followed by a regression analysis to identify the key variables that best explain the runoff coefficient, RC, and the streamflow lag time, $\Delta_{P/Q}$. All used metrics are summarized in Table 1.

**Commenté [AM23]:** Point 37b. Precise in the answer that it is at event scale only. There is no intra-event correlation.

[revised manuscript text omitted]

Commenté [AM39]: Rephrased according to the remark of Reviewer #1 – Point 39

Commenté [AM40]: Added according to Reviewer #1, point 1b

Commenté [AM41]: Added according to Reviewer #1 point 2 and Reviewer #2 point 8b

**5 Discussion**

**5.1 Spatial heterogeneity of rainfall**

One of the key identified metrics to characterize the spatial distribution of rainfall in relation to RC and lag prediction is $I_{ASYM}$ splits the catchment into two parts, and aggregates rainfall observations into two values. Among the records showing a strong rainfall asymmetry, 7 out of the 8 events are too small to cause a detectable streamflow response (Figure 5), but one does create a streamflow response although it only rains over half of the 12 rain gauge stations. Despite of this absence of a strong asymmetry in the 14 rainfall events that cause a streamflow response, the regression analysis  suggests that  the spatial distribution might play an important role for the explanation of the lag time. The importance of this asymmetry predictor can be related to the fact that it captures the key feature of the spatial catchment organisation in terms of distance to the outlet, drainage density and subsurface storage potential.

The second dominant metric of spatial rainfall distribution to predict the RC and the lag is $D_{STREAM}$ (pseudo-dynamic). This suggests that for this catchment, the rainfall distance to the outlet is the overall the dominant predictor for the analyzed streamflow response metrics.

It is noteworthy that this analysis could be affined by investigating different splitting geometries, e.g. by splitting the catchment into west and east parts, thereby separating the large slopes (west) from the steep slopes (east).This and similar spatial asymmetry metrics are case-specific as they rely on the particular geomorphology and topography of the catchment and are thus not directly applicable to other catchments. In particular $I_{ASYM}$ cannot be used as a tool to compare different catchments. The rainfall distance metrics to the stream network ($D_{HILLS}$) and along the stream network ($D_{STREAM}$) were designed here to overcome the limitations of the simple asymmetry measure. The prominent role of $D_{STREAM}$ - pseudo-dynamic to explain the lag time and RC underlines the importance of characterizing the spatial heterogeneity in terms of geomorphological distances to the actual stream network, which requires more detailed network expansion analyses in future studies.

We could expect that in that kind of steep environments, the residence time in hillslopes strongly dominates over residence times in the stream network (Nicotina et al., 2008); the fact that $D_{STREAM}$ outperforms here $D_{HILLS}$ for the prediction of RC and lag time may show that even in steep environments, with a priori fast instream processes and limited storage, the riparian area and related subsurface exchange processes could play a more prominent role . The fact that the travel distance in the stream network explains more of the RC variation than $D_{HILLS}$ might be an indirect effect: the longer the travel distance in the stream network, the more likely are delays due to exchange with groundwater in the riparian area. This implies that along-stream processes might need a better representation in rainfall-runoff models, even for small and steep catchments; to date, these processes are often ignored in rainfall-runoff hydrological models at this scale, or are represented with a simple constant velocity transport term (e.g. Schaefli et al., 2014).

However, future work on the role of water residence time in the stream network will necessarily require more detailed field data on the temporal evolution of the stream network. This will in addition open new perspectives to quantify how the stream

**Commenté [AM42]:** Discussion about the splitting line added according to Reviewer #1, point 10.

**Commenté [AM43]:** Modified according to Reviewer #1 – Point 40

[revised manuscript text omitted]

**Commenté [AM48]:** Subsets details added according to Reviewer #1, point 3

Table 3. List of recorded precipitation events with streamflow response (event series #3 of Table 2). Full details are available in the Supplementary Material.

| Date | $P_{DURATION}$ [min] | $Q_{DURATION}$ [min] | $\Delta_{P/Q}$ [min] | $P_{ALL}$ [mm] | $P_{NORTH}$ [mm] | $P_{SOUTH}$ [mm] | $P_{NORTH}/P_{ALL}$ [-] | $P_{SOUTH}/P_{ALL}$ [-] | $I_{ASYM}$ [-] | $W_{3\,days}$ [mm] | Stream network | $Q_{INIT}$ [mm] | $Q_{FAST}$ [mm] | RC [-] | $D_{HILLS}$ [m] | $D_{STREAM}$ [m] | HAND $H_{HAND}$ [m] |
|---|---|---|---|---|---|---|---|---|---|---|---|---|---|---|---|---|---|
| 2-Jul | 42 | 44 | 24 | 7.7 | 4.1 | 3.6 | 0.53 | 0.47 | -0.06 | 3.2 | dry | 7.9 | 0.9 | 0.12 | 1521 | 4008 | 611 |
| 3-Jul | 40 | 135 | 23 | 12.1 | 7.4 | 4.6 | 0.62 | 0.38 | -0.24 | 12.7 | dry | 7.5 | 8.5 | 0.71 | 1336 | 3842 | 550 |
| 5-Jul | 224 | 309 | 71 | 8.2 | 4.0 | 4.2 | 0.49 | 0.51 | 0.03 | 29.8 | wet | 6.0 | 6.0 | 0.74 | 755 | 4374 | 350 |
| 6-Jul | 478 | 587 | 65 | 20.2 | 8.6 | 11.6 | 0.43 | 0.57 | 0.15 | 40.3 | wet | 5.8 | 25.9 | 1.29 | 874 | 4450 | 355 |
| 14-Jul | 358 | 302 | 49 | 18.7 | 10.5 | 8.2 | 0.56 | 0.44 | -0.12 | 0.0 | dry | 4.5 | 12.9 | 0.69 | 1263 | 3574 | 554 |
| 15-Jul | 136 | 281 | 33 | 10.7 | 6.0 | 4.7 | 0.56 | 0.44 | -0.13 | 18.9 | dry | 5.5 | 9.5 | 0.89 | 1122 | 3377 | 528 |
| 20-Jul | 288 | 228 | 49 | 18.8 | 8.6 | 10.2 | 0.46 | 0.54 | 0.09 | 3.4 | dry | 4.8 | 14.2 | 0.76 | 1282 | 3823 | 541 |
| 24-Jul | 220 | 229 | 45 | 8.0 | 7.5 | 0.5 | 0.94 | 0.06 | 0.02 | 12.2 | dry | 3.1 | 30.4 | 3.78 | 740 | 2184 | 419 |
| 14-Aug | 204 | 152 | 47 | 11.1 | 4 | 7.1 | 0.37 | 0.64 | 0.27 | 10.2 | dry | 4.0 | 7.8 | 0.70 | 1286 | 4305 | 540 |
| 17-Aug | 152 | 109 | 38 | 11.9 | 6.2 | 5.7 | 0.52 | 0.48 | -0.04 | 17.5 | dry | 3.2 | 4.9 | 0.42 | 1122 | 3780 | 490 |
| 23-Aug | 388 | 237 | 47 | 22.1 | 8.8 | 13.3 | 0.40 | 0.60 | 0.20 | 5.4 | dry | 2.4 | 13.5 | 0.61 | 1371 | 3756 | 563 |
| 24-Aug | 158 | 107 | 40 | 8.1 | 4.4 | 3.7 | 0.54 | 0.46 | -0.08 | 29.5 | wet | 4.1 | 6.5 | 0.81 | 692 | 4114 | 320 |
| 29-Aug | 72 | 116 | 48 | 4.8 | 2.2 | 2.6 | 0.46 | 0.54 | 0.07 | 12.4 | dry | 3.0 | 2.3 | 0.48 | 1207 | 3526 | 524 |
| 01-sept | 628 | 341 | 101 | 11.4 | 4.3 | 7.2 | 0.38 | 0.63 | 0.25 | 20.4 | wet | 3.4 | 16.4 | 1.44 | 725 | 4487 | 331 |
| 13-sept | 370 | 59 | 45 | 10.9 | 7.0 | 3.8 | 0.65 | 0.35 | -0.29 | 0.0 | dry | 2.6 | 4.4 | 0.40 | 1291 | 3594 | 556 |

Cellules insérées

Commenté [AM50]: Added according to Reviewer #1 – point 18

Commenté [AM49]: Columns added according to Reviewer #1, point 15

Table 4. Correlations between rainfall  metrics and hydrologic response metrics for  ). series #4 of Table 2. Absolute values equal or higher than 0.60 are in bold.

| | $P_{ALL}$ [mm] | $P_{NORTH}$ [mm] | $P_{SOUTH}$ [mm] | $P_{max\ ALL}$ [mm.h$^{-1}$] | $P_{max\ NORTH}$ [mm.h$^{-1}$] | $P_{max\ SOUTH}$ [mm.h$^{-1}$] | $I_{ASYM}$ [-] | $W_{3\ days}$ [mm] | $Q_{INIT}$ [mm] | $Q_{FAST}$ [mm] | $P_{DURATION}$ [min] | $Q_{DURATION}$ [min] | $\Delta_{P/Q}$ [min] |
|---|---|---|---|---|---|---|---|---|---|---|---|---|---|
| $P_{ALL}$ [mm] | - | | | | | | | | | | | | |
| $P_{NORTH}$ [mm] | **0.89** | - | | | | | | | | | | | |
| $P_{SOUTH}$ [mm] | **0.94** | **0.69** | - | | | | | | | | | | |
| $P_{max\ ALL}$ [mm.h$^{-1}$] | 0.01 | 0.19 | -0.12 | - | | | | | | | | | |
| $P_{max\ NORTH}$ [mm.h$^{-1}$] | 0.09 | 0.33 | -0.11 | **0.96** | - | | | | | | | | |
| $P_{max\ SOUTH}$ [mm.h$^{-1}$] | 0.19 | 0.19 | 0.16 | **0.87** | **0.78** | - | | | | | | | |
| $I_{ASYM}$ [-] | 0.25 | -0.20 | 0.55 | -0.42 | -0.56 | -0.06 | - | | | | | | |
| $W_{3\ days}$ [mm] | -0.19 | -0.30 | -0.09 | -0.22 | -0.27 | -0.23 | 0.18 | - | | | | | |
| $Q_{INIT}$ [mm] | -0.13 | 0.00 | -0.21 | 0.52 | 0.54 | 0.27 | -0.28 | 0.26 | - | | | | |
| $Q_{FAST}$ [mm] | **0.77** | 0.58 | **0.80** | -0.17 | -0.16 | -0.08 | 0.43 | 0.33 | -0.01 | - | | | |
| $P_{DURATION}$ [min] | 0.56 | 0.38 | **0.62** | -0.59 | -0.52 | -0.48 | 0.44 | 0.14 | -0.43 | **0.74** | - | | |
| $Q_{DURATION}$ [min] | 0.56 | 0.39 | **0.61** | -0.27 | -0.27 | -0.17 | 0.42 | 0.52 | 0.11 | **0.89** | **0.64** | - | |
| $\Delta_{P/Q}$ [min] | 0.13 | -0.11 | 0.29 | **-0.71** | **-0.71** | -0.58 | 0.59 | 0.41 | -0.33 | 0.52 | **0.81** | **0.60** | - |
| RC [-] | 0.31 | 0.13 | 0.40 | -0.25 | -0.29 | -0.22 | 0.44 | **0.65** | -0.05 | **0.81** | **0.67** | **0.80** | **0.72** |

Table 5. Correlations between distance metrics for rainfall events with streamflow response (series #4 of Table 2). Absolute values equal or higher than 0.60 are in bold. Correlations for all rainfall events are available in the Supplementary Material.

[revised manuscript text omitted]

---

## Author Response (AR1)

Dear Editor and Reviewers,

Thank you for your detailed comments and suggestions about our manuscript now entitled "*Even event-scale hydrological response characterization benefits from high density rain gauge observations*". The paper has been revised accordingly. Please find hereafter the details of the changes in the form of an item-by-item response (in green) to your comments (in black). If our corrections are direct implementations of your remarks, the answer to the comment is simply 'Ok'. Please note that the line numbers indicated hereafter refer to the ones of the track-change revised manuscript attached below the answers. The not unduly increase the length of this rebuttal, we refer at times to the public discussion.

We hope that our responses address all the raised concerns,

Best regards,

Anthony Michelon, Lionel Benoit, Harsh Beria, Natalie Ceperley, Bettina Schaefli

**Responses to the comments of Reviewer #1:**

1a - The title states "value of high density rain gauge observations for... hydrology". I'm struggling with this holistic formulation.

The title has been changed from "On the value of high density rain gauge observations for small Alpine headwater catchment hydrology" to "Even event-scale hydrological response characterization benefits from high density rain gauge observations".

1b - Indeed, the value is "only" (please don't get me wrong here) based on prediction of RC and deltaP/Q. While a realistic estimate of these characteristics is valuable, the uncertainties resulting from the final network with 3 rain gauges for these two criteria is not shown and should be added in a later version of the manuscript.

Thanks for this important suggestion. To go further with the uncertainties resulting from the best 3raingauge network we added and discuss in "4.4.3 Optimum network evaluation" the Figure 14 (new) showing the RC and lag time  $\Delta_{P/Q}$  calculated from the best 1-station and 3-station network compared to the full raingauge network (L534-543).

1c - In general, I'm missing the runoff peak as important characteristic in the manuscript. Maybe the authors can involve it/comment on it why it was not considered.

This point has been clarified in the public discussion.

(https://editor.copernicus.org/index.php?\_mdl=msover\_md&\_jrl=13&\_lcm=oc108lcm109w&\_acm=g et\_comm\_sup\_file&\_ms=87052&c=189590&salt=10720610621776386148) and we added the figure showing the peak flows to the Supplementary Material (Figure S5).

1d - Also, although the analysis is designed mainly for discharge estimation, results should be also interpreted in terms of rainfall (e.g. resulting areal rainfall (extremes) for different rain gauge network densities, spatial rainfall characteristics...).

The impact of the raingauge density over i) the maximum rainfall intensities and ii) the number of misestimated events has been added in the new subsection "4.4.1 Raingauge density analysis" (L498-514) coming with the Figure 7 (new).

2 - Based on the comment before, the impact of the rain gauge network densities (and rain gauge locations) on the runoff is not analysed. In the additionally uploaded comment the main author states a rainfall-runoff modelling would go beyond the scope of the study. I do not agree with that and recommend this modelling approach to analyse the impact on the resulting runoff itself instead on single runoff statistics. To attribute the spatial rainfall variability, a distributed rainfall-runoff model would be the best solution.

We decided to add a modelling component to this paper; the model is discussed in the public discussion (https://editor.copernicus.org/index.php?\_mdl=msover\_md&\_jrl=13&\_lcm=oc108lcm109w&\_acm=g et comm sup file& ms=87052&c=189590&salt=10720610621776386148).

Corresponding modifications of the paper are i) at the end of the introduction (L92-94), ii) presenting the model used in the method part "3.6 Rainfall-runoff model" (L335-350), iii) in the results section in "4.4.3 Optimum network evaluation" (L544-555), iv) with the Figure 15 summarizing the results of the different simulations and v) in the Supplementary Material part 1 with the Figure S9 (map of subcatchments), Figure S10 (the results of all simulations per event) and Figure S11 (the results of simulations per event, cumulated over time).

3 - Also, I was wondering why is there not a consistent number of events analysed throughout the manuscript. I understand that there are always measuring issues and maybe some observations are questionable, but then please remove them at the beginning. There could be one number of rainfall

events considered and one subset of them for discharge analysis, but at the current state results from different subsections cannot be compared with each other due to the different populations of considered events.

The different subsets used through this study are visually clarified by the table 2 (new) and introduced in the main text when discussing of the rainfall events subsets are brought up for the first time in "4.1.1 Areal rainfall and asymmetry" and in the same way with streamflow events in "4.2.1 Observed streamflow events". This table is also referred to several times later throughout the paper when using different subsets.

4 - L25-27 It should be mentioned here again that this issue is related to mountainous areas and is not a problem in general.

Ok (L30).

5 - Fig. 2 I don't see the additional worth of showing Fig. 2 and recommend to leave it out, especially since it is included in the supplement as Fig S2 as well.

The figure showing a picture of the hydrological station (previously Figure 2) has been removed from the paper. The Figure S2 in the Supplementary Material part 1 fulfills the aim of illustrating the measurement site.

6 - L90 "average elevation" Please change to mean or median, depending on how you determined the "average" value.

Ok. The misuse of "average" has been changed to "mean" when needed throughout the whole paper.

7 - L117-118 The construction of the rating curve is not interesting for the manuscript and can be left out, also the elements regarding its construction in the supplement.

The details about the rating curve have been moved from the main text to the caption of Figure S2 and Figure S3 in the Supplementary Material part 1.

8a - L154-155 The term interpolation is not suitable in my opinion due to the rainfall generation mechanisms behind. I suggest "areal rainfall is generated after Benoit et al. (2018a) by constraining actual observations at rain gauge locations". The authors should give a less brief explanation, since in the cited manuscript different versions are applied for rainfall generation (three versions due to different covariance models) and it remains unclear for the reader, which model is used for the current study.

The explanation of the rainfall generation has been revised and extended to make the gridding process clearer (L170-177). The reference to 'interpolation' is kept for the general gridding process only but has been removed from the description of the stochastic model. In practice, the stochastic model used in the present study corresponds to the model version C in Benoit et al. 2018a because it is the best suited version for high resolution data. This is now clearly specified in the revised manuscript. In addition, we provide an open-source MatLab implementation of the model to help interested readers better grasp each step of the gridding process.

8b - Why did the authors choose this rainfall generation instead of a regionalization approach as kriging (maybe with altitude as additional information), inverse distance weighting or Thiessen polygons. The latter is chosen later in the manuscript nevertheless due to computational efforts, so why not for the whole study? Was it the authors intention to add an uncertainty analysis.

Indeed, we choose the stochastic rainfall generation for the valuable estimation of the errors it provides. The Thiessen method also used throughout this paper fills the weak points of the stochastic approach, namely i) the computation time, which is very short using the Thiessen method and allows to explore within a reasonable amount of time all the possible combinations of raingauge networks for their

optimization, and ii) to calculate rainfall fields even with a low number of raingauges. For this last point the stochastic method require at least 5 stations to capture correctly the spatial and temporal rainfall characteristics.

**9 - L154-163 The authors should bring this argument in context with the catchment concentration time.**

We clarify the justification about inter-event time chosen to separate 2 consecutive events, introducing the catchment's response time and the recent paper of Beven (2020) that extensively clarify this concept. This justification reads as: "Accordingly, we assume that this event gives a rough estimate of the catchment's response time (Beven, 2020) i.e. of the time required until the entire catchment contributes to the streamflow response, including the delay caused by runoff transfer to the stream network and from there to the outlet from the hydrologically most distant parts of the catchment. The 90 minutes were therefore selected to maximize the chances of observing a distinct streamflow response for two distinct consecutive rainfall events." (L183-186)

10 - L165-166 The location of the line chosen for the splitting of the catchment seems to be chosen arbitrary. Would a line constructed perpendicular to the main flow direction of the river (or even better, not a straight line but following the lines perpendicular to the isohypses to separate flows exactly) lead to more representable results, since the catchment is then split into a real upper and lower part? Or (thinking the other way around) does it not matter at all and the splitting line could be also drawn from South to North as long as both parts have the same area?

The possible geometries of the splitting line used to compute  $I_{ASYM}$  is now discussed in "5.1 Spatial heterogeneity of rainfall". This justification reads as: "It is noteworthy that this analysis could be affined by investigating different splitting geometries, e.g. by splitting the catchment into west and east parts, thereby separating the large slopes (west) from the steep slopes (east). This and similar spatial asymmetry metrics are case-specific as they rely on the particular geomorphology and topography of the catchment and are thus not directly applicable to other catchments. In particular  $I_{ASYM}$  cannot be used as a tool to compare different catchments." (L570-573)

Furthermore, we also justify the use of the current north/south splitting line when introducing it for the first time in "3.2.2 Spatial rainfall pattern metrics", discussing the Strahler stream order. This justification reads as "This heuristic splitting into two parts is interesting here due to i) the elongated catchment shape and furthermore ii) the clearly distinct stream network organisation in the upper (southern) part of the catchment with more branching than in the northern part (reflected in the Strahler stream order that does not further increase in the norther part, see Figure 1). Accordingly, we assume the rainfall events falling exclusively on one or the other part of the catchment lead to a distinct streamflow response, with a faster and stronger response for events falling on the northern part (closer to outlet, steeper hillslopes, less storage potential than for the southern part)." (L196-201)

**11 - L211-215 I suggest to move this paragraph to the beginning of section 3.3.2**

This comment refers to the statement on how baseflow is separated ("The beginning and the end of the streamflow response determine the initial and final baseflow, respectively; the streamflow volume above the line connecting these two points is considered here as fast runoff.") and we think that it is an integral part of event identification. However, we changed the subsection title "3.3.1 Event identification" to "3.3.1 Identification of streamflow events and fast runoff".

12 - L217-218 The authors declare volume and lag time as "the two key characteristics of streamflow reaction". I do not agree with that. The most important characteristic is peak flow, followed by volume and then lag time and flatting behaviour. Even if all characteristics are considered equal important, the authors should state why peak is not considered in the study. If there were attempts to include peaks which did not work, the authors should state so as "lessons learned" in the manuscript.

L234-239: we corrected the beginning of section 3.3.2. It now read as: "The key metrics to characterize the streamflow response are the peak flow, the fast streamflow volume, the lag time elapsed between rainfall and streamflow response, and the flatting behaviour. For technical reasons we discarded the peak flow (see section 3.3.1**Erreur ! Source du renvoi introuvable.**) and consequently the flatting behaviour. We use the fast streamflow volume through the runoff coefficient (RC), which is obtained by dividing the fast runoff volume by the total rainfall for the given event. The lag time [...]" (L262-266).

And at the end of the section 3.3.1: "It is noteworthy that we do not use peak streamflow to characterize streamflow events, for two reasons: i) given the small size of the catchment and the complex temporal distribution of rain intensities, the streamflow response has rarely a single, well identifiable peak (all events are plotted in Figure S5 in Supplementary Material Part 1); ii) peak streamflow identification is further complicated by the noise in the stage recordings."

13 - L219-221 Is this criterion developed by the authors or should a reference be cited in this context? How was 1/3 chosen as threshold? This value should be catchment-dependent in my opinion, or not? Please clarify.

L241-244: we clarified how the 1/3 threshold on rainfall and streamflow reaction was selected. It now reads as: "Since the start of excess rainfall is not known, the concept of peak flow is difficult to apply to our observed events (Section 3.3.1) and given the varying shape of our hydrographs, we empirically tested different lag formulations; the lag between 1/3 of the rainfall event volume and 1/3 of the streamflow event volume gives the best results in the regression analysis, and is therefore retained. It is noted  $\Delta_{P/Q}$  in the following." (L271-275).

14 - L222 Why is this criterion "1/3 of the rainfall amount" more robust than "start of the rainfall event", although both starting points are linear correlated?

The formulation was not well chosen and has been reformulated along with the sentence of the previous point (L271-275).

15 - L275 Same differences lead to higher asymmetry values for smaller values. To avoid a misinterpretation ("Interestingly...") Pnorth and Psouth could be normalized by the mean event rainfall amount. This would provide deeper insights, especially since larger differences between both parts cannot be seen in the current approach if they occur for events with high rainfall amounts.

We added the columns  $P_{\text{NORTH}}/P_{\text{ALL}}$  and  $P_{\text{SOUTH}}/P_{\text{ALL}}$  in the Table 3.

16 - L323-327. I cannot follow the argumentation here. Please explain in detail how you achieve this conclusion and consider at least one or two sentences for each argument.

The paragraph has been reformulated. It now reads as: "The correlation analysis (Table 4) reveals a strong correlation between rainfall amounts and  $Q_{\text{FAST}}$  (0.77, **Erreur ! Source du renvoi introuvable.**). This suggests that streamflow responses are triggered by saturation-excess, rather than by infiltration capacity-excess: If saturation is exceeded, every unit of rainfall leads to a corresponding unit increase of streamflow, which in turn leads to a strong linear correlation between rainfall amounts and fast streamflow volumes. Furthermore, saturation-excess also implies that a longer rainfall event leads to a higher streamflow response volume (once the saturation threshold is reached, all rainfall contributes to streamflow). This is confirmed by the high correlation (0.74) between the rainfall duration  $P_{\text{DURATION}}$  and  $Q_{\text{FAST}}$ . If, on the contrary, the driving process was the exceedance of the soil infiltration capacity, then only rainfall intensities above the capacity threshold would trigger a corresponding streamflow increase; small rainfall amounts would trigger almost no response. In this case (infiltration-excess), there would be no linear correlation between rainfall amounts or rainfall duration and streamflow amounts, but a strong correlation between fast streamflow amounts and high or maximum precipitation intensity; positive correlations between  $Q_{\text{FAST}}$  and  $P_{\text{max ALL}}$ ,  $P_{\text{max NORTH}}$  or  $P_{\text{max SOUTH}}$  are however all absent (values

of -0.17, -0.16 and -0.08, Table 4). In addition, saturation-excess as a main driver of the fast streamflow response is further confirmed by the clear threshold effect for the generation of streamflow as a function of total event rainfall (**Erreur ! Source du renvoi introuvable.**); a streamflow response only occurs for total rainfall higher than 5 mm." (L429-443)

17 - L330 "to reach a higher "RC" Please rephrase, the manuscript is about observations, not modelling.

We rephrased "to reach a higher RC, we need a higher level of saturation [...]" by "we observe a higher RC when the level of saturation increases [...]" (L447).

18 - L341 composites: If there is a differentiation into wet and dry state, how do the authors achieve only one value for each criterion? Are two values estimated (for wet and dry) and then the arithmetic mean is mentioned? Please clarify!

First, we changed the name "composite network" to "pseudo-dynamic network" in the entire document. A column has been added to Table 3 to show which network (dry/wet) is used for each of the 15 rainfall events having a streamflow reaction, and the missing explanation of the network extent and pseudo-dynamic network calculation is now explained in "3.4.1 Pseudo-dynamic stream network extent". It reads as:

"In absence of exact observations of the stream network extent before the start of each streamflow event, we propose here to use a pseudo-dynamic stream network extent which assigns the dry or the wet network to each streamflow. The network state is chosen based on a measure of the initial catchment wetness conditions." (L284-285)

"This correlation analysis yields an optimum antecedent wetness indicator corresponding to the rainfall over the 3 days preceding the start of a rainfall event, noted  $W_{3\text{days}}$ . Using this indicator, the pseudodynamic network extent is obtained by assigning the dry network state to rainfall events that have  $W_{3\text{days}} < 20$  mm and the wet network state to rainfall events that show  $W_{3\text{days}} \ge 20$  mm. This threshold of 20 mm is selected by maximizing the correlation coefficient between  $D_{\text{HILLS}}$  and RC (see Section **Erreur** ! **Source du renvoi introuvable.**)." (L95-299).

19 - L351-355 It would be nice to have a table with all criteria, where it is stated which one was removed (and why) and which ones were kept. Maybe the information can be added to Table 5 or 6?!

We discuss in this section ("4.3 Identification of dominant hydrologic drivers via regression analysis") only the best models. Among all the models tested through this regression analysis (combining models having one or two explanatory variables), the selection is exclusively based on AICc ranking and R2. The rejection of the models having a lower rank is therefore not detailed in the text or in the tables. We made the model selection method clearer in the text and it now reads as: "The tested models, based on one or two explanatory variables, are summarized in **Erreur ! Source du renvoi introuvable.** for RC and in **Erreur ! Source du renvoi introuvable.** for RC and in **Erreur ! Source du renvoi introuvable.** for  $\Delta_{P/Q}$ . The analysis is based on 14 events (after removing the 24 July event, subset #4 of Table 2) and the best models are selected based on their AICc ranking and coefficient of determination (R2)." (L480-482)

20 - L354 Again, it feels as the number of considered events changes among all subsections.

This has been clarified (see also point 3 above).

21 - L380 What is the reason for IASYM preference in the Southern part? Due to the steeper areas? I would have estimated Northern part, since the hydrograph would have already been smoothed when originated in the South. Please try to find physical explanations to your results.

Thanks for pointing this out. Due to a legacy effect the sentence "And for a single station network, the metric  $I_{ASYM}$  prefers a station location in the southern part rather than in the northern part" is false and has been removed, as well as the plot of the best 1-station network for  $I_{ASYM}$  in the figure 12.

22 - General: Please double-check the abbreviation for "meter above sea level"; I have only seen "m a.s.l." and "m asl" so far, but not "m asl."

The abbreviations of meter above sea level have been corrected from "m asl." to "m asl" throughout the whole paper.

23 - L155 Benoit et al. 2018 <- a or b? I assume a.

Ok (it is "2018a" indeed).

24 - Eq 2, 3, 4 I'm a bit confused what rainfall characteristic is used as input for these equations. Is every raster cell with rainfall used (so I understood it from the text) or only the centre of the rainfall events (as mentioned in Table 1)?

The rainfall characteristics and space-time resolution used into the Equations 2, 3 and 4 have been clarified. The text now reads as "[...] where *i* and *j* are the coordinates of rainfall location within the grid, P(i,j,t) is the rainfall amount previously calculated using the stochastic method (section 3.2.1) for each of the 10 x 10 meters grid cell at each 2-minute time step *t*, and  $d_{HILLS}(i,j)$  is the distance of this grid cell to the nearest stream network grid cell (following the line of steepest descent in the 2 x 2 m DEM (swissALTI3D, 2012))." (L217-220)

25 - L163 "overlooked" -> ignored

Ok.

26 - Eq. 2, 3, 4 The term in the numerator should be put in brackets (Eq. 2: "P(..)dHills" -> "(P(..)dHills)")

The equations formulation (missing brackets) have been corrected for  $D_{\text{HILLS}}$ ,  $D_{\text{STREAM}}$  and  $H_{HAND}$  (Equations 2, 3 and 4, respectively).

27 - L195 DHAND is not a distance as indicated by the D, and in the text the variable is introduced with HAND. I suggest to stick to HAND throughout the manuscript to avoid confusions with the other two "real" distances".

We now use the abbreviation  $H_{HAND}$  instead of  $D_{HAND}$  throughout the whole paper, figures and associated documents.

28 - L202 Section 3.5 includes no network extent description. Is it missing in the manuscript?

The missing explanation of the network extent is now explained in "3.4.1 Pseudo-dynamic stream network extent" (L278-299).

29 - L268 317.8 mm - Is it areal rainfall amount sum or sum over all stations?

We specified (L355) that the value of 317.8 mm is the areal rainfall amount.

30 - L268-269 please provide also the mean values, not only the highest and lowest values, so that the reader get a "feeling" for the rainfall events.

The mean values have been added (L356).

31 - L275 again, please don't use the term average, use mean or median to be more concise. Since Iasym can be positive and negative, the median of its absolute values would be worth to show instead of just the mean, since positive and negative values are levelling out each other.

Indeed, in this case using the median value of  $I_{ASYM}$  (0.025) is better than using the mean, it has been corrected (L363-364). Also, the misuses of "average" has been corrected when needed throughout the whole paper.

32 - Fig. 5 and 6 For a logical order the figures should show the rainfall events first, followed by the discharge plot.

The figures showing rainfall events records (Figure 3 and Figure 4 in the main text, and all the figures of the Supplementary Material part 2) have been rearranged to have rainfall data above streamflow data.

33 - L279 "One strongly asymmetric and high intensity event" -> "One strong asymmetric and very intense event"

Ok.

34 - L283 A volume can't be fast (check also for later occurrences...)

Ok.

35 - L288 In the sentence before authors mention that the number of events under consideration are reduced by "1", but here again 48 events are studied (also in the following subsections).

Indeed, it is confusing. The line it is referred to "This event and its streamflow reaction are excluded from further analysis" has been replaced by "This event and its streamflow reaction are excluded from further analysis involving the hydrological response" (L373-374).

36 - L289 The authors should state what wet and dry networks are. I found it later in the caption of Table 1 in S1, but it would lead to clarifications here. Also, the Table 1 in S1 should be shown in the manuscript, since the written part in Section 4.1.2 is more confusing than explaining for me.

The dry and wet networks are now introduced in "2 Study area". It now reads as: "The actual extent of the stream network is based on observations during Summer 2017 (dry and wet periods) and its exact path was calculated using the Swiss digital elevation model at a resolution of 2 m (swissALTI3D, 2012)." (L126-127)

It is detailed later on in "3.4.1 Pseudo-dynamic stream network extent" and it reads as "The extent of the stream network evolves as a function of the catchment wetness conditions. Its minimal and maximal extent (Erreur ! Source du renvoi introuvable.) are determined manually by identifying the uppermost points of the catchment where streamflow was observed in the field during summer baseflow (minimum extent, called *dry* state) and during summer high flow (maximum extent, called *wet* state)."

37a - Fig9 "events without reaction are not shown" belongs to part b), not a). Please correct the caption.

Ok.

37b - General: Maybe I missed it, but which temporal resolution was used to calculate the correlation (and other criteria)? 2min as this is the resolution of the rain gauge? Or are values aggregated up to e.g. 1h? This has a high impact on the values of the correlation coefficient.

The temporal resolution of times series used for correlation calculations is 2 minutes. The correlation between events is done at the event-scale. This is now clarified in the text.

38 - L339 "absence of correlation". Correlation cannot be absent. Better to speak of low correlation or provide absolute values.

Ok.

39 - L384-386 "is assessed", "is evaluated" – two verbs, please rephrase the sentence.

Ok. It now reads as "Considering the small dataset underlying this analysis (23 events), the robustness of the best networks is assessed for two selected metrics (for the  $P_{ALL}$  and  $I_{ASYM}$ ) by re-computing the optimal network if between 1 and 3 events are removed from the dataset." (L329-331)

**40 - L402 "what we previously thought"? What was the hypothesis of the authors before?**

We reformulated the explanation about the outperformance of  $D_{\text{STREAM}}$  over  $D_{\text{HILLS}}$  for the prediction of RC and lag time. It now reads as: "We could expect that in that kind of steep environments, the residence time in hillslopes strongly dominates over residence times in the stream network (Nicotina et al., 2008); the fact that  $D_{\text{STREAM}}$  outperforms here  $D_{\text{HILLS}}$  for the prediction of RC and lag time may show that even in steep environments, with a priori fast instream processes and limited storage, the riparian area and related subsurface exchange processes could play a more prominent role. The fact that the travel distance in the stream network explains more of the RC variation than  $D_{\text{HILLS}}$  might be an indirect effect: the longer the travel distance in the stream network, the more likely are delays due to exchange with groundwater in the riparian area." (L578-584)

**41 - L421 "three station network" It would be nice to provide the resulting density here as well as "(general) recommendation".**

The results are now also presented in term of raingauge density in the figure 7 and in the text referring to (L505-506) and later in the text L536-537.

**Responses to the comments of Reviewer #2:**

1) I already reviewed the initial version of this paper, which aims at highlighting the values of high density rain gauges networks for hydrological purposes in a small catchment of mountainous areas. I still believe that the topic is interesting and relevant for the community. It furthermore has other potential applications in urban areas which are also small and quickly reactive catchments where rainfall variability has strong consequences.

A reference to urban hydrology (Cristiano et al., 2017) has been added into the introduction. It now reads as "While our analysis focuses here on a small natural headwater catchment, it is noteworthy that the developed rainfall monitoring and data analysis framework might also be of interest for urban hydrology, which deals with similar questions regarding how spatial rainfall patterns, runoff generation processes and flow network geometry lead to peak flows in urban drainage systems (for a review, see the work of Cristiano et al., 2017)." (L77-81)

2) The minor difficulties with regards to the presentation and understanding of the paper have been corrected. Results are now better presented with the new figures. However, the main point was not addressed, i.e. the fact that the authors aims at showing the importance of grasping the spatio-temporal variability of the rainfall process in the prediction of flows, but the chosen indicators are only event based averages.

We removed the misleading formulation "runoff prediction" from the abstract, that has also been adapted to the changes made to the paper.

3) Furthermore, the main rainfall variability (which is at the core of the paper) indicator used is too simplistic since it is basically an asymmetry indicator on the total depth splitting the catchment in two. So I still think that indicators actually accounting for the spatiotemporal variability of the rainfall and hydrologic response should be implemented to actually address the stated topic of the paper. Implementing them requires major modifications of the paper. I guess that this would enable to highlight more precisely the importance of dense networks of rainfall measurement devices.

We agree that  $I_{ASYM}$  is a simple indicator to capture the key rainfall field properties for the hydrological response. In other studies and namely in urban hydrology such an indicator is typically based e.g. on the variogram or on the spatial moments of rainfall with continuously observed rainfall fields (radar images). We added this comment in "3.2.2 Spatial rainfall pattern metrics" and it reads as: "Spatial rainfall patterns are classically characterized with geostatistical tools, including variograms (Berne et al., 2004) or with spatial moments of rainfall (Smith et al., 2002;Zoccatelli et al., 2011;Mei et al., 2014), in particular in presence of observed rainfall fields, e.g. from radar images. Here we propose to use more hydrological-process oriented metrics that explicitly account for known features of the catchment and the stream network." (L190-194)

The asymmetry indicator is just one of the indicators used in the study, along with the geomorphological distances, which corresponds to the above first order spatial moments, albeit decomposed according to hillslope and stream network flow distances. As we answered in the public discussion, we tried the second order moment of distance metrics, but it does not show any noticeable correlation with a rainfall or streamflow metric. We added this result in the text: "It is noteworthy that these two metrics,  $D_{HILLS}$  and  $D_{STREAM}$  correspond to the aforementioned first order spatial rainfall moments, albeit decomposed according to hillslope and stream network distances, similar to what was proposed by Zoccatelli et al., 2015 in their analytical framework to quantify the smoothing of spatial rainfall moments; however, no significant correlation could be found the retained streamflow metrics." (L229-233)

Finally, we also added the section "4.1.3 Temporal evolution of rainfall metrics", please see our answer to the point 7 below.

4) l. 110 - 115: "The actual extent of the stream network is based on observations during dry and wet periods during Summer 2017 and its exact path was calculated using the Swiss digital elevation model at a resolution of 2 m (swissALTI3D, 2012)." I think that the intrinsic fractal nature of river networks should be mentioned and discussed. The concept of variable network used after also seems interesting.

We implemented at the end of "5.1 Spatial heterogeneity of rainfall" a comment about the fractality of the river network. It reads as: "However, future work on the role of water residence time in the stream network will necessarily require more detailed field data on the temporal evolution of the stream network. This will in addition open new perspectives to quantify how the stream network extension is imprinted in the streamflow response: in fact, as discussed by Rinaldo et al. (1995), the intrinsic fractal nature of the stream network is not transferred to the streamflow response and, accordingly, there is potential to infer the stream network extension from observed streamflow records, provided that we have high resolution rainfall data to disentangle the different effects." (L587-592)

5) l. 150-151: "Some additional artefacts were recorded, probably generated by strong winds creating resonance. These periods have been manually removed from the data". It should clarified how the data was selected for being removed and what portion was removed.

We clarified how raw rainfall data were selected and some parts removed. It now reads as: "Additional artefacts were recorded, probably generated by strong winds creating resonance. Some stations in fact recorded very strong and highly variable rainfall over several hours during periods with high wind velocity but during days without any observed rainfall in the combined MeteoSwiss radar-rain gauge data (Sideris et al., 2014). Four periods (over 4 different days) have been manually removed from the data." (L164-167)

6) 1. 154-157: It is a great improvement to use this stochastic procedure. Nevertheless, I believe that more details on the interpolation procedure are needed. It should be clarified how the 20 samples are used (computing the error bars in 8-10)?

We added details about the stochastic procedure and error bar computation. It now reads as: "Before further analysis, the rainfall amounts measured by each station were interpolated to a 10 by 10 m grid at a 2 min time step using a high-resolution stochastic approach developed by Benoit et al. (2018a). In a nutshell, it generates an ensemble of stochastic space-time rain fields constrained by the actual observations at the rain gauge locations. The resulting ensemble (here composed of 20 realizations) can be used to analyze spatial rainfall uncertainty or to construct a single rainfall estimator. Following Benoit et al. (2018a), a non-separable and asymmetric covariance function was used to perform the simulations, which allows modelling rainfall advection and diffusion observed in the raw data. Areal rainfall time series are calculated for each of the 20 realization, and from these a single time series (mean and standard deviation) of the areal rainfall." (L170-177)

7) Eq. 1 on I\_ASYM. As already mentioned, it seems a too simplistic indicator to grasp spatio-temporal variability of the rainfall process. An initial simple suggestion could for instance be to look for the temporal evolution of I ASYM during an event. But other indicators are needed

We added figures showing the evolution of  $D_{\text{STREAM}}$ ,  $D_{\text{HILLS}}$  and  $I_{\text{ASYM}}$  in the supplementary Material, and added in the text a qualitative discussion of the temporal evolution of the rainfall metrics in "4.1.3 Temporal evolution of rainfall metrics". It reads as:

"We computed the temporal evolution of the rainfall metrics to unravel potential temporal evolution patterns in  $I_{\text{ASYM}}$ ,  $D_{\text{HILLS}}$  and  $D_{\text{STREAM}}$  and their relation to the streamflow response (full results are available in the Supplementary Material part 1). The temporal evolution of the two distance metrics is overall rather flat with no clear fluctuation patterns. There is only one event with a pronounced temporal trend for  $D_{\text{HILLS}}$  (Q event #1).

For  $I_{ASYM}$ , some events show interesting temporal patterns. For example, during the double peak runoff of **Erreur ! Source du renvoi introuvable.**,  $I_{ASYM}$  shows an almost constant negative value suggesting that the corresponding double peak rainfall event remained stationary on the northern part of the catchment over its entire duration and therefore caused the double peak streamflow response.

For the first two streamflow events, the  $I_{ASYM}$  metric switches from strongly positive to close to zero during the event, implying that the rainfall field moved towards the outlet during the event; in other words, the rainfall cloud follows the overall water movement through the catchment and thereby leads to a stream response concentration. This might explain why these two events are the only ones that show a pronounced single peak streamflow response. However, given the low number of observed events and the diversity of temporal patterns, these insights cannot be further used for a quantitative analysis." (L391-404)

**8a) l. 212-215: the explanation on why not using streamflow variations (notably peak flow) is not very convincing.**

This point has been clarified in the public discussion.

(https://editor.copernicus.org/index.php?\_mdl=msover\_md&\_jrl=13&\_lcm=oc108lcm109w&\_acm=g et\_comm\_sup\_file&\_ms=87052&c=189590&salt=10720610621776386148) and we added the figure showing the peak flows to the Supplementary Material (Figure S5).

8b) If the purpose is to investigate the importance of spatiotemporal variability, I guess studying the temporal variability of the simulated streamflow is needed.

We decided to add a modelling component to this paper; the model is discussed in the public discussion (https://editor.copernicus.org/index.php?\_mdl=msover\_md&\_jrl=13&\_lcm=oc108lcm109w&\_acm=g et comm sup file& ms=87052&c=189590&salt=10720610621776386148).

Corresponding modifications of the paper are i) at the end of the introduction (L92-94), ii) presenting the model used in the method part "3.6 Rainfall-runoff model" (L335-350), iii) in the results section in "4.4.3 Optimum network evaluation" (L544-555), iv) with the Figure 15 summarizing the results of the different simulations and v) in the Supplementary Material part 1 with the Figure S9 (map of subcatchments), Figure S10 (the results of all simulations per event) and Figure S11 (the results of simulations per event, cumulated over time.

**On the value of**Even event-scale hydrological response characterization benefits from high density rain gauge observations**

Anthony Michelon1, Lionel Benoit1, Harsh Beria1, Natalie Ceperley1,2, Bettina Schaefli1,2

1 Institute of Earth Surface Dynamics (IDYST), Faculty of Geosciences and Environment, University of Lausanne, Lausanne, 1015, Switzerland
 2 Now at: Institute of Geography (GIUB), Faculty of Science, University of Berne, Switzerland

*Correspondence to*: Anthony Michelon (anthony.michelon@unil.ch)

**Abstract.**

5

- Spatial rainfall patterns exert a key control on the catchment scale hydrologic response. Despite recent advances in radar-based rainfall sensing, rainfall observation remains a challenge particularly in mountain environments. This paper analyzes the importance of high-density rainfall observations for a 13.4 km2 catchment located in the Swiss Alps where rainfall events were monitored during 3 summer months using a network of 12 low-cost, drop-counting rain gauges. We developed a data-based analysis framework to assess the importance of high-density rainfall observations to help predict hydrologic processes.the hydrological response. The framework involves the definition of spatial rainfall distribution metrics based on hydrological and
- 15 geomorphological considerations, and thea regression analysis of how these metrics explain the hydrologic response in terms of runoff coefficient and lag time. The gained insights on dominant predictors are then used to investigate the optimal raingaugerain gauge network density for predicting the hydrologicalstreamflow response metrics-in, including an extensive test of the studied eatchment.
[revised manuscript text omitted]
 raingaugethe rain gauge locations (over. The resulting ensemble (here composed of 20 realizations), and ) can be used to use this ensemble analyze spatial rainfall uncertainty or to interpolate sparse rain observationsconstruct a single rainfall estimator.

Following Benoit et al. (2018a), a non-separable and asymmetric covariance function was used to perform the simulations,

- 175 which allows modelling rainfall advection and diffusion observed in the raw data. Areal rainfall time series are calculated for each of the 20 realization, and from these a single time series (mean and standard deviation) of the areal rainfall.
   Using the interpolatedareal rainfall fields, time series, the rainfall events wereare identified as rainy periods with rainfall higher than 1 mm separated by at least 90 minutes without rain, with rainfall smaller than 1 mm. This inter-event duration was selected based on of 90 minutes corresponds to the observed delay between the rainfall onset and the streamflow response for the large
- 180 event recorded on August 23rd (detailed in the part 2 of supplementary material); for details see Supplementary Material), which occurred during an otherwise dry period. The streamflow reactionresponse to the first half-hour of this rainfall event was caused only by rainfall in the southern half of the catchment (stations 8 to 12). Ninety minutes was), corresponding thereby to the most distant event (from the outlet). [Accordingly, we assume that this event gives a rough estimate of the catchment's response time (Beven, 2020) i.e. of the time required until the entire catchment contributes to the streamflow response,
- 185 including the delay caused by runoff transfer to the stream network and from there to the outlet from the hydrologically most distant parts of the catchment. The 90 minutes were therefore selected to maximize the chances of observing a distinct streamflow reactionresponse for two distinct consecutive rainfall events. In addition, events with a total amount of rainfall under 1 mm are overlooked in the following.

**3.2.2 Spatial rainfall pattern metrics**

190 To investigate the relationship between dominant spatial rainfall patterns and streamflow response Spatial rainfall patterns are classically characterized with geostatistical tools, including variograms (Berne et al., 2004) or with spatial moments of rainfall (Smith et al., 2002;Zoccatelli et al., 2011;Mei et al., 2014), in particular in presence of observed rainfall fields, e.g. from radar

6

**Commenté [AM5]:** Clarified according to Reviewer #2 – Point 5

**Commenté [AM6]:** Changed according to Reviewer #1 – point 23

**Commenté [AM7]:** Modified according to Reviewer #2 – Point 6 and Reviewer #1 – 8a

**Commenté [AM8]:** Catchment time definition added according to Reviewer #1, point 9.

| 1   | images. Here we propose to use more hydrological-process oriented metrics that explicitly account for known features of the                                                                                     |                                                                            |
|-----|-----------------------------------------------------------------------------------------------------------------------------------------------------------------------------------------------------------------|----------------------------------------------------------------------------|
|     | catchment and the stream network.                                                                                                                                                                               |
Commenté [AM9]: Added according to Reviewer #2 – Point 3               |
| 195 | To build a first such metric, the catchment is split into two parts of equal area by a west-east line (Figure 1), delimiting an area                                                                            |                                                                            |
|     | close to the outlet in the northern part, and an area farther away in the southern part. This heuristic splitting into two parts is                                                                             |                                                                            |
|     | interesting here due to i) the elongated catchment shape and furthermore ii) the clearly distinct stream network organisation in                                                                                |                                                                            |
|     | the upper (southern) part of the catchment with more branching than in the northern part (reflected in the Strahler stream order                                                                                |                                                                            |
|     | that does not further increase in the norther part, see Figure 1). Accordingly, we assume the rainfall events falling exclusively                                                                               |                                                                            |
| 200 | on one or the other part of the catchment lead to a distinct streamflow response, with a faster and stronger response for events                                                                                |                                                                            |
|     | falling on the northern part (closer to outlet, steeper hillslopes, less storage potential than for the southern part).                                                                                         |
Commenté [AM10]: Splitting line detailed according to                  |
| '   | The interpolated amounts of rainfall received by the southern and northern parts of the catchment, $P_{\text{NORTH}}$ and $P_{\text{SOUTH}}$ , are                                                              | Reviewer #1, point 10                                                      |
|     | compared and normalized by the total amount of rainfall to create an index of spatial rainfall asymmetry $I_{ASYM}$ :                                                                                           |                                                                            |
|     | $I_{ASYM} = \frac{P_{SOUTH} - P_{NORTH}}{(P_{SOUTH} + P_{NORTH})},$ (1)                                                                                                                                         |                                                                            |
| 205 | If rainfall is equally distributed between the northern and the southern parts, then $I_{ASYM} = 0$ . The extreme values -1 and 1                                                                               |                                                                            |
|     | express rainfall concentration exclusively in the northern or the southern part of the catchment, respectively. We consider a                                                                                   |                                                                            |
|     | rainfall event as asymmetric when at least 2 times more rain ishas precipitated over one part of the catchment than over the                                                                                    |                                                                            |
| '   | other, i.e. when $I_{ASYM}$ is below -0.33 or above +0.33.                                                                                                                                                      |                                                                            |
|     | To further analyze the relationships between the spatial distribution of rainfall and the streamflow response, we characterize                                                                                  |                                                                            |
| 210 | the geomorphological distance of incoming rainfall from the outlet, assuming that this distance should reflect to some degree                                                                                   |                                                                            |
|     | the timing and the shape of the streamflow reactionresponse of the catchment: following the terminology of Rinaldo et al.                                                                                |                                                                            |
| •   | (2006b), transport at the basin scale can be analyzed in terms of travel in the unchannelled state (i.e. in the hillslopes) and                                                                                 |                                                                            |
|     | travel in the channelled state (i.e. in the stream network).                                                                                                                                                    |                                                                            |
|     | Accordingly, we estimate for each rainfall event the weighted averagemean unchannelled distance to the stream network as:                                                                                       |                                                                            |
| 215 | $D_{HILLS} = \frac{1}{t} \sum_{i} \frac{\sum_{i} \sum_{j} P(i,j,t) d_{HILLS}(i,j))}{\sum_{i} \sum_{j} P(i,j,t)} \sum_{t} \frac{\sum_{i} \sum_{j} P(i,j,t)}{\sum_{i} \sum_{j} P(i,j,t)} ,$                       | Commenté [AM11]: Brackets added according to Reviewer #1 – point 26 |
| 1   | (2)                                                                                                                                                                                                             |                                                                            |
| 1   | where $t$ is the time step, $i$ and $j$ are the coordinates of rainfall location within the grid, $P(i,j,t)$ is the rainfall amount previously                                                                  |                                                                            |
|     | calculated using the stochastic method (section 3.2.1) for each of the 10 x 10 meters grid cell at each 2-minute time step t, and                                                                               |                                                                            |
| 1   | $d_{HILLS}(i, j)$ is the distance of this grid cell to the nearest stream network grid cell (following the line of steepest descent in the                                                                      |                                                                            |
| 220 | 2 x 2 m DEM (swissALTI3D, 2012)).                                                                                                                                                                               |
Commenté [AM12]: Modified according to Reviewer #1 – point      |
| 1   | Similarly, we compute the weighted averagemean channelled distance between a point of introduction into the stream network                                                                                      | 24                                                                         |
| 1   | and the outlet as:                                                                                                                                                                                              |                                                                            |
|     | $D_{STREAM} = \frac{1}{t} \sum_{i} \frac{\sum_{i} \sum_{j} (P(i,j,t)) d_{STREAM}(i,j))}{\sum_{i} \sum_{j} P(i,j,t)} \sum_{i} \frac{\sum_{i} \sum_{j} (P(i,j,t)) d_{STREAM}(i,j))}{\sum_{i} \sum_{j} P(i,j,t)},$ | Commenté [AM13]: Brackets added according to Reviewer #1 – point 26        |
| 1   | (3)                                                                                                                                                                                                             |                                                                            |

225 where  $d_{STREAM}(i, j)$  is the distance along the stream network from the point of introduction to the outlet. For each cell of the stream network, this distance is calculated once based on the 2 x 2 m DEM. The DHILLS metric gives an estimate of the average distance that incoming rainfall has to travel on the hillslopes before reaching the stream network, and DSTREAM the average distance for the water particle entering the stream network to reach the outlet. It is noteworthy that these two metrics,  $D_{HILLS}$  and  $D_{STREAM}$  correspond to the aforementioned first order spatial rainfall 230 moments, albeit decomposed according to hillslope and stream network distances, similar to what was proposed by Zoccatelli et al., 2015 in their analytical framework to quantify the smoothing of spatial rainfall organisation effects by channel residence time. It would be tempting to use also higher order rainfall moments; however, no significant correlation could be found to retained the streamflow metrics. Commenté [AM14]: Added according to Reviewer #2 – Point 3 In addition to the above two metrics related to the theory of geomorphological dispersion (Rinaldo et al., 2006b), we use the 235 height above the nearest drainage (HANDHHAND) terrain metric (Renno et al., 2008;Gharari et al., 2011;Nobre et al., 2011) to account for the topography. Based on the 2 x 2 m DEM, the normalized terrain heights dHAND are calculated by comparing the elevation of each grid cell to the elevation of the nearest stream network cell in which the water is routed. The average HANDmean  $H_{HAND}$  value for a rainfall event is given by:  $D_{HAND}H_{HAND} = \frac{1}{t} \sum_{t} \frac{\sum_{t} \sum_{j} (\mathcal{P}(i,j,t)) d_{HAND}(i,j))}{\sum_{t} \sum_{t} \mathcal{P}_{t} \mathcal{P}_{t}(i,t,t)} \sum_{t} \frac{\sum_{l} \sum_{j} (\mathcal{P}(i,j,t)) h_{HAND}(i,j))}{\sum_{t} \sum_{t} \mathcal{P}_{t}(i,t,t)}$ Commenté [AM15]: Brackets added according to Reviewer #1 - $\sum_{i} \sum_{j} P(i,j,t)$  $\sum_{i} \sum_{j} P(i,j,t)$ point 26 240 (4)Since the extent of the stream network is dynamic, its minimal and maximal extent () are determined manually by identifying the uppermost points of the catchment where streamflow has been observed in the field during summer baseflow (minimum extent) and during high flow (maximum extent). The 3 distance metrics are computed with respect to both the dry and wet river network extents the network extent to be used per rainfall event is then determined during the rainfall-streamflow

**response analysis (Section 3.4.1).**

**3.3 Streamflow response**

245

**3.3.1 Event identification**

**3.3.1 Identification of streamflow events and fast runoff**

The beginning and the end of each streamflow event are identified manually using a data visualization tool (developed in MathWorks MatLab 2017a, see Figure 3 and Figure 4). This choice of a visual expertise was made based on the observation that automatic identification of streamflow events would require almost a case-by-case filtering and parametrization, and thus would not be generalizable. This is partly related to a potentially high signal-to-noise ratio for river stage recordings during sediment transport events, a phenomenon potentially very important after a strong streamflow variation. The result of this visual identification for each streamflow event is displayed in the part 2 of Supplementary Material. **Commenté [AM16]:** Renamed as stated in the answer to Reviewer #1 – Point 11

The beginning and the end of the streamflow response determine the initial and final baseflow, respectively; the streamflow volume above the line connecting these two points is considered here as fast runoff. It is noteworthy that we do not use peak streamflow to characterize streamflow events, for two reasons: i) given the small size of the catchment and the complex temporal distribution of rain intensities, the streamflow response has rarely a single, well identifiable peak; (all events are plotted in Figure S5 in Supplementary Material Part 1); ii) peak streamflow identification is further complicated by the noise in the stage recordings.

**3.3.2 Streamflow metrics**

265

The key metrics to characterize the hydrologiestreamflow response in terms of are the peak flow, the fast streamflow volume, the lag time elapsed between rainfall and streamflow response, and the flatting behaviour. For technical reasons we discarded the peak flow (see section 3.3.1timing,) and consequently the flatting behaviour. We use the runoff coefficient and the lag time, the two key characteristics of streamflow reaction fast streamflow volume through the runoff coefficient (RC)), which is

 obtained by dividing the fast runoff volume by the total rainfall for the given event. A metric for the elapsed

 The lag time between the rainfall event and the streamflow reaction is obtained

 when one third of the rainfall event has fallen and when one third of the corresponding streamflow volume has passed the gauge and is called  $A_{PO}$ . Given the visual assessment of the start of the streamflow event, this measure is deemed more robust

270 than the clapsed time between the start of the event, and is indicative for the time when a significant part of excess rainfall (the part of rainfall that causes the streamflow response) and the peak flow (McCuen, 2009). Since the start of excess rainfall is not known, the concept of peak flow is difficult to apply to our observed events (Section 3.3.1) and given the varying shape of our hydrographs, we empirically tested different lag formulations; the lag between 1/3 of the rainfall event volume and 1/3 of the streamflow volume has reached the outletevent volume gives the best results in the regression analysis, and is therefore retained. It is noted *Δ*PQ in the following.

**3.4 Rainfall-streamflow response characterization**

**3.4.1 We analyze Pseudo-dynamic stream network extent**

The extent of the relationships between the spatial distribution of rainfallstream network evolves as a function of the catchment wetness conditions. Its minimal and maximal extent (Figure 1) are determined manually by identifying the hydrological
 response based on a correlation analysisuppermost points of the above metrics, followed by a regression analysis to identify the key variables that explain the runoff coefficient and catchment where streamflow lag time. This analysis requires a was observed in the field during summer baseflow (minimum extent, called *dry* state) and during summer high flow (maximum extent, called *wet* state).
 In absence of exact observations of the stream network extent before the start of each streamflow event, we propose here to

285 use a pseudo-dynamic stream network extent which assigns the dry or the wet to each streamflow. The network state is chosen

9

**Commenté [AM17]:** Modified according to Reviewer #1, point

Commenté [AM18]: Modified according to Reviewer #1, point

**Commenté [AM19]:** Details about the 1/3 threshold added according to Reviewer #1, point 13

Commenté [AM20]: Added according to Reviewer #1 - point 36

Commenté [AM21]: Added according to Reviewer #1 – point 18

based on a measure forof the initial catchment wetness conditions, which areis known to be the major variable explaining the dynamics of the hydrological response to different rainfall events (Penna et al., 2011;Rodriguez-Blanco et al., 2012), in particular through the creation of runoff thresholds (Zehe et al., 2005;Tromp-van Meerveld and McDonnell, 2006). Many studies use the baseflow before the start of a streamflow event as an indicator for the antecedent moisture statewetness

- 290 conditions of the catchment. For snow-influenced catchments with a highly seasonal streamflow regime, this indicator might not reflect the actual saturationwetness conditions. Hence, we rather quantify initial wetness conditions in terms of antecedent rainfall, i.e. using the cumulative rainfall (in mm) that occurred during a period from 1 to 5 days before ana given rainfall event. All usedThe actual time span is selected based on a correlation analysis between antecedent rainfall over 1 to 5 days and the retained streamflow metrics (Section 4.2.1are-summarized in Table 1.- and following).
- 295 This correlation analysis yields an optimum antecedent wetness indicator corresponding to the rainfall over the 3 days preceding the start of a rainfall event, noted  $W_{3 days}$ . Using this indicator, the pseudo-dynamic network extent is obtained by assigning the dry network state to rainfall events that have  $W_{3 days} \le 20$  mm and the wet network state to rainfall events that show  $W_{3 days} \ge 20$  mm. This threshold of 20 mm is selected by maximizing the correlation coefficient between  $D_{HILLS}$  and RC (see Section 0).]

**300 3.4.2 Regression analysis**

We analyze the relationships between the spatial distribution of rainfall and the hydrological response based on a correlation analysis between the spatial rainfall pattern metrics (Section 3.2.2) and the streamflow metrics (Section 3.3.2) at the event scale, followed by a regression analysis to identify the key variables that best explain the runoff coefficient, RC, and the streamflow lag time,  $\Delta_{PQ}$ . All used metrics are summarized in Table 1.

305 After the initial screening via correlation analysis, we use a pure quadratic regression to further investigate which combination of rainfall pattern characteristicsmetrics and initial wetness conditions arecondition yields the best predictors of the runoff coefficientprediction of RC and the lag timedpred. Pure quadratic regression (i.e. without multiplication of explanatory variables) is chosen because the small number of observed streamflow events prevents using more complex models. Model selection is performed using the Akaike Information Criterion (AIC)(Akaike, 1974), noted here as  $I_{AIC}$ :

**310 $I_{AIC} = n \ln \left(\frac{S_{RSS}}{n}\right) + 2k + C ,$**

(5)

(6)

where *n* is the number of events, *k* the number of coefficients,  $S_{RSS}$  the residual sum of squares and *C* a constant that can be ignored when comparing different models based on the same data set. As we manage small sample sizes (Burnham et al., 2011), we compute and use a corrected version of the AIC (AICc, noted here  $I_{AICc}$ ):

 $I_{AICC} = I_{AIC} + \frac{2k(k+1)}{n-k-1}$

315 For both AIC and AICc, the best model is the one having the lowest score.

event scale only. There is no intra-event correlation.

Commenté [AM23]: Point 37b. Precise in the answer that it is at

**Commenté [AM22]:** Added according to Reviewer #1 – point 18

**3.5 MeasurementRain gauge network configuration analysis**

Assuming that the actual rainfall measurement network is sufficient to capture the full spatial distribution of rainfall in the studied catchment, we assess the ability of partial networks to reproduce the identified best explanatory variables. The aim is twofold: i) identifying the best configuration for a future permanent observation network and ii) evaluate the added value of additional rain gauges in a partial network with respect to the identified key metrics (Section 4.4 and 0).

The quality of a partial network configuration is evaluated comparing the value (e.g. total rainfall) by event obtained with the partial network to the reference value obtained with the full network setup. We evaluate all the possible combinations of partial networks composed of less than 12 stations, i.e. 4094 possibilities. Each configuration is evaluated based on the root mean square error (RMSE):

325 RSME :=
$$\sqrt{\sum_{vt} \frac{(X_k(t) - X_{ref}(t))^2}{N}}$$
,

(7)

where  $X_k$  is the selected rainfall metric (e.g. rainfall amount) at time step *t* corresponding to the *k*-th network configuration,  $X_{ref}$  the respective value obtained reference network set-up, and *N* the number of time steps. The rainfall amounts measured by each station were interpolated to a 10 by 10 m grid at a 2 min time step using the Thiessen polygons method. The interpolation method developed by Benoit et al. (see section 3.2) cannot be used in this context because i) it requires at least 5 measuring points to perform adequately and ii) the computation time would be excessive to explore the 4094 combinations of stations for

**each event.**

330

320

The best network for each number of stations is the one with the lowest RMSE. A sensitivity analysis is completed by removing from 1 to 3 rainfall events to the 23 events dataset, yielding 2047 datasets evaluated for each partial network configuration. The most frequent network configuration validates the robustness of the result.

**335 3.6 Rainfall-runoff model**

To further validate the obtained optimal rain gauge network configuration, we set up a a semi-distributed, event-based rainfallrunoff model. This model first simulates the mobilization of water at the sub-catchment scale (25 sub-catchments) using a Soil Conservation Service Curve Number (SCS-CN) approach (SCS, 1972). Next, the streamflow response is obtained by convolving the resulting hillslope responses with a travel path distribution derived from the stream network geometry (Schaefli

- 340 et al., 2014). The subcatchments and the stream network geometry are identified using *TopoToolbox* (https://topotoolbox.wordpress.com), in which travel paths correspond to the distance between the bottom part of each sub-catchment and the catchment outlet. In this model we focus on the fast response (i.e. runoff) of the catchment, and baseflow (defined here as the average discharge during the 30 min preceding event start) is subtracted from the actual discharge prior to runoff modeling. For calibration, the model is run using the mean of the 20 stochastic rainfall realizations as reference input;
- 345 it is then calibrated against observed runoff (i.e. discharge baseflow) through likelihood maximization assuming that the model residuals are normally distributed (e.g. Schaefli et al., 2007). After calibration the event-based runoff model is applied to the different network configurations to test how rain gauge network geometry influences the simulated runoff response. As

the stochastic rainfall interpolation cannot be performed with a number of observation points as low as 3 stations (or less), we use the Thiessen polygons method to interpolate the rainfall fields from the 1 to 3-station rain gauge network obtained during optimal network analysis.

**4 Results**

350

**4.1 Rainfall events**

**4.1.1 AmountsAreal rainfall and asymmetry**

The available 3-month measurements window between July 1st and September 23th 2018 captured 48 rain events (detailed in the part 2 of the Supplementary Material) for a total areal rainfall amount of 317.8 mm. The areal rainfall amount per event 355 ranges from 1 mm to 43.5 mm1 (mean of 6.6 mm), and event duration ranges from 32 minutes to 10.5 hours- (mean of 2.8 hours); these records do not show any evidence of altitude effect on the rainfall amount (R2 = 0.06). Despite the sequential deployment of the 12 rain gauges and other technical issues (see section 3.1), the rainfall events were all measured by at least 7 stations; 36 out of 48 events were recorded by at least 10 stations and 23 events were recorded by 12 stations. The different 360 subsets used in this study are detailed in Table 2. Details for all recorded rainfall events and the corresponding streamflow are shown in summary plots, as illustrated in Figure 3 and Figure 4 (all events are presented in the Supplementary Material). Most events show a relatively homogeneous spatial distribution of rainfall events (see an example in Figure 4), with only few events showing a strong asymmetry (Figure 5): the correlation between PNORTH and PSOUTH equals 0.91, with an averagea median  $I_{ASYM}$  of -0.04025. Interestingly, strong spatial asymmetry mainly affects events with low rainfall amounts, with 7 out of 8 365 asymmetric events (when  $|I_{ASYM}| > 0.33$ ) receiving below 5 mm (Figure 5). For the events that actually triggered a streamflow reactionresponse, the correlation between PNORTH and PSOUTH is thus significantly lowerhigher (r=0.69, Table 4). One stronglystrong asymmetric and high intensityvery intense event occurred on July 24th at 6:32 PM (Figure 3). The rainfall map shows a heterogeneous distribution of rainfall, centered close to the outlet in the northern part of the catchment, over 6 out of the 12 stations. One of the rain gauges recorded up to 35.3 mm of rainfall, whereas 1.8 km upstream, half of the stations

- 370 (on the southern and western parts of the catchment) did not record any rainfall. The interpolated amount of rainfall over the basin was 8.0 ± 1.3 mm, and a fast runoff volume between 28.3 and 32.5 mm was measured, resulting in a runoff coefficient between 3.0 and 4.8 that remains difficult to explain. One possible explanation is that important rainfall amounts fell on the north-eastern part of the catchment, over steep slopes that are difficult to access and were therefore not gauged. This event and its streamflow reactionresponse are excluded from further analysis involving the hydrological response (see also Section 4.2)
- and the summary of analysed events in Table 2).

Commenté [AM24]: Added according to Reviewer #1 point 2 and Reviewer #2 point 8b

Commenté [AM25]: Changed according to Reviewer #1 – Point 29 Commenté [AM26]: Mean values added according to Reviewer #1 – Point 30 Commenté [AM27]: Altitude effect added according to Reviewer #1 – Point 8b Commenté [AM28]: Subsets details added according to Reviewer #1, point 3

**Commenté [AM29]:** Changed according to Reviewer #1 – Point 31

Commenté [AM30]: Changed according to Reviewer #1 – point

Commenté [AM31]: Modified according to Reviewer #1 – Point

**Commenté [AM32]:** Modified according to reviewer #1 – Point

**4.1.2 Geomorphological and topographicalStream network distance metrics**

For the 48 recorded rainfall events, allthe three distance metrics  $D_{HILLS}$ ,  $D_{STREAM}$  and  $H_{HAND}$  show a-significantly different distribution of the distances median values if they are computed with respect to the wet network than with respect to the dry network; we can reject for each metric the hypothesis that the distributions they have the same median value for the wet state

- 380 and the dry state with a Wilcoxon rank sum test at level 0.05 (see distributions in Figures S4S6 and S5). The threeS7 of the Supplementary Material part 1). However, each of the distance metrics showshows a strong correlation between its values for the wet and for the dry network state (from 0.94 for  $D_{HAND} H_{HAND}$  to 1.00 for  $D_{STREAM}$ , Figure 7). The between-metric correlation between the distance metricsfor all 48 rainfall events (Table S2 in Supplementary Material part 1) ranges for the wet state range from 0.7678 (DHILLS - DSTREAM) to 0.95 (DHILLS -  $D_{HAND} H_{HAND}$ ) and for the dry state from 0.7570 (DSTREAM -
- 385  $D_{\text{HAND}}\underline{H}_{\underline{HAND}}$  to 0.95 ( $D_{\text{HILLS}} \underline{D}_{\text{HAND}}$ ). For the  $\underline{H}_{\underline{
[revised manuscript text omitted]

**Commenté [AM39]:** Rephrased according to the remark of Reviewer #1 – Point 39

Commenté [AM40]: Added according to Reviewer #1, point 1b

**Commenté [AM41]:** Added according to Reviewer #1 point 2 and Reviewer #2 point 8b

**5 Discussion**

**5.1 Spatial heterogeneity of rainfall**

One of the key identified metrics to characterize the spatial distribution of rainfall; in relation to RC and lag prediction is *I*ASYM splits. Itsplits the catchment into two parts, and averagesaggregates rainfall observations into two values. Among the records showing a strong rainfall asymmetry, 7 out of the 8 events are too small to cause a detectable streamflow reactionresponse (Figure 5), but one does create a reactionstreamflow response although it only rains over half of the 12 rain gauge stations. Despite of this absence of a strong asymmetry in the 14 rainfall events that cause a streamflow reaction, the spatial distribution might play an important role for the explanation of the lag time. The importance of this asymmetry predictor can be related to

565 the fact that it captures the key feature of the spatial catchment organisation in terms of distance to the outlet, drainage density and subsurface storage potential.

The second dominant metric of spatial rainfall distribution to predict the RC and the lag is *D*STREAM (pseudo-dynamic). This suggests that for this catchment, the rainfall distance to the outlet is the overall the dominant predictor for the analyzed streamflow response metrics.

- 570 It is noteworthy that this analysis could be affined by investigating different splitting geometries, e.g. by splitting the catchment into west and east parts, thereby separating the large slopes (west) from the steep slopes (east). This and similar spatial asymmetry metrics are case-specific as they rely on the particular geomorphology and topography of the catchment and are thus not directly applicable to other catchments. In particular *I*ASYM cannot be used as a tool to compare different catchments. The rainfall distance metrics to the stream network (*D*HILLS) and along the stream network (*D*STREAM) were designed here to
- 575 overcome the limitations of the simple asymmetry measure. The prominent role of DSTREAM eompositepseudo-dynamic to explain the lag time and RC underlines the importance of characterizing the spatial heterogeneity in terms of geomorphological distances to the actual stream network, which requires more detailed network expansion analyses in future studies.
   We could expect that in that kind of steep environments, the residence time in hillslopes strongly dominates over residence
- times in the stream network (Nicotina et al., 2008); the fact that  $D_{\text{STREAM}}$  outperforms here  $D_{\text{HILLS}}$  for the prediction of RC and lag time is an interesting result; it underlines may show that even in steep environments, with a priori fast instream processes
- and limited storage, the riparian area and related subsurface exchange processes could play a more prominent role than what we previously thought. The fact that the travel distance in the stream network explains more of the RC variation than  $D_{\rm HILLS}$ might be an indirect effect: the longer the travel distance in the stream network, the more likely are delays due to exchange with groundwater in the riparian area. This implies that along-stream processes might need a better representation in rainfall-
- 585 runoff models, even for small and steep catchments; to date, these processes are often ignored in rainfall-runoff hydrological models at this scale, or are represented with a simple constant velocity transport term (e.g. Schaefli et al., 2014).
  However, future work on the role of water residence time in the stream network will necessarily require more detailed field data on the temporal evolution of the stream network. This will in addition open new perspectives to quantify how the stream

19

**Commenté [AM42]:** Discussion about the splitting line added according to Reviewer #1, point 10.

Commenté [AM43]: Modified according to Reviewer #1 - Point

[revised manuscript text omitted]

---

## Referee Report (RR1)

This was the second time I was involved as a reviewer for this manuscript. The majority of my comments from the previous review round have been addressed. However, there are some open issues and minor comments, which should be replied to/commented on in the manuscript before a publication can be recommended.

Major comment:

The authors provided in the last review round no uniform reply to the reviewers comments. Instead, the replies were scattered among the official reply and author comments in the public discussion of the manuscript. Indeed, stimulating the public discussion is important and I encourage the authors to do so in the future as well. However, for the non-public review the authors should provide a point-by-point reply to the reviewers in one document to avoid that the reviewers waste time 'searching' for the correct replies. I contacted the handling editor after the first submission of the scattered reply asking for a 'complete' reply from the authors. However, there are still replies linking to other documents. I did not take them into account the replies due to a very practical reason: I'm in a train with the printed version and cannot access other documents. Hence all issues which are not addressed directly in this reply remain unsolved for me. I strongly recommend to always provide a 'complete' reply, also for upcoming articles of the first author.

Detailed comments:

L333 The authors state the r-r model is calibrated using "the mean of the 20 stochastic rainfall realizations" (see also L535). If the "mean" is used, all rainfall peak intensities between the stations are smoothed out. The authors should comment why the think this smoothed rainfall time series is an appropriate input time series. Why were not simply all 20 realisations been taken?

Fig. 13. Why not using a simple barplot here? The information would be much easier to catch. Except the stochastic rainfall model I do not see any clear network setup choice here. It is not as clear as authors state in l544-555.

Comments from the previous review round:

Title: „Even event-scale hydrological response benefits from high density rain gauge observations" – Well, the rephrased title can be questioned, since the high network density is of course espially for events important. Why not „Identification of required rain gauge density for hydrological response analysis in small mountainous catchment " (or similar)?

Former comment 1b - Indeed, the value is "only" (please don't get me wrong here) based on prediction of RC and deltaP/Q. While a realistic estimate of these characteristics is valuable, the uncertainties resulting from the final network with 3 rain gauges for these two criteria is not shown and should be added in a later version of the manuscript.

We initially proposed in the public discussion to add two figures showing i) the RC (Figure A) and ii) the lag time ΔP/Q (Figure B) by comparing the values obtained from the best 1-station or 3-station raingauge network vs. the reference value calculated from the full raingauge network. These two scatter plots are shown below, but we finally found that these results were more visible as a polar plots. These two plots are gathered in the Figure 12 of the manuscript in "4.4.3 Optimum network evaluation", and they show the RC and lag time ΔP/Q calculated from the best 1-station and 3-station network compared to the full raingauge network (L534-543). As the stochastic method for generating rainfall fields cannot be used with a number of points as low as 1 or 3 stations, we performed the computations using the Thiessen polygons methods and consequently no error bars are associated to these plots.

-> For me Fig. A and B are much better to interpret and it is easier to „catch" the relevant information in comparison to Fig. 12 and 13 (although it would be useful to have the full network in both figures on the same axis, not once on y (Fig. A) and once on x (Fig. B). Especially when lines cross each other it takes minutes to intepret which line shows the better fit. I also have in mind it it not recommended to use these polar plots for more than three datasets, because they are hard to read then (rule-of-thumb, of course). I'm wondering why ,worst-3-stations' are not implemented in Fig. A and B?

Nevertheless, the Figure C compares the two methods (stochastic vs Thiessen polygons) when the RC and the lag time ΔP/Q are computed from the full raingauge network. We observe for both the RC (Figure A) and ΔP/Q (Figure B) a lower dispersion of values while increasing the density of the raingauge network.

->The „lower dispersion" is hard to see/interpret. From Fig. A and B it seems both station sets lead to similar good results. For Fig. A and B, should there not always be one point for 1-station and 3-station network realted to one value from the x-axis? So the x-axis value of ~250 min has only one point fo the 3-station network, but none for the 1-station network. IF there are points missing, it is hard to judge on the dispersion.

Former comment 1c - In general, I'm missing the runoff peak as important characteristic in the manuscript. Maybe the authors can involve it/comment on it why it was not considered.

This point has been clarified in the public discussion. (https://editor.copernicus.org/index.php?_mdl=msover_md&_jrl=13&_lcm=oc108lcm109w&_acm=get_comm_sup_file&_ms=87052&c=189590&salt=10720610621776386148) and we added the figure showing the peak flows to the Supplementary Material (Figure S5).

-> Issue remains unsolved.

2 - Based on the comment before, the impact of the rain gauge network densities (and rain gauge locations) on the runoff is not analysed. In the additionally uploaded comment the main author states a rainfall-runoff modelling would go beyond the scope of the study. I do not agree with that and 5 recommend this modelling approach to analyse the impact on the resulting runoff itself instead on single runoff statistics. To attribute the spatial rainfall variability, a distributed rainfall-runoff model would be the best solution.

Accordingly to our answer in the public discussion, we added a modelling component to this paper; the model is discussed in the public discussion (https://editor.copernicus.org/index.php?_mdl=msover_md &_jrl=13&_lcm=oc108lcm109w&_acm=get_comm_sup_file&_ms=87052&c=189590&salt=10720610621776386148). Corresponding modifications of the paper are i) at the end of the introduction (L92-94), ii) presenting the model used in the method part "3.6 Rainfall-runoff model" (L335-350), iii) in the results section in "4.4.3 Optimum network evaluation" (L544-555), iv) with the Figure 15 summarizing the results of the different simulations and v) in the Supplementary Material part 1 with the Figure S9 (map of subcatchments), Figure S10 (the results of all simulations per event) and Figure S11 (the results of simulations per event, cumulated over time).

I'm struggling where to find the new supplementary. It was not uploaded with the revised version, I'm afraid I cannot review that part.

Technical corrections / minor comments:

General: Please check all brackets with numerous references, spaces are missing everywhere)

L24 Please add at the end of the sentence: "…for the studied catchment (0.22 rain gauges/km²)."

L207 "precipitated" -> "fallen"

L226 "line" -> "straight line"

L320 "(Section 4.4 and 0)" <- What does "0" refer to?

Eq. 7 Please correct the formula: "0" instead of =:=", root over whole term, providing proper limits for the sum operator.

L329 year for reference is missing

L343-344 repetition (see L226)

L549 space missing

L552 What kind of reference is Beria 2020b? Is it a technical report, a doctoral thesis or an institute publication?

---

## Author Response (AR2)

**Response to editor and reviewers**

We would like to thank the editor and the reviewers for the re-review of our paper and provide below our answers to the overall comment of the editor and to the additional comments of reviewer #1. Our response is in normal, orange font, the comments are in italic.

**Editor comment**

**Dear Authors,**

The two referees who reviewed the revised manuscript are generally happy with the improvements made. Referee 1 raises a couple of comments that remain to be addressed.

A specific comment for the following points:

- title: the reviewer makes a nice title suggestion, consider changing accordingly.

Please see our title suggestion below in the response to the reviewer.

- referee makes a comment about the radar plot and alternative figures you provided. I agree with the reviewer that radar plots are difficult to interpret and a different format would be preferred. Yet I had trouble locating the alternative figures A and B you are referring to in your response. Please check and consider an alternative format to the radar plots.

The figures A and B are referring to the scatter plots from our answer to the point 1b in our merged author-reply document from the public discussion. Thanks for pointing this out. We decided to adapt those figures. Please see below the Figure 4 in our response to the reviewer.

After reading the manuscript, your first discussion point keeps puzzling me. Only 1 of the streamflowgenerating events is asymmetric, yet asymmetry comes out as an important metric to explain variation in RC and lag time. I wonder how such regression results can be valid if the explanatory variable shows so little variability?

Thanks for pointing out the slightly misleading formulations. This comment refers to this the part of the manuscript in "5.1 Spatial heterogeneity of rainfall" copied hereafter (L536-542):

"Among the records showing a strong rainfall asymmetry, 7 out of the 8 events are too small to cause a detectable streamflow response (Figure 5), but one does create a streamflow response although it only rains over half of the 12 rain gauge stations. Despite of this absence of a strong asymmetry in the 14 rainfall events that cause a streamflow response, the regression analysis suggests that the spatial distribution might play an important role for the explanation of the lag time. The importance of this asymmetry predictor can be related to the fact that it captures the key feature of the spatial catchment organisation in terms of distance to the outlet, drainage density and subsurface storage potential."

What we called "asymmetric" is based on the threshold of  $|I_{ASYM}| > 0.33$  which indicates that one part of the catchment received at least 2 times more rainfall than the other part; but  $I_{ASYM}$  is a value between -1 and 1 that describes the spatial rainfall distribution even when  $|I_{ASYM}|

**Distribution of $\mathbf{I}_{\mathbf{ASYM}}$ at the event scale**

*Figure 1. Distribution of I*ASYM for the 48 rainfall events (blue) and the subset of rainfall events associated with a streamflow reaction (brown).

We adapted slightly the formulation (L200-203) in "3.2.2 Spatial rainfall pattern metrics" by changing "We consider a rainfall event as asymmetric when at least 2 times more rain has precipitated over one part of the catchment than over the other, i.e. when IASYM is below -0.33 or above +0.33". It now reads as: "A value of -0.33 or 0.33 indicates that the catchment received at least 2 times more rain over one part of the catchment than the other."

And in "4.1.1 Areal rainfall and asymmetry" by changing (L347-348) "Interestingly, strong spatial asymmetry mainly affects events with low rainfall amounts, with 7 out of 8 asymmetric events (when  $|I_{ASYM}| > 0.33$ ) receiving below 5 mm (Figure 5). " into "Interestingly, strong spatial asymmetry mainly affects events with low rainfall amounts, with 7 out of 8 events with  $|I_{ASYM}| > 0.33$  receiving below 5 mm (Figure 5)."

Quite a lot of papers have looked at this relation between rainfall spatial variability (storm position, movement, relative to the catchment and streamflow network), some of which you cite in the Introduction, yet it's worth putting your results in relation to the literature also in the Discussion part. The one paper you cite here is a modeling study, while there are several data-driven studies on the topic of "storm position and hydrologic response" that are more closely related to your study (see for instance work by J.A. Smith and his group).

Thank you for pointing this out. We added one sentence introducing 2 references in the discussion part at the end of the "5.1 Spatial heterogeneity of rainfall" (L568-571):

"Finally, we would like to point out here that this result on the prominent role of travel time along the stream network opens interesting new analogies with urban hydrology, where introduction times to the network are typically short (Smith et al., 2013). Future work might show what methods from urban hydrology (Cristiano et al., 2017) could be transposed to the analysis of spatial rainfall variability in small alpine catchments."

**Reviewer 1: Major comment:**

The authors provided in the last review round no uniform reply to the reviewers comments. Instead, the replies were scattered among the official reply and author comments in the public discussion of the manuscript.

We decided to refer to the public discussion to emphasize which parts of the revision were exactly as discussed in the Online Discussion. We did not consider that this might be impractical for an offline review and would like to apologize for this.

Indeed, stimulating the public discussion is important and I encourage the authors to do so in the future as well. However, for the non-public review the authors should provide a point-by-point reply to the reviewers in one document to avoid that the reviewers waste time 'searching' for the correct replies. I contacted the handling editor after the first submission of the scattered reply asking for a 'complete' reply from the authors. However, there are still replies linking to other documents.

The handling editor contacted us indeed, but we misunderstood her comment and did not understand that the links to the public discussion were the problem. Instead we tried to solve some discrepancies that could exist between our author-reply in the Public Discussion and the author-reply document we uploaded with the revised manuscript. As mentioned above, in our answers we referred to the public discussion to underline that the implemented solution was exactly the one discussed previously, but this was apparently not a good choice.

I did not take them into account the replies due to a very practical reason: I'm in a train with the printed version and cannot access other documents. Hence all issues which are not addressed directly in this reply remain unsolved for me. I strongly recommend to always provide a 'complete' reply, also for upcoming articles of the first author.

The two instances where we referred to the Online Discussion are i) the point concerning peak flow (point 1c for reviewer #1, point 8a for reviewer #2) and ii) the comment on modelling (point 2 for reviewer #1, point 8b for reviewer #2). The answers to these points are reported in Appendix 1 and 2 of this document, respectively.

Furthermore, we realize just now that some citations from the track change manuscript in our answer include several broken links (to sections) appearing when converting our document into PDF (i.e. *"3.3.1Erreur ! Source du renvoi introuvable."*). We apologize for having missed this during the revision submission. Those are just references to sections, i.e. do not contain further information.

**Reviewer 1: Detailed comments:**

1 - L333 The authors state the r-r model is calibrated using "the mean of the 20 stochastic rainfall realizations" (see also L535). If the "mean" is used, all rainfall peak intensities between the stations are smoothed out. The authors should comment why they think this smoothed rainfall time series is an appropriate input time series. Why were not simply all 20 realisations been taken?

Thanks for this important comment; it is important to clarify that we used the temporal mean (no specified in the paper) and furthermore, the 20 stochastic realizations are all conditioned on the precipitation observed at the stations and they accordingly do not smooth out the observed peaks as illustrated in the Figure 2 below.

We updated the manuscript as follows (L324-330):

Old: "For calibration, the model is run using the mean of the 20 stochastic rainfall realizations as reference input; it is then calibrated against observed runoff (i.e. discharge - baseflow) through likelihood maximization assuming that the model residuals are normally distributed (e.g. Schaefli et al., 2007)."

New: "The model is calibrated against observed runoff (i.e. discharge - baseflow) through likelihood maximization assuming that the model residuals are normally distributed (e.g. Schaefli et al., 2007). The reference input field for model calibration is the mean of the 20 stochastic rainfall realizations at each time step (note since all realizations are conditioned on the observed precipitation events, this mean preserves the individual observed peaks of precipitation)."

Figure 2. Areal rainfall for 15 rainfall events having a streamflow reaction. For each event is represented the 20 areal rainfall realizations (blue) and the mean value of the 20 realizations (orange).

2 - Fig. 13. Why not using a simple barplot here? The information would be much easier to catch. Except the stochastic rainfall model I do not see any clear network setup choice here. It is not as clear as authors state in I544-555.

Thanks for pointing this out, we decided to use a simple plot to present this result. The Figure 13 of the manuscript has been updated by the Figure 3 (and figure caption) below.